# AdaCubic: An Adaptive Cubic Regularization Optimizer for Deep Learning

**Ioannis Tsingalis**                                    *tsingalis@csd.auth.gr*
*Department of Informatics*
*Aristotle University of Thessaloniki, Greece*

**Constantine Kotropoulos**                              *costas@csd.auth.gr*
*Department of Informatics*
*Aristotle University of Thessaloniki, Greece*

**Corentin Briat**                                       *corentin@briat.info*
*School of Life Sciences*
*University of Applied Sciences Northwestern Switzerland, Muttenz, Switzerland*

**Reviewed on OpenReview:** *https://openreview.net/forum?id=pZBQ7J37lk*

## Abstract

A novel regularization technique, ADACUBIC, is proposed that adapts the weight of the cubic term. The heart of ADACUBIC is an auxiliary optimization problem with cubic constraints that dynamically adjusts the weight of the cubic term in Newton's cubic regularized method. We use Hutchinson's method to approximate the Hessian matrix, thereby reducing computational cost. We demonstrate that ADACUBIC inherits the cubically regularized Newton method's local convergence guarantees. Our experiments in Computer Vision, Natural Language Processing, and Signal Processing tasks demonstrate that ADACUBIC outperforms or competes with several widely used optimizers. Unlike other adaptive algorithms that require hyperparameter fine-tuning, ADACUBIC is evaluated with a fixed set of hyperparameters, rendering it a highly attractive optimizer in settings where fine-tuning is infeasible. This makes ADACUBIC an attractive option for researchers and practitioners alike. To our knowledge, ADACUBIC is the first optimizer to leverage cubic regularization in scalable deep learning applications.

https://github.com/iTsingalis/AdaCubic

## 1 Introduction

Deep Neural Networks (DNNs) have demonstrated strong performance on a variety of machine learning tasks (Pouyanfar et al., 2018; Dargan et al., 2020). DNN models are non-convex (Jin et al., 2021; Danilova et al., 2022; Pooladzandi et al., 2022b). Accordingly, saddle points may arise during the optimization procedure (Bedi et al., 2021). In Dauphin et al. (2014), it is shown that the saddle points affect the efficiency of a DNN. Therefore, methods that avoid saddle points are necessary, as discussed next.

The Cubic Regularized (CR) Newton's method was introduced in (Nesterov & Polyak, 2006). This method effectively circumvents saddle points in a non-convex setting. The first research direction focuses on carefully selecting the regularization parameter for the cubic term. In Cartis et al. (2011a), an Adaptive Regularized Cubic (ARC) method is presented where the cubic regularization term is adapted dynamically, similarly to the radius in the Trust Region methods (Conn et al., 2000). To mitigate the computational burden of deriving the Hessian matrix and the gradient in ARC, Carmon & Duchi (2019) solves the CR sub-problem using gradient descent. Alternatively, one solves the cubic sub-problem using a subsampled gradient and

a Hessian-vector product (Tripuraneni et al., 2018). In Kohler & Lucchi (2017), a subsampled scheme for the gradient and the Hessian matrix is exploited, achieving the same convergence rate as ARC. In Wang et al. (2020b), momentum information is utilized to improve the convergence rate of CR. Inspired by Fang et al. (2018), a recursive stochastic variance reduced CR method is proposed in Zhou & Gu (2020), yielding a better convergence rate than that reported in (Tripuraneni et al., 2018). In Huang et al. (2022), the CR method was applied to solve unconstrained convex-concave saddle point problems.

In a second research direction, it has been demonstrated that injecting a random perturbation whenever a saddle point is encountered can facilitate escape from saddle points. In Ge et al. (2015); Jin et al. (2017), both negative curvature and random perturbation are applied to Stochastic Gradient Descent (SGD) to escape saddle points. Within the same scope, in Allen-Zhu (2018); Royer & Wright (2018), it is shown that negative curvature and random perturbation can be used to find an $(\epsilon_g, \epsilon_H)$-stationary point faster than the first-order methods. A drawback of these methods is the need to compute the smallest eigenvalue of the Hessian matrix and the corresponding eigenvector. Several methods have been proposed to address this limitation. In Li (2019), it is shown that a perturbed version of Stochastic Recursive Gradient Descent, without using the Hessian matrix information, also converges to an $(\epsilon_g, \epsilon_H)$-stationary point. In Allen-Zhu & Li (2018); Zhang & Li (2021), a robust Hessian matrix power method is proposed to compute the negative curvature near saddle points, yielding faster convergence than the standard perturbed gradient descent methods. In Chen et al. (2022), to achieve a better convergence rate, the average movement of the iterates is controlled by a step-size shrinkage scheme (Li, 2019).

In a third research direction, escaping saddle points relies on momentum information (Wang et al., 2021b). First-order methods with random initialization and momentum information are shown to be able to escape saddle points in (Sun et al., 2019). A greater momentum in SGD enlarges the projection to an escape direction, leading to a fast saddle point escape (Wang et al., 2020a). In Levy et al. (2021), a parameter-free recursive momentum method is proposed for non-convex optimization. In Wang et al. (2020a), it is shown that acceleration can be achieved for non-quadratic functions under Polyak-Łojasiewicz condition and non-convexity. In Wang et al. (2021a), it is presented that the momentum term accelerates the training of a one-layer-wide ReLU network.

A fourth research direction employs variance reduction to escape saddle points. In Allen-Zhu & Hazan (2016), the minimization of the sum of smooth functions is studied, where variance reduction is applied to speed up convergence in both the stochastic and the deterministic case. A general variance-reduction estimation method is introduced that is not restricted to gradients only (Fang et al., 2018). This method has been applied to numerous problems and has achieved convergence rates superior to those reported in (Allen-Zhu & Hazan, 2016). In Nguyen et al. (2017a), a recursive gradient estimator for convex optimization is introduced. The latter estimator is then extended to non-convex problems in (Nguyen et al., 2017b). In Ge et al. (2019), the first variance reduction technique not based on a separate negative curvature search subroutine is proposed.

Last but not least, a second-order optimizer, called AdaHessian, has been introduced in (Yao et al., 2021). AdaHessian is based on the Adaptive Moments Estimation (Adam) optimizer (Kingma & Ba, 2015) and leverages Hutchinson's method to approximate the curvature information with low computational cost (Bekas et al., 2007). The convergence rate of AdaHessian for a strongly convex and smooth loss function can be found in (Yao et al., 2021; Pooladzandi et al., 2022a). Based on the *Hessian power*, the convergence rate of AdaHessian for a strongly convex and smooth loss function matches that of either gradient descent or Newton's method (Jahani et al., 2021; Sadiev et al., 2022).

In this paper, we focus on the CR Newton method (Nesterov & Polyak, 2006) and propose a novel algorithm that dynamically adapts the weight of the cubic term in the cubic subproblem. The adaptation of the cubic term is achieved by utilizing an auxiliary cubically constrained optimization problem. The proposed algorithm, AdaCubic, leverages the advantages of CR theory and Hutchinson's estimation technique. In more detail, the **contributions** of this paper are:

- A novel method is proposed that automatically adapts the regularization parameter $M$ in the cubic sub-problem and avoids saddle points. The primary theoretical contributions concerning the adaptation of $M$ are encapsulated in Lemma 2, Theorems 1 and 2, as well as the methodologies detailed

in Algorithms 1 and 2. Figure 6, in Appendix A, depicts how the key lemmata, theorems, and corollaries are logically connected throughout Sections 2 to 4 and Appendices B.1 to B.13.

- The proposed optimizer does not need the computation of Krylov sub-space Wang et al. (2020b); Zhou & Gu (2020); Kohler & Lucchi (2017) or the calculation of the smallest eigenvalue Allen-Zhu & Li (2018); Allen-Zhu (2018); Park et al. (2020) to obtain an optimal solution. The optimal solution is obtained by leveraging Hutchinson's method that approximates the diagonal of the Hessian matrix (Bekas et al., 2007). In this way, the proposed method exhibits low memory complexity.

- The convergence rate of ADACUBIC is established by exploiting the diagonal structure of the approximate Hessian matrix, which is computed using data batches. This property makes ADACUBIC particularly appealing for deep learning applications.

- ADACUBIC is tested on Computer Vision, Natural Language Processing, and Signal Processing tasks, demonstrating a competitive or better performance when compared to SGD Robbins & Monro (1951), ADAM Kingma & Ba (2015), and ADAHESSIAN (Yao et al., 2021) optimizers. It should be noted that the parametrization of ADACUBIC is performed by employing a well-known set of parameters used in Trust Region algorithms (Conn et al., 2000, Section 17.1). These parameters are used universally in the experimental evaluation, casting ADACUBIC as an attractive optimizer when fine-tuning is prohibitive.

The paper is organized as follows. Section 2 details the proposed optimization framework. The convergence analysis of the proposed optimization framework is demonstrated in Section 3. Section 4 presents the algorithms that compute the optimal solution of the proposed optimization framework in Section 2. Experimental results, computational complexity, and conclusions are presented in Sections 5, 6, and 7, respectively.

## 2 Proposed Optimization Framework

**Outline.** Section 2.1 introduces the fundamental definitions used throughout the paper, including the basic formulation of the CR method, which serves as a core building block of the proposed framework. Section 2.2 then introduces an auxiliary constrained optimization problem that forms the foundation of the ADACUBIC. The key intuition is to reformulate the classical CR method as a constrained problem in which the cubic regularization term appears explicitly as a constraint. By leveraging Lagrange multiplier theory, this reformulation yields an adaptive update mechanism in which the strength of the cubic regularization term of the CR method is automatically adjusted during optimization. To derive this update mechanism Lemmata 1, 2, Theorem 1, Corollary 1, and Theorem 2 are introduced.

Lemma 1 establishes that the auxiliary constrained problem admits a global minimizer and ensures that each optimization step is well defined. Lemma 2 is used to establish Theorem 1, which in turn is used to derive Corollary 1. Corollary 1 shows that the auxiliary optimization problem is characterized by *strong duality* (Boyd & Vandenberghe, 2004, Section 5.4). The latter theoretical results are then combined to derive Theorem 2, which provides the basis to replace the fixed cubic regularization parameter of the CR method with an adaptive one and finally derive the ADACUBIC optimizer presented in Section 4.

### 2.1 Preliminaries

To simplify notation, the iteration index $k$ in $\mathbf{x}_k \in \mathbb{R}^d$ will be explicitly denoted when necessary. Otherwise, it will be suppressed. Let $\nabla_{\mathbf{x}}^2 f(\mathbf{x}_k)$ and $\nabla_{\mathbf{x}} f(\mathbf{x}_k)$ be the Hessian matrix and the gradient of the function $f(\mathbf{x}_k)$ with respect to (w.r.t.) $\mathbf{x}$. In the following, the subscript $\mathbf{x}$ in $\nabla_{\mathbf{x}}^2$ and $\nabla_{\mathbf{x}}$ is omitted for simplicity, resulting in $\nabla^2 f(\mathbf{x}_k)$ and $\nabla f(\mathbf{x}_k)$, respectively. The spectrum of the symmetric $d \times d$ matrix $\nabla^2 f(\mathbf{x}_k)$ is denoted by $\lambda(\nabla^2 f(\mathbf{x}_k)) = \{\lambda_i(\nabla^2 f(\mathbf{x}_k))\}_{i=1}^d$. Suppose that the eigenvalues are sorted in descending order, i.e.,

$$\lambda_1(\nabla^2 f(\mathbf{x}_k)) \geq \cdots \geq \lambda_d(\nabla^2 f(\mathbf{x}_k)) = \lambda_{\min}(\nabla^2 f(\mathbf{x}_k)). \tag{1}$$

If $\nabla^2 f(\mathbf{x}_k)$ is indefinite, i.e.,

$$\lambda_d(\nabla^2 f(\mathbf{x}_k)) < 0 \quad \text{and} \quad \lambda_i(\nabla^2 f(\mathbf{x}_k)) > 0, \quad i < d, \tag{2}$$

then $f(\mathbf{x})$ is non-convex. The notations $\nabla^2 f(\mathbf{x}_k) \succeq 0$ or $\nabla^2 f(\mathbf{x}_k) \succ 0$ indicate that the Hessian matrix is positive semi-definite or positive definite, respectively. Let $\partial_\tau \mathscr{F}$ be the partial derivative of a function $\mathscr{F} \colon \mathbb{R} \to \mathbb{R}$ w.r.t. the real-valued variable $\tau$. Moreover, let $\odot$ and $\oslash$ denote the element-wise product and division, respectively. In addition, let $\mathrm{diag}(\nabla^2 f(\mathbf{x}_k)) = \left[[\nabla^2 f(\mathbf{x}_k)]_{11}, \ldots, [\nabla^2 f(\mathbf{x}_k)]_{dd}\right]^T \in \mathbb{R}^d$ be a column vector containing the diagonal elements of the Hessian matrix and $\mathrm{Diag}(\nabla^2 f(\mathbf{x}_k)) = \nabla^2 f(\mathbf{x}_k) \odot \mathbf{I}$ a $d \times d$ stand for a diagonal matrix retaining the diagonal elements of the Hessian matrix, where $\mathbf{I}$ is the identity matrix. $\|\cdot\|_2$ refers to the vector $\ell_2$ norm or to the spectral norm of a matrix. The $d$-dimensional vector of ones is denoted by $\mathbb{1}_d$.

A non-convex optimization problem is defined by

$$\min_{\mathbf{x} \in \mathbb{R}^d} \quad f(\mathbf{x}) \triangleq \frac{1}{n} \sum_{\ell=1}^{n} f_\ell(\mathbf{x}), \tag{3}$$

where $f \colon \mathbb{R}^d \to \mathbb{R}$ and $f_\ell \colon \mathbb{R}^d \to \mathbb{R}$ are non-convex functions. Solving (3) is generally NP-Hard (Murty & Kabadi, 1987; Hillar & Lim, 2013). As a result, a reasonable goal is to find an $\epsilon$-stationary point, i.e., an approximate local minimum, by checking $\|\nabla f(\mathbf{x})\|_2 \le \epsilon$, where $\nabla f(\mathbf{x}) \in \mathbb{R}^d$ is treated as a column vector. However, $\epsilon$-stationary points can be non-degenerate saddle points (i.e., the Hessian matrix at all saddle points has negative eigenvalues) or even local extrema in non-convex optimization. To avoid saddle points, second-order methods are used to find an $(\epsilon_g, \epsilon_H)$-stationary point by checking

$$\|\nabla f(\mathbf{x})\|_2 \le \epsilon_g \quad \text{and} \quad \lambda_{\min}\left(\nabla^2 f(\mathbf{x})\right) \ge -\epsilon_H, \tag{4}$$

where $\epsilon_g, \epsilon_H > 0$, and $\lambda_{\min}\left(\nabla^2 f(\mathbf{x})\right)$ denotes the minimal eigenvalue of the Hessian matrix. CR technique is designed to avoid saddle points (Nesterov & Polyak, 2006). Starting from an arbitrary point $\mathbf{x}_0$, the update rule of CR that solves (3) is written as

$$\mathbf{s}_{k+1} = \arg\min_{\mathbf{s} \in \mathbb{R}^d} m_M(\mathbf{s}) \tag{5}$$

where

$$m_M(\mathbf{s}) \triangleq f(\mathbf{x}_k) + \nabla f(\mathbf{x}_k)^T \mathbf{s} + \frac{1}{2} \mathbf{s}^T \nabla^2 f(\mathbf{x}_k) \, \mathbf{s} + \frac{M}{6} \|\mathbf{s}\|_2^3, \tag{6}$$

$\mathbf{x}_{k+1} \triangleq \mathbf{x}_k + \mathbf{s}_{k+1}$, and $M > 0$ is the regularization parameter that can be fixed or adaptive (Nesterov & Polyak, 2006; Cartis et al., 2011a). In the following sections, the problem formulation and its solution are presented.

## 2.2 Problem Formulation

**Auxiliary Problem.** We are interested in developing an adaptive method for selecting $M$ in (5). To do so, we introduce the auxiliary constrained optimization problem

$$\begin{aligned} \arg\min_{\mathbf{s} \in \mathbb{R}^d} \quad & \hat{m}(\mathbf{s}) \triangleq f(\mathbf{x}_k) + \nabla f(\mathbf{x}_k)^T \mathbf{s} + \frac{1}{2} \mathbf{s}^T \nabla^2 f(\mathbf{x}_k) \, \mathbf{s} \\ \text{subject to} \quad & g_\xi(\mathbf{s}) \triangleq \frac{1}{6}\left(\|\mathbf{s}\|_2^3 - \xi\right) \le 0, \end{aligned} \tag{7}$$

for $\xi \ge 0$. The Lagrangian function of (7) is

$$\mathcal{L}_\xi(\mathbf{s}, \nu) = f(\mathbf{x}_k) + \nabla f(\mathbf{x}_k)^T \mathbf{s} + \frac{1}{2} \mathbf{s}^T \nabla^2 f(\mathbf{x}_k) \, \mathbf{s} + \frac{\nu}{6}\left(\|\mathbf{s}\|_2^3 - \xi\right), \tag{8}$$

where $\nu$ is the Lagrange multiplier. Let $\Omega = \{\mathbf{s} \mid g_\xi(\mathbf{s}) \le 0\}$. The minimizer we are seeking in (7) lies either within the interior of $\Omega$ (i.e., $g_\xi(\mathbf{s}) < 0$) or lies on the boundary of $\Omega$ (i.e., $g_\xi(\mathbf{s}) = 0$). Lemma 1 is an immediate result of the previous discussion.

**Lemma 1.** *A vector* $\mathbf{s}^*$ *is a minimizer of* $\hat{m}(\mathbf{s})$ *subject to* $\|\mathbf{s}^*\|_2^3 \leq \xi$ *if and only if satisfies*

$$\left( \nabla^2 f(\mathbf{x}_k) + \frac{\nu^*}{2} \|\mathbf{s}^*\|_2 \ \mathbf{I} \right) \mathbf{s}^* = -\nabla f(\mathbf{x}_k), \tag{9}$$

$$\nabla^2 f(\mathbf{x}_k) + \frac{\nu^*}{2} \|\mathbf{s}^*\|_2 \ \mathbf{I} \succeq 0, \tag{10}$$

*and* $\nu^* \left( \|\mathbf{s}^*\|_2^3 - \xi \right) = 0$*, where* $\nu^* \geq 0$*. If* $\nabla^2 f(\mathbf{x}_k) + \frac{\nu^*}{2} \|\mathbf{s}^*\|_2 \ \mathbf{I} \succ 0$*, then the minimizer* $\mathbf{s}^*$ *is unique.*

The condition $\nu^* \left( \|\mathbf{s}^*\|_2^3 - \xi \right) = 0$ in Lemma 1 is called Complementary Slackness (CS) condition. The proof of Lemma 1 can be found in Appendix B.1.

**Definition 1.** *For some* $\nu \geq 0$*, denote*

$$\mathcal{D}_\nu \triangleq \left\{ r \mid \nabla^2 f(\mathbf{x}_k) + \frac{\nu \, r}{2} \, \mathbf{I} \succ \mathbf{0}, \quad r > 0 \right\}. \tag{11}$$

Next, it is proven that problem (7) is characterized by strong duality. To do so, Lemma 2 and Theorem 1 are introduced. Lemma 2, is used as a preliminary result to prove Theorem 1. Corollary 1 establishes the strong duality of problem (7) as an immediate outcome of Theorem 1.

**Lemma 2** (Proof in Appendix B.2)**.** *For* $r \in \mathcal{D}_\nu$ *we have*

$$\min_{\mathbf{s} \in \mathbb{R}^d} \mathcal{L}_\xi(\mathbf{s}, \nu) = \max_{r \in \mathcal{D}_\nu} \mathscr{L}_\xi(\mathbf{s}(\nu, r), \nu, r), \tag{12}$$

*where*

$$\mathscr{L}_\xi(\mathbf{s}(\nu, r), \nu, r) = -\frac{1}{2} \nabla f(\mathbf{x}_k)^T \left( \nabla^2 f(\mathbf{x}_k) + \frac{\nu \, r}{2} \, \mathbf{I} \right)^{-1} \nabla f(\mathbf{x}_k) - \frac{\nu}{6} \xi - \frac{\nu}{12} r^3. \tag{13}$$

*For* $r \in \mathcal{D}_\nu$ *the direction*

$$\mathbf{s}(\nu, r) = -\left( \nabla^2 f(\mathbf{x}_k) + \frac{\nu \, r}{2} \, \mathbf{I} \right)^{-1} \nabla f(\mathbf{x}_k), \tag{14}$$

*satisfies*

$$\mathcal{L}_\xi(\mathbf{s}(\nu, r), \nu) = \mathscr{L}_\xi(\mathbf{s}(\nu, r), \nu, r) + \frac{4}{3\nu} \frac{\left( r + 2 \|\mathbf{s}(\nu, r)\|_2 \right)}{\left( r + \|\mathbf{s}(\nu, r)\|_2 \right)^2} \left( \partial_r \mathscr{L}_\xi(\mathbf{s}(\nu, r), \nu, r) \right)^2. \tag{15}$$

*For* $r^* \in \mathcal{D}_\nu$ *that maximizes* $\max_{r \in \mathcal{D}_\nu} \mathscr{L}_\xi(\mathbf{s}(\nu, r), \nu, r)$*,*

$$\mathbf{s}(\nu, r^*) = -\left( \nabla^2 f(\mathbf{x}_k) + \frac{\nu}{2} \|\mathbf{s}(\nu, r^*)\|_2 \ \mathbf{I} \right)^{-1} \nabla f(\mathbf{x}_k) \tag{16}$$

*is the minimizer of* $\min_{\mathbf{s} \in \mathbb{R}^d} \mathcal{L}_\xi(\mathbf{s}, \nu)$ *in (12).*

**Theorem 1** (Proof in Appendix B.3)**.** *We have*

$$\min_{\mathbf{s} \in \mathbb{R}^d} \max_{\nu \geq 0} \mathcal{L}_\xi(\mathbf{s}, \nu) = \max_{\nu \geq 0, \ r \in \mathcal{D}_\nu} \mathscr{L}_\xi(\mathbf{s}(\nu, r), \nu, r), \tag{17}$$

*where* $\mathscr{L}_\xi(\mathbf{s}(\nu, r), \nu, r)$ *is defined in (13). For* $r \in \mathcal{D}_\nu$*, the direction*

$$\mathbf{s}(\nu, r) = -\left( \nabla^2 f(\mathbf{x}_k) + \frac{\nu \, r}{2} \, \mathbf{I} \right)^{-1} \nabla f(\mathbf{x}_k), \tag{18}$$

*satisfies*

$$\mathcal{L}_\xi(\mathbf{s}(\nu, r), \nu) = \mathscr{L}_\xi \big( \mathbf{s}(\nu, r), \nu, r \big) - \nu \ \partial_\nu \mathscr{L}_\xi \big( \mathbf{s}(\nu, r), \nu, r \big) + \frac{4}{3\nu} \frac{(r + 2 \ \|\mathbf{s}(\nu, r)\|_2)}{(r + \|\mathbf{s}(\nu, r)\|_2)^2} \big( \partial_r \mathscr{L}_\xi(\mathbf{s}(\nu, r), \nu, r) \big)^2. \tag{19}$$

*For the optimal values* $\nu^*$ *and* $r^* \in \mathcal{D}_\nu$ *that maximize* $\max_{\nu \geq 0, \ r \in \mathcal{D}_\nu} \mathscr{L}_\xi(\mathbf{s}(\nu, r), \nu, r)$*,*

$$\mathbf{s}^*(\nu^*, r^*) = -\left( \nabla^2 f(\mathbf{x}_k) + \frac{\nu^*}{2} \|\mathbf{s}(\nu^*, r^*)\|_2 \ \mathbf{I} \right)^{-1} \nabla f(\mathbf{x}_k), \tag{20}$$

*is the minimizer of* $\min_{\mathbf{s} \in \mathbb{R}^d} \max_{\nu \geq 0} \mathcal{L}_\xi(\mathbf{s}, \nu)$ *in (17), i.e., the optimal* $\mathbf{s}^*$ *in Lemma 1.*

**Corollary 1.** *The constrained optimization problem (7) is characterized by strong duality, i.e.,*

$$\min_{\mathbf{s}\in\mathbb{R}^d} \max_{\nu\geq 0} \mathcal{L}_\xi(\mathbf{s}, \nu) = \max_{\nu\geq 0} \min_{\mathbf{s}\in\mathbb{R}^d} \mathcal{L}_\xi(\mathbf{s}, \nu). \tag{21}$$

*Proof.* See the proof of Theorem 1. □

Given Corollary 1, the equivalence between problems (5) and (7) is established in Theorem 2. The equivalence implies that both problems have the same optimum.

**Theorem 2** (Proof in Appendix B.4). *Let $\nu^*$ be the optimal dual variable of the constrained optimization problem (7). The following optimization problems*

$$\min_{\mathbf{s}\in\mathbb{R}^d} m_M(\mathbf{s}) \text{ and } \min_{\mathbf{s}\in\mathbb{R}^d} \hat{m}(\mathbf{s}) \text{ subject to } g_\xi(\mathbf{s}) \leq 0 \tag{22}$$

*are equivalent w.r.t. the optimal solution $\mathbf{s}^*$, when $M = \nu^*$ and $\xi = \|\mathbf{s}^*\|_2^3$.*

## 3 Local Convergence Analysis

**Outline.** This section provides the local convergence analysis of Algorithm 1. It begins with Assumption 1, which defines the Lipschitz continuity constants for $f_i(\mathbf{x})$, $\nabla f_i(\mathbf{x})$, and $\nabla^2 f_i(\mathbf{x})$. Subsequently, Theorem 3 establishes the local convergence of Algorithm 1 when using the exact gradient and Hessian matrix.

Adequate agreement between the exact gradient $\nabla f(\mathbf{x}_k)$ and the approximate gradient $\mathbf{g}_k$ is established in Assumption 2. This assumption is grounded on Wang et al. (2019, Assumption 2) and facilitates the approximation of the gradient using a sampling scheme in Lemma 5, akin to the one outlined in (Kohler & Lucchi, 2017, Theorem 7).

A sufficient agreement between the exact diagonal Hessian matrix $\text{Diag}(\nabla^2 f(\mathbf{x}_k))$ and the approximate diagonal Hessian matrix $\mathbf{B}_k$ in (30), is established in Assumption 3. Assumption 3 is a direct application of (Wang et al., 2019, Assumption 2). Additionally, Assumption 3 supports Lemma 6, while Lemma 3 is pivotal for establishing Lemma 4. Lemma 4 is used in Lemmata 5, 6, and Corollary 2. Lemmata 5 and 6 provide the deviation bounds for the gradient and Hessian matrix, along with the corresponding conditions required for these bounds to hold. These conditions are consolidated in Corollary 2, which ensures the validity of both deviation bounds.

The analysis concludes by discussing the local convergence of the sub-sampled case, where the exact gradient $\nabla f(\mathbf{x}_k)$ and diagonal Hessian matrix $\text{Diag}(\nabla^2 f(\mathbf{x}_k))$ are replaced with their sub-sampled approximations $\mathbf{g}_k$ and $\mathbf{B}_k$, respectively.

**Convergence Analysis.** Next, we begin with the main results of the analysis. Assumption 1 is commonly used in previous works (Nesterov & Polyak, 2006; Cartis et al., 2011a;b; Kohler & Lucchi, 2017) and is applied here in combination with Remark 1.

Let $\mathcal{F} \subseteq \mathbb{R}^d$ be a closed convex set with a non-empty interior. Let $\mathbf{x}_0 \in \text{int } \mathcal{F}$ be a starting point of the iterative optimization scheme in the interior of $\mathcal{F}$.

**Assumption 1** (Continuity). *The convergence analysis is based on the following assumptions:*

- *The functions $f_i(\mathbf{x})$ are twice-continuously differentiable and bounded from below by $f_i^{\text{low}}$.*

- *The functions $f_i(\mathbf{x})$, $\nabla f_i(\mathbf{x})$, and $\nabla^2 f_i(\mathbf{x})$ are Lipschitz continuous in $\mathcal{F}$ with Lipschitz constants $L_f$, $L_g$, and $L_H$, respectively.*

**Remark 1.** *Due to the triangle inequality, it follows that the Lipschitz continuity also holds for $f(\mathbf{x})$, $\nabla f(\mathbf{x})$, and $\nabla^2 f(\mathbf{x})$, with Lipschitz constants $L_f$, $L_g$, and $L_H$, respectively. In addition, given Assumption 1, $f(\mathbf{x})$ is also lower bounded by some $f^{\text{low}}$.*

By leveraging Theorem 2, the iteration complexity of Algorithm 1 is equivalent to that performed by the cubic regularization method in (Nesterov & Polyak, 2006). Theorem 3 analyses the iteration complexity of Algorithm 1 by adapting the analysis from Nesterov & Polyak (2006, Theorem 1), when $\mathrm{Diag}(\nabla^2 f(\mathbf{x}))$ replaces $\nabla^2 f(\mathbf{x})$.

**Theorem 3** (Proof in Appendix B.12). *Suppose Assumption 1 holds. Also, let the sequence $\mathbf{x}_i$, with $i \geq 0$, be generated by Algorithm 1 when $\mathrm{Diag}(\nabla^2 f(\mathbf{x}_i))$ is used. Then, after $k$ iterations, the sequence $\{\mathbf{x}_i\}_{i\geq 1}$ satisfies*

$$\min_{1 \leq i \leq k} \|\nabla f(\mathbf{x}_i)\|_2 \leq \mathcal{O}\left(\frac{1}{k^{2/3}}\right). \tag{23}$$

If we want to find the iteration $k$ that satisfies $\min_{1 \leq i \leq k} \|\nabla f(\mathbf{x}_i)\|_2 \leq \epsilon$, we upper bound (23) by $\epsilon$ and we conclude that

$$k \geq \mathcal{O}\left(\frac{1}{\epsilon^{3/2}}\right). \tag{24}$$

**Deviation Bounds.** Rather than utilizing deterministic gradient and Hessian information, we can employ estimates of the gradient, the Hessian matrix, and the loss function, which are derived from an independent set of points $\mathcal{B}_k$, i.e.,

$$\mathbf{g}_k = \frac{1}{|\mathcal{B}_k|} \sum_{i \in \mathcal{B}_k} \nabla f_i(\mathbf{x}_k), \tag{25}$$

$$\mathbf{H}_k = \frac{1}{|\mathcal{B}_k|} \sum_{i \in \mathcal{B}_k} \nabla^2 f_i(\mathbf{x}_k), \tag{26}$$

and

$$F(\mathbf{x}_k) = \frac{1}{|\mathcal{B}_k|} \sum_{i \in \mathcal{B}_k} f_i(\mathbf{x}_k). \tag{27}$$

**Assumption 2** (Sufficient agreement of $\mathbf{g}_k$ and $\nabla f(\mathbf{x}_k)$). *There is a constant $C_g > 0$ such that the inexact gradient $\mathbf{g}_k$ satisfies, for all $k \geq 0$,*

$$\|\mathbf{g}_k - \nabla f(\mathbf{x}_k)\|_2 \leq C_g \|\mathbf{s}_k\|_2^2. \tag{28}$$

For some $\mathbf{x}_k$, the computation of the Hessian matrix $\mathbf{H}_k \in \mathbb{R}^{d \times d}$ in (26) is expensive due to the large size $d$ of $\mathbf{x}_k$. Only the Hessian-vector product can be calculated at a reasonable computational complexity (Pearlmutter, 1994). Let $\mathcal{H}_k : \mathbb{R}^d \to \mathbb{R}^d$ be a function such that $\mathcal{H}_k(\mathbf{v}) \triangleq \mathbf{H}_k \mathbf{v}$, where $\mathbf{H}_k$ is not accessible. Given the Hessian-vector product operator $\mathcal{H}_k$, the diagonal of $\mathbf{H}_k$, i.e., $\mathbf{h}_k \triangleq \mathrm{diag}(\mathbf{H}_k)$, is approximated by the Hutchinson's method as Bekas et al. (2007)

$$\mathbf{b}_k = \left[\sum_{s=1}^{\mathcal{S}} \mathcal{H}_k(\mathbf{v}_s) \odot \mathbf{v}_s\right] \oslash \left[\sum_{i=1}^{\mathcal{S}} \mathbf{v}_s \odot \mathbf{v}_s\right] = \frac{1}{\mathcal{S}} \sum_{s=1}^{\mathcal{S}} \mathcal{H}_k(\mathbf{v}_s) \odot \mathbf{v}_s \in \mathbb{R}^d, \tag{29}$$

where $\mathbf{v}_s \sim \texttt{Rademacher}(0.5)$ and $\mathcal{S}$ is the number of random vectors used in the approximation. Thus, the diagonal approximate Hessian matrix $\mathbf{B}_k \in \mathbb{R}^{d \times d}$ is given by

$$\mathbf{B}_k \triangleq \mathrm{Diag}(\mathbf{b}_k) = \frac{1}{\mathcal{S}} \sum_{s=1}^{\mathcal{S}} \mathrm{Diag}\left(\mathcal{H}_k(\mathbf{v}_s) \odot \mathbf{v}_s\right). \tag{30}$$

The approximate Hessian matrix (30) is used in the description of Algorithms 1 and 2, in Section 4. It is worth noting that in the code implementation of Algorithms 1 and 2 only the diagonal of $\mathbf{B}_k$ is computed, which reduces the memory cost from $d \times d$ to $d$.

**Assumption 3.** *There is a constant $C_B > 0$ such that the inexact Hessian $\mathbf{B}_k$ satisfies, for all $k \geq 0$,*

$$\left\| \mathbf{B}_k - \text{Diag}(\nabla^2 f(\mathbf{x}_k)) \right\|_2 \leq C_B \left\| \mathbf{s}_k \right\|_2 . \tag{31}$$

By replacing $\nabla f(\mathbf{x}_k)$ and $\nabla^2 f(\mathbf{x}_k)$ with $\mathbf{g}_k$ and $\mathbf{B}_k$ we get

$$\mathfrak{m}_M(\mathbf{s}) \stackrel{\Delta}{=} F(\mathbf{x}_k) + \mathbf{g}_k^T \mathbf{s} + \frac{1}{2} \mathbf{s}^T \mathbf{B}_k \mathbf{s} + \frac{M}{6} \left\| \mathbf{s} \right\|_2^3 . \tag{32}$$

Note that the conditions under which $\nabla f(\mathbf{x}_k)$ and $\nabla^2 f(\mathbf{x}_k)$ can be substituted with $\mathbf{g}_k$ and $\mathbf{B}_k$ are detailed in Lemmata 5 and 6, respectively.

Let $(\mathbf{s}_{k+1}, \nu_{k+1})$ be the output of Algorithm 2 for $\mathbf{B} = \mathbf{B}_k$, $\mathbf{g} = \mathbf{g}_k$, and so on. Algorithm 2 is called in line 3 of Algorithm 1. Then, recall that $(\mathbf{s}_{k+1}, \nu_{k+1})$ is a minimizer of (7) and according to Theorem 2 it is also a minimizer of problem (5) for $M = \nu_{k+1}$. The first- and second-order optimality conditions

$$\mathbf{s}_{k+1}^T \nabla_{\mathbf{s}} \mathfrak{m}_{M=\nu_{k+1}}(\mathbf{s}_{k+1}) = 0 \tag{33}$$

and

$$\mathbf{s}_{k+1}^T \left( \nabla_{\mathbf{s}}^2 \mathfrak{m}_{M=\nu_{k+1}}(\mathbf{s}_{k+1}) \right) \mathbf{s}_{k+1} \geq 0, \tag{34}$$

get us to Lemma 3. Lemma 3 is exploited to prove Lemma 4. Lemma 4 is used by Lemmata 5, 6, and Corollary 2.

**Lemma 3** (Approximate model minimizer)**.** *Let*

$$\mathbf{s}_{k+1} = \underset{\mathbf{s} \in \mathbb{R}^d}{\arg\min} \ \mathfrak{m}_M(\mathbf{s}). \tag{35}$$

*Then, the following statements hold*

$$\mathbf{g}_k + \mathbf{B}_k \mathbf{s}_{k+1} + \frac{M}{2} \left\| \mathbf{s}_{k+1} \right\|_2 \mathbf{s}_{k+1} = \mathbf{0}, \tag{36}$$

$$\mathbf{B}_k + \frac{M}{2} \left\| \mathbf{s}_{k+1} \right\|_2 \mathbf{I} \succeq \mathbf{0}, \tag{37}$$

*and*

$$\mathbf{g}_k^T \mathbf{s}_{k+1} + \frac{1}{2} \mathbf{s}_{k+1}^T \mathbf{B}_k \mathbf{s}_{k+1} + \frac{M}{6} \left\| \mathbf{s}_{k+1} \right\|_2^3 \leq -\frac{M}{12} \left\| \mathbf{s}_{k+1} \right\|_2^3 . \tag{38}$$

*Recall that $\mathbf{x}_{k+1} = \mathbf{x}_k + \mathbf{s}_{k+1}$ and from Theorem 2, $M = \nu_{k+1}$.*

*Proof.* The reader is referred to (Wang et al., 2019, Lemma 3). $\square$

**Lemma 4.** *Let $\{F(\mathbf{x}_k)\}$ be bounded from below by $F^{\text{low}}$. Also, let $\mathbf{s}_{k+1}$ satisfy the first two conditions in Lemma 3 and let $M$ be bounded from below by some $M^{\text{low}}$. Then*

$$\left\| \mathbf{s}_{k+1} \right\| \to 0, \ as \ k \to \infty. \tag{39}$$

*Proof.* First, note that by Assumption 1, $F(\mathbf{x})$ is also bounded from below by some $F^{\text{low}}$. Additionally, since $M$ is bounded from below and $M = \nu_{k+1}$, as indicated in Theorem 2, $\nu_{k+1}$ is also bounded from below. The lower bound of $M$ is further discussed in Lemma 14 in Appendix B.11.

Following similar lines to Cartis et al. (2011a, Lemma 5.1), we focus on the sub-sequence of successful iterations, as in (Cartis et al., 2011a; Conn et al., 2000). Thus, from the successful iteration in Algorithm 1, i.e., when $\rho_k \in [\eta_1, \eta_2)$, we have

$$F(\mathbf{x}_k) - F(\mathbf{x}_{k+1}) \geq \eta_1 (F(\mathbf{x}_k) - \mathfrak{m}_{M=\nu_{k+1}}(\mathbf{s}_{k+1}))$$

$$\stackrel{(32)}{\geq} \eta_1 \left( -\mathbf{g}_k^T \mathbf{s}_{k+1} - \frac{1}{2} \mathbf{s}_{k+1}^T \mathbf{B}_k \mathbf{s}_{k+1} - \frac{M^{\text{low}}}{6} \left\| \mathbf{s}_{k+1} \right\|_2^3 \right), \quad (40)$$

which by applying (38) yields

$$F(\mathbf{x}_k) - F(\mathbf{x}_{k+1}) \geq \eta_1 \frac{M^{\text{low}}}{12} \left\| \mathbf{s}_{k+1} \right\|_2^3. \quad (41)$$

Summing over all iterates from 0 to $k-1$ in (41) we obtain

$$F(\mathbf{x}_0) - F(\mathbf{x}_{k+1}) \geq \frac{\eta_1}{12} M^{\text{low}} \sum_{k=0}^{k-1} \left\| \mathbf{s}_k \right\|_2^3, \quad (42)$$

which taking into account that $\{F(\mathbf{x}_k)\}$ is bounded below yields

$$\frac{12}{\eta_1 M^{\text{low}}} \left( F(\mathbf{x}_0) - F^{\text{low}} \right) \geq \sum_{k=0}^{k-1} \left\| \mathbf{s}_k \right\|_2^3. \quad (43)$$

Thus, the series $\sum_{k=0}^{k-1} \left\| \mathbf{s}_k \right\|_2^3$ is convergent and (39) holds. The same conclusion is also derived in (Cartis et al., 2011a, Lemma 5.1). $\qquad \square$

**Lemma 5.** *Let the approximate gradient $\mathbf{g}_k$ be computed on a set of points $\mathcal{B}_k^g$, with cardinality $|\mathcal{B}_k^g|$. For $\epsilon \geq 4\sqrt{2} L_f \sqrt{\frac{\ln \frac{1}{\delta} + \frac{1}{4}}{|\mathcal{B}_k^g|}}$ we have with high probability $1 - \delta$ that*

$$\left\| \mathbf{g}_k - \nabla f(\mathbf{x}_k) \right\|_2 \leq \epsilon. \quad (44)$$

*In addition, if*

$$|\mathcal{B}_k^g| \geq 32 L_f^2 \frac{\ln \frac{1}{\delta} + \frac{1}{4}}{C_g^2 \left\| \mathbf{s}_k \right\|_2^4}, \quad (45)$$

*and Lemma 4 holds, $\mathbf{g}_k$ satisfies Assumption 2.*

*Proof.* The proof can be found in Appendix B.5. $\qquad \square$

**Lemma 6.** *Let the approximate diagonal Hessian matrix $\mathbf{B}_k$ be computed on a set of points $\mathcal{B}_k^H$, with cardinality $|\mathcal{B}_k^H|$. For $\epsilon \geq \sqrt{d} L_g \frac{\ln \frac{2d}{\delta}}{\mathcal{S} |\mathcal{B}_k^H|}$ we have with high probability $1 - \delta$ that*

$$\left\| \mathbf{B}_k - \text{Diag}(\nabla^2 f(\mathbf{x}_k)) \right\|_2 \leq \epsilon. \quad (46)$$

*In addition, if*

$$|\mathcal{B}_k^H| \geq \sqrt{d} L_g \frac{\ln \frac{2d}{\delta}}{\mathcal{S} \left\| \mathbf{s}_k \right\|_2 C_B} \quad (47)$$

*and Lemma 4 holds, $\mathbf{B}_k$ satisfies Assumption 3.*

*Proof.* The proof can be found in Appendix B.6. $\qquad \square$

**Corollary 2.** *If*

$$|\mathcal{B}_k| \geq \max \left\{ 32 L_f^2 \frac{\ln \frac{1}{\delta} + \frac{1}{4}}{C_g^2 \left\| \mathbf{s}_{k-1} \right\|_2^4}, \sqrt{d} L_g \frac{\ln \frac{2d}{\delta}}{\mathcal{S} \left\| \mathbf{s}_{k-1} \right\|_2 C_B} \right\}, \quad (48)$$

*then $\mathbf{g}_k$ and $\mathbf{B}_k$ satisfy Assumptions 2 and 3 with probability $1 - \delta$, for $\delta \in (0, 1]$.*

*Proof.* We combine the results of Lemma 5 and 6. Note that $\|\mathbf{s}_{k-1}\|$ is used instead of $\|\mathbf{s}_k\|$. Due to Lemma 4, $\|\mathbf{s}_k\|_2 \leq \|\mathbf{s}_{k-1}\|_2 \Leftrightarrow \|\mathbf{s}_k\|_2^{-1} \geq \|\mathbf{s}_{k-1}\|_2^{-1}$. This modification is useful for the practical application of the sampling schemes. However, this poses a challenge since $C_g$, $C_H$, $L_f$, and $L_g$ are not easily accessible. $\square$

**Remark 2.** *Lemma 4 and Corollary 2 imply that the sample size is eventually equal to the entire sample size $n$ as Algorithm 1 converges. Thus we have*

$$\mathbf{g}_k \to \nabla f(\mathbf{x}_k) \ \text{and} \ \mathbf{B}_k \to \text{Diag}(\nabla^2 f(\mathbf{x}_k)) \ \text{as} \ k \to \infty. \tag{49}$$

*This allows us to invoke the deterministic local convergence guarantees as $k \to \infty$ in Theorem 3. However, stochastic first- and second-order information from $\mathbf{g}_k$ and $\mathbf{B}_k$ is used.*

## 4 Algorithmic Solution

In Theorem 1, it was shown that the optimal $(\nu^*, r^*)$ solving $\max_{\nu \geq 0, \ r \in \mathcal{D}_\nu} \mathscr{L}_\xi(\nu, r)$ is used in (18) to compute the minimizer of (7). To solve (7) and compute $(\nu^*, r^*)$ Algorithm 1 and 2 are utilized, respectively. In particular, Lemma 7 is employed in Algorithm 2, which is essential for calculating the values of $\nu^*$ and $r^*$.

Let *VSI*, *SI*, and *UI* stand for the Very Successful, Successful, and Unsuccessful Iteration, respectively, in Algorithm 1 (Conn et al., 2000, Section 6.1). Denote $\lambda_d^+(\mathbf{B}_k)$ as the minimal non-negative diagonal shift that makes $\mathbf{B}_k$ sufficiently positive definite to allow a stable computation of the TR step. Details on the selection of $\lambda_d^+(\mathbf{B}_k)$ in Algorithm 2 can be found in Conn et al. (2000, Section 7.3.11) and (Gould et al., 1999).

**Lemma 7** (Proof in Appendix B.7). *The optimal values $\nu^*$ and $r^*$ achieving*

$$\max_{\nu \geq 0, \ r \in \mathcal{D}_\nu} \mathscr{L}_\xi(\nu, r) \tag{50}$$

*are given by $r^* = \sqrt[3]{\xi}$ and by solving*

$$\phi(\nu^*, r^*) = \frac{1}{\|\mathbf{s}(\nu^*, r^*)\|_2} - \frac{1}{\sqrt[3]{\xi}} = 0, \tag{51}$$

*w.r.t. $\nu^*$, respectively.*

Next, we clarify the role and physical interpretation of the ADACUBIC hyperparameters as they appear in Algorithms 1 and 2:

- $\eta_1$ **(acceptance threshold).** $\eta_1 \in (0, 1)$ is the minimum ratio between the actual loss reduction and the predicted reduction of the cubic model required to accept a step. If $\rho_k \geq \eta_1$, the step is considered successful and the parameters are updated. This parameter $\eta_1$ controls how cautiously the algorithm accepts update steps. Smaller values make acceptance easier, while larger values enforce stricter agreement between the cubic model $\mathfrak{m}_{\nu_{k+1}}(\mathbf{s}_{k+1})$ and the objective function $F(\mathbf{x}_k + \mathbf{s}_{k+1})$.

- $\eta_2$ **(very successful threshold).** $\eta_2 \geq \eta_1$ identifies very successful iterations. When $\rho_k \geq \eta_2$, the effective trust-region boundary is expanded, allowing larger steps in subsequent iterations. This mechanism accelerates convergence when the cubic model $\mathfrak{m}_{\nu_{k+1}}(\mathbf{s}_{k+1})$ is highly accurate.

- $\alpha_1$ **(expansion factor).** $\alpha_1 \geq 1$ controls the increase of the trust-region parameter $\xi_k$ after very successful iterations, thereby expanding the effective trust-region boundary.

- $\alpha_2$ **(shrinkage factor).** $\alpha_2 \in (0, 1)$ decreases the trust-region boundary after unsuccessful iterations ($\rho_k \leq \eta_1$). By shrinking the trust-region boundary, more conservative updates are obtained, thereby improving robustness in regions where the cubic model $\mathfrak{m}_{\nu_{k+1}}(\mathbf{s})$ is less accurate.

---

**Algorithm 1** ADACUBIC algorithm

---

1: Set $\xi_k \leftarrow 1$, $\kappa_{\text{easy}} \in (0,1)$, $0 < \alpha_2 < 1 \leq \alpha_1$, and $0 < \eta_1 \leq \eta_2 < 1$.
2: **repeat**                                                                                                                         ▷ $k$-th iteration, $k = 0, 1, \dots$
    ▷   The function $F$, $\mathbf{B}_k$, and $\mathbf{g}_k$ are evaluated on the same batch.
    ▷   ROOTFINDER is Algorithm 2.
3:     $\mathbf{s}_{k+1}, \nu_{k+1} \leftarrow$ ROOTFINDER$(\mathbf{B}_k, \mathbf{g}_k, \xi_k, \kappa_{\text{easy}})$
4:     Compute $\rho_k$ using
$$\rho_k = \frac{F(\mathbf{x}_k) - F(\mathbf{x}_k + \mathbf{s}_{k+1})}{F(\mathbf{x}_k) - \mathfrak{m}_{\nu_{k+1}}(\mathbf{s}_{k+1})}$$
5:     **if** $\rho_k \geq \eta_1$ **then**
6:         $\mathbf{x}_{k+1} \leftarrow \mathbf{x}_k + \mathbf{s}_{k+1}$
7:     **else**
8:         $\mathbf{x}_{k+1} \leftarrow \mathbf{x}_k$
9:     **end if**
10:    Update $\xi_k$ using
$$\xi_{k+1} \leftarrow \begin{cases} \max\left\{\alpha_1 \left\|\mathbf{s}_{k+1}\right\|_2^3, \, \xi_k\right\} & \text{if } \rho_k \geq \eta_2 \quad \triangleright \text{VSI} \\ \text{keep the same } \xi_k & \text{if } \rho_k \in [\eta_1, \eta_2) \quad \triangleright \text{SI} \\ \max\left\{\alpha_2 \left\|\mathbf{s}_{k+1}\right\|_2^3, \epsilon_m\right\} & \text{if } \rho_k \leq \eta_1 \qquad \triangleright \text{UI} \end{cases}$$
    where $\epsilon_m \approx 10^{-6}$.
11: **until** execution stops (e.g., after a specific number of training epochs)

---

**Algorithm 2** Find model minimizer

---

1: **procedure** ROOTFINDER$(\mathbf{B}, \mathbf{g}, \xi, \kappa_{\text{easy}})$
2:     Set $r \leftarrow \sqrt[3]{\xi}$
3:     **if** $\mathbf{B}$ is positive definite **then**
4:         $\nu \leftarrow 0$
5:     **else**
        ▷   For some $\lambda_d^+(\mathbf{B})$ barely smaller than $\lambda_d(\mathbf{B})$
6:         $\nu \leftarrow -2\, \lambda_d^+(\mathbf{B}) \,/\, r$
7:     **end if**
8:     Compute $\mathbf{s} = -(\mathbf{B} + \frac{1}{2}\nu\, r\, \mathbf{I})^{-1}\, \mathbf{g}$
9:     **if** $\left\|\mathbf{s}\right\|_2^3 \leq \xi$ **then**
10:        **if** $\mathbf{B}$ is positive definite or $\left\|\mathbf{s}\right\|_2^3 = \xi$ **then**
11:            **return** $\mathbf{s}, \nu$
12:        **else**
13:            Compute the eigenvector $\mathbf{u}_d$ that corresponds to the eigenvalue $\lambda_d(\mathbf{B})$. Then find the root $\alpha$
            of the equation $\left\|\mathbf{s} + \alpha\, \mathbf{u}_d\right\|_2 = \xi^{1/3}$ which makes the model $\mathfrak{m}_\nu(\mathbf{s} + \alpha\, \mathbf{u}_d)$ the smallest.
14:            **return** $\mathbf{s} + \alpha\, \mathbf{u}_d, \nu$
15:        **end if**
16:    **end if**
    ▷ The following, produces $\mathbf{s}^*$ and $\nu^*$ in Lemma 1.
17:    **while** $\left|\, \left\|\mathbf{s}\right\|_2 - \xi^{1/3}\right| \leq \kappa_{\text{easy}}\, \xi^{1/3}$ **do**
    ▷ By Remark 3, $\nu$ increases.
18:        $\nu \leftarrow \nu - \phi(\nu, r) \,/\, \partial_\nu \phi(\nu, r)$
19:        $\mathbf{s} = -(\mathbf{B} + \frac{1}{2}\nu\, r\, \mathbf{I})^{-1}\, \mathbf{g}$
20:    **end while**
21:    **return** $\mathbf{s}, \nu$
22: **end procedure**

---

- $\kappa_{\textbf{easy}}$ **(root-finding tolerance).** $\kappa_{\text{easy}} \in (0, 1)$ specifies the error tolerance to terminate the Newton iterations when solving the cubic subproblem in Algorithm 2. $\kappa_{\text{easy}}$ determines how close the norm of the computed step should be to the trust-region boundary before the termination of the dual variable calculation. Smaller values enforce higher accuracy in solving the subproblem, while larger values favor computational efficiency.

Overall, $\eta_1$ and $\eta_2$ govern step acceptance, $\alpha_1$ and $\alpha_2$ regulate updates of the trust-region boundary, and $\kappa_{\text{easy}}$ balances accuracy and efficiency in the inner solver of Algorithm 2. AdaCubic adaptively computes the dual parameter $\nu_{k+1}$, which determines the step $\mathbf{s}_{k+1}$, the acceptance ratio $\rho_k$, and consequently the evolution of the trust-region parameter $\xi_k$. The dual variable $\nu_{k+1}$ encodes local curvature information through the Hessian approximation and acts as an adaptive term in the cubic subproblem. This relationship enables an automatic adjustment of $\xi_k$, allowing AdaCubic to respond effectively to the local geometry of the non-convex loss landscape and to achieve competitive performance across the benchmarks in Section 5.

## 5 Experimental Evaluation

Experiments are conducted on computer vision, natural language processing, and signal processing tasks, where the results obtained with the proposed AdaCubic optimizer are compared with those obtained with the SGD, Adam, and AdaHessian optimizers. The natural language processing experiments are conducted using the Hugging Face Transformers library (Wolf et al., 2020). For SGD, Adam, and AdaHessian, the Learning Rate (LR) is fine-tuned. For AdaCubic, the parameters $\eta_1 = 0.05$, $\eta_2 = 0.75$, $\alpha_1 = 2.5$, $\alpha_2 = 0.25$, and $\kappa_{\text{easy}} = 0.01$ are chosen universally in the experimental evaluation. These parameters are chosen based on the analysis in (Conn et al., 2000, Section 17.1). Table 1 summarizes the universal hyperparameter values used by AdaCubic across all benchmarks.

Table 1: Universal AdaCubic hyperparameter settings. All hyperparameters in Algorithm 2 are fixed across benchmarks. $\epsilon_m$ denotes a numerical safeguard used in Algorithm 1.

| | AdaCubic Hyperparameters | | | | | |
|---|---|---|---|---|---|---|
| **Hyperparameter** | $\eta_1$ | $\eta_2$ | $\alpha_1$ | $\alpha_2$ | $\kappa_{\text{easy}}$ | $\epsilon_m$ |
| **Assigned Value** | 0.05 | 0.75 | 2.5 | 0.25 | 0.01 | $10^{-6}$ |

Tables 2 and 3 summarize the experimental configurations for each benchmark, including datasets, model architectures, optimizers, and LR settings.

Table 2: Summary of model architectures, training settings, and optimizers used in all experiments.

| Task | Dataset | Model | Batch | Epochs | Optimizers |
|---|---|---|---|---|---|
| CV | CIFAR-10 | ResNet20 / ResNet32 | 256 | 500 | SGD, Adam, AdaHessian, AdaCubic |
| | CIFAR-100 | ResNet18 | 256 | 200 | SGD, Adam, AdaHessian, AdaCubic |
| NLU | SST-2, QNLI, RTE, WNLI | SqueezeBERT | 32 | 15 | SGD, AdaHessian, AdaCubic |
| | MRPC, QQP | SqueezeBERT | 32 | 15 | SGD, AdaHessian, AdaCubic |
| | STS-B, MNLI | SqueezeBERT | 32 | 15 | SGD, AdaHessian, AdaCubic |
| LM | WikiText-2 | RoBERTa / BERT / DistilBERT | 8 | 6 | SGD, AdaHessian, AdaCubic |
| | PTB | RoBERTa / BERT / DistilBERT | 8 | 6 | SGD, AdaHessian, AdaCubic |
| CMI | VISION | ResNet18 | 256 | 100 | Adam, AdaCubic |

**Computer Vision (CV).** To prove the effectiveness of AdaCubic, experiments are conducted using `CIFAR-10` and `CIFAR-100` datasets (Krizhevsky, 2009). The experimental results are summarized in Table 4. In all experiments, a batch size of 256 is used. The mean accuracy and standard deviation (std) over five runs are reported for each experiment. The number of epochs used to train the models on `CIFAR-10` and `CIFAR-100` is 500 and 200, respectively. In addition, the optimizers are fine-tuned w.r.t. the initial LR and the decaying LR scheme. For SGD, Adam, and AdaHessian, the initial learning rates are 0.1, 0.001, and

Table 3: Summary of LRs used in all experiments. For the CV benchmark, LRs are decayed by a factor of 10 at epochs 80 and 120 on `CIFAR-10`, and by a factor of 20 at epochs 60, 120, and 160 on `CIFAR-100`. For NLU, LM, and CMI benchmarks, no LR decay is applied. For CMI, LR is decayed by a factor of 10 at epochs 80 and 120. AdaCubic is used with a fixed universal parameter set and does not require LR tuning.

| Optimizer | Task | Dataset(s) | Initial LR | LR Schedule / Tuning |
|---|---|---|---|---|
| SGD | CV | `CIFAR-10` / `CIFAR-100` | 0.1 | Step decay (tuned) |
| Adam | CV | `CIFAR-10` / `CIFAR-100` | $10^{-3}$ | Step decay (tuned) |
| AdaHessian | CV | `CIFAR-10` / `CIFAR-100` | 0.15 | Step decay (tuned) |
| AdaCubic | CV | `CIFAR-10` / `CIFAR-100` | no LR | Universal parameters |
| SGD | NLU | `SST-2, QNLI, RTE, WNLI` | $2 \times 10^{-2}$ | Tuned |
| SGD | NLU | `STS-B` | $2 \times 10^{-3}$ | Tuned |
| AdaHessian | NLU | `SST-2, QNLI, STS-B, MNLI` | $2 \times 10^{-3}$ | Tuned |
| AdaHessian | NLU | `MRPC, RTE` | $2 \times 10^{-4}$ | Tuned |
| AdaHessian | NLU | `WNLI` | $2 \times 10^{-2}$ | Tuned |
| AdaCubic | NLU | All `GLUE` tasks | no LR | Universal parameters |
| SGD | LM | `WikiText-2, PTB` | $5 \times 10^{-3}$ | Tuned |
| AdaHessian | LM | `WikiText-2` (all models) | $5 \times 10^{-4}$ | Tuned |
| AdaHessian | LM | `PTB` (RoBERTa) | $5 \times 10^{-3}$ | Tuned |
| AdaHessian | LM | `PTB` (BERT, DistilBERT) | $5 \times 10^{-4}$ | Tuned |
| AdaCubic | LM | `WikiText-2, PTB` | no LR | Universal parameters |
| Adam | CMI | `VISION` | $10^{-4}$ | Tuned |
| AdaCubic | CMI | `VISION` | no LR | Universal parameters |

0.15. Furthermore, for AdaHessian, $\beta_1$ and $\beta_2$ are set to 0.9 and 0.999, respectively. On `CIFAR-10`, the LR is decayed by a factor of 10 at epochs 80 and 120, while on `CIFAR-100`, the LR is decayed by a factor of 20 at epochs 60, 120, and 160. In addition, spatial averaging (Yao et al., 2021) is used for AdaCubic and AdaHessian on `CIFAR-100`. The entries corresponding to the best accuracy are marked in bold. $\Delta$ reports the accuracy differences between AdaCubic and the strongest competing optimizer in each setting. When spatial averaging is used, the accuracy is shown in gray.

On the `CIFAR-10` dataset, both AdaHessian and AdaCubic demonstrate higher accuracy than conventional optimization methods like SGD and Adam. It is worth noting that, while both methods excel, AdaHessian achieves a slight edge in accuracy over AdaCubic for `ResNet20` and `ResNet32` by 0.15% and 0.5%, respectively. This performance distinction underscores the effectiveness of AdaCubic and positions it as a formidable competitor to AdaHessian in enhancing model accuracy on the `CIFAR-10` dataset.

On the `CIFAR-100` dataset without spatial averaging, AdaCubic falls behind SGD, Adam, and AdaHessian by margins of 0.81%, 0.23%, and 0.64%, respectively. However, with spatial averaging, both AdaHessian and AdaCubic achieve improved accuracy. This comparative analysis highlights AdaCubic's distinct performance characteristics, demonstrating its unique capabilities relative to other optimizers in challenging scenarios, such as on the `CIFAR-100` dataset.

Figure 1 depicts the training loss of `ResNet20` (top) and `ResNet32` (bottom) on `CIFAR-10` for Adam, AdaHessian, and AdaCubic optimizers. As can be seen, the losses of Adam and AdaHessian decrease dramatically at epoch 80, when the LR has decayed by a factor of 10. As can be seen, only by using an adaptive LR can the training loss reduction of Adam and AdaHessian match that of AdaCubic. In the last epochs, the loss of AdaCubic is lower than that of Adam and higher than that of AdaHessian. It should be noted that, in all experiments, AdaCubic is used with the same set of parameters and achieves competitive performance compared to the remaining fine-tuned optimizers.

On `CIFAR-10`, AdaCubic consistently outperforms first-order methods (SGD, Adam) and ranks second to AdaHessian, with very small gaps of 0.15% and 0.5% for `ResNet20` and `ResNet32`, respectively, as

Table 4: Accuracy (%) and std of the accuracy measures for `ResNet18/20/32` models on `CIFAR-10` and `CIFAR-100` datasets. $\Delta$ reports the gap between the strongest competing optimizer and ADACUBIC.

| | CIFAR-10 | | CIFAR-100 |
|---|---|---|---|
| | ResNet20 | ResNet32 | ResNet18 |
| SGD | $88.52 \pm 0.24$ | $89.02 \pm 0.20$ | $\mathbf{72.62} \pm 0.002$ |
| ADAM | $90.26 \pm 0.19$ | $91.24 \pm 0.20$ | $72.04 \pm 0.13$ |
| ADAHESSIAN | $\mathbf{91.64} \pm 0.46$ | $\mathbf{93.15} \pm 0.12$ | $72.45 \pm 0.16$ |
| | - | - | $72.59 \pm 0.271$ |
| ADACUBIC | $91.49 \pm 0.46$ | $92.65 \pm 0.19$ | $71.81 \pm 0.003$ |
| | - | - | $72 \pm 0.337$ |
| $\Delta$ | $0.15 \pm 0.36$ | $0.5 \pm 0.07$ | $0.81 \pm 0.001$ |
| | - | - | $0.59 \pm 0.066$ |

Table 5: Figures of merit on `GLUE` benchmark using SGD, ADAHESSIAN, and ADACUBIC optimizers on natural language understanding tasks. $\Delta$ reports the gap between the strongest competing optimizer and ADACUBIC

| Dataset | SGD | ADAHESSIAN | ADACUBIC | $\Delta$ |
|---|---|---|---|---|
| | **Accuracy (%)** | | | |
| SST-2 | **91.62** | 90.71 | 90.71 | 0.91 |
| QNLI | **90.37** | 89.47 | 90.01 | 0.36 |
| RTE | **70.39** | 64.98 | **70.39** | 0.00 |
| WNLI | **56.33** | **56.33** | **56.33** | 0.00 |
| | $F_1$ / **Accuracy (%)** | | | |
| MRPC | **0.9094 / 87.25** | 0.8562 / 78.18 | 0.9042 / 86.76 | 0.0052/0.49 |
| QQP | **0.8775 / 90.89** | 0.8742 / 90.82 | 0.8723 / 90.40 | 0.0052/0.49 |
| | **Pearson / Spearman Corr.** | | | |
| STS-B | **0.8863 / 0.8845** | 0.8786 / 0.8735 | 0.8832 / 0.8814 | 0.0031/0.0031 |
| | **Matched / Mismatched Accuracy (%)** | | | |
| MNLI | **82.45 / 82.05** | 81.65 / 81.57 | 81.88 / 81.89 | 0.57/0.16 |

summarized in Table 4. On `CIFAR-100` without spatial averaging, ADACUBIC trails the best-performing optimizer by at most 0.81%. Due to its larger number of classes and increased classification difficulty, CIFAR-100 will possibly lead to optimization regimes with stronger parameter interactions. Since ADACUBIC, like ADAHESSIAN, relies on a diagonal approximation of the Hessian, it does not explicitly capture such off-diagonal curvature effects, which may partially explain the observed gap. Importantly, when spatial averaging is applied, the performance of ADACUBIC improves and becomes closer to that of ADAHESSIAN and SGD, confirming that part of the gap is related to high-variance curvature estimation.

**Natural Language Understanding (NLU).** Table 5 summarizes the results on the natural language understanding task. The `GLUE` benchmark (Wang et al., 2018) is used to train the `SqueezeBERT` (Iandola et al., 2020) model for 15 epochs. For SGD and ADAHESSIAN, the initial LR is fine-tuned in all datasets. For SGD, the initial LR is set to $2 \cdot 10^{-2}$ for all datasets except from `STS-B` where it is set to $2 \cdot 10^{-3}$. For ADAHESSIAN, the initial LR is set to $2 \cdot 10^{-3}$ for `SST-2`, `STS-B`, `MNLI`, and `QNLI`, to $2 \cdot 10^{-4}$ for `MRPC` and `RTE`, and to $2 \cdot 10^{-2}$ for `WNLI`.

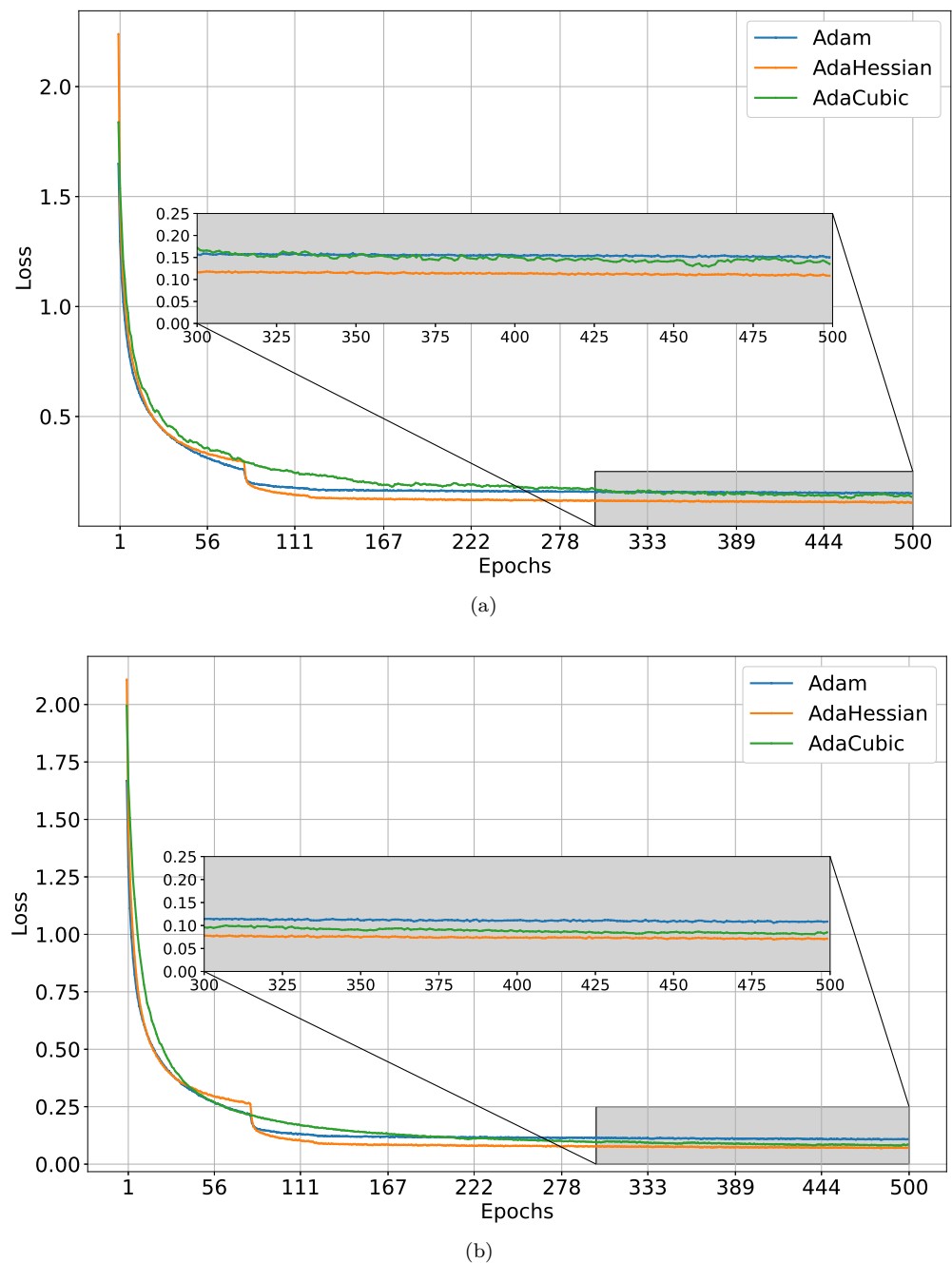

Figure 1: Training loss curve of `ResNet20` (top) and `ResNet32` (bottom) on `CIFAR-10` for ADAM, ADAHES­SIAN, and ADACUBIC optimizers.

The default parameters of the `SqueezeBERT` model can be found in the official Hugging Face library[1]. The dataset acronyms in the Hugging Face library are `SST-2`, `QNLI`, `RTE`, `WNLI`, `MRPC`, `QQP`, `STS-B`, and `MNLI`, while the model acronym is `squeezebert/squeezebert-uncased`.

To simplify the experimental evaluation, the experiments are divided into four groups, each corresponding to a different performance measure. Group 1 consists of the `SST-2`, `QNLI`, `RTE`, and `WNLI` datasets. Group

---

[1]`https://github.com/huggingface/transformers/tree/main/examples/pytorch/text-classification`

2 consists of the `MRPC` and `QQP` datasets, while groups 3 and 4 consist of the `SST-B` and `MNLI` datasets, respectively. The entries corresponding to the best metrics are marked in bold. $\Delta$ reports the accuracy differences between ADACUBIC and the strongest competing optimizer in each setting.

- *Group 1.* Concerning accuracy measure, ADACUBIC and ADAHESSIAN demonstrate the same performance on `SST-2`, while SGD performs better by 0.91%. On `QNLI`, SGD outperforms ADACUBIC by 0.36%, while ADACUBIC outperforms ADAHESSIAN by 0.54%. On `RTE`, ADACUBIC and SGD achieve the same performance, while ADAHESSIAN is outperformed by 5.41%. On `WNLI`, all optimizers achieve the same performance. Overall, the mean accuracies achieved by SGD, ADAHESSIAN, and ADACUBIC are 77.17%, 75.37%, and 76.86%, respectively. It can be observed that, on average, SGD outperforms ADACUBIC by 0.31%, while ADACUBIC outperforms ADAHESSIAN by 1.5%.

- *Group 2.* Concerning $F_1$ measure on `MRPC`, SGD outperforms ADACUBIC by 0.0052, while ADACUBIC outperforms ADAHESSIAN by 0.048. On the same dataset, SGD achieves higher accuracy than ADACUBIC by 0.49%, whereas ADACUBIC outperforms ADAHESSIAN by 8.58%. Concerning $F_1$ measure on `QQP`, SGD outperforms ADAHESSIAN by 0.0052, while ADAHESSIAN outperforms ADACCUBIC by 0.0019. On the same dataset, SGD achieves higher accuracy than ADAHESSIAN by 0.07%, while ADAHESSIAN outperforms ADACUBIC by 0.42%. Overall, the mean $F_1$ values achieved by SGD, ADAHESSIAN, and ADACUBIC are 0.89345, 0.8652, and 0.88825, respectively, while the mean accuracies are 89.07%, 84.5%, and 88.58%, respectively. This way, on average, SGD outperforms ADACUBIC by 0.0052 and 0.49%, on $F_1$ and accuracy measures, respectively, while ADACUBIC outperforms ADAHESSIAN by 0.02305 and 4.08%, respectively.

- *Group 3.* Concerning Pearson correlation index, SGD outperforms ADACUBIC by 0.0031, while ADACUBIC outperforms ADAHESSIAN by 0.0046. Regarding the Spearman correlation index, SGD outperforms ADACUBIC by 0.0031, while ADACUBIC outperforms ADAHESSIAN by 0.0079.

- *Group 4.* Concerning matched accuracy (Wang et al., 2018), SGD outperforms ADACUBIC by 0.57%, while ADACUBIC outperforms ADAHESSIAN by 0.23%. Concerning mismatched accuracy (Wang et al., 2018), SGD outperforms ADACUBIC by 0.16%, while ADACUBIC outperforms ADAHESSIAN by 0.32%.

It is worth noting that ADACUBIC exhibits the second-best performance with a pre-fixed universal set of parameters, while SGD and ADAHESSIAN are fine-tuned w.r.t. the initial LR.

**Language Modeling (LM).** Tables 6 and 7 summarize the results on the language modeling, where perplexity (Jelinek et al., 1977) is used as an evaluation metric. `PTB` (Marcus et al., 1994) and `wikitext-2` (Merity et al., 2017) datasets are used to train `RoBERTa` (Liu et al., 2019), `BERT` (Devlin et al., 2018), and `DistilBERT` (Sanh et al., 2019) models with SGD, ADAHESSIAN, and ADACUBIC optimizers.

Table 6: Perplexity achieved by SGD, ADAHESSIAN, and ADACUBIC on `wikitext-2` dataset.

| Optimizer | RoBERTa | BERT | DistilBERT |
|---|---|---|---|
| SGD | **3.547** | 13.380 | **6.118** |
| ADACUBIC | 3.756 | **5.759** | 6.565 |
| ADAHESSIAN | 4.374 | 16.151 | 6.822 |

Table 7: Perplexity using SGD, ADAHESSIAN, and ADACUBIC on the `PTB` dataset.

| Optimizer | RoBERTa | BERT | DistilBERT |
|---|---|---|---|
| SGD | **4.345** | 17.344 | 8.299 |
| ADACUBIC | 5.145 | **14.170** | **7.334** |
| ADAHESSIAN | 7.582 | 20.851 | 10.182 |

The initial LR of SGD is fine-tuned to $5 \cdot 10^{-3}$ for all models and both datasets. For ADAHESSIAN, the initial LR is fine-tuned to $5 \cdot 10^{-4}$ for all models on `wikitext-2` dataset. On `PTB` dataset, the initial LR of ADAHESSIAN optimizer is set to $5 \cdot 10^{-3}$ to train `RoBERTa` model, while the remaining models are trained with initial LR $5 \cdot 10^{-4}$. The remaining parameters for the trained models can be found in the official Hugging Face library[2]. The dataset acronyms in the Hugging Face library are `ptb_text_only` and `wikitext-2-raw-v1`. In contrast, the model acronyms are `roberta-base`, `bert-base-cased`, and `distilbert-base-uncased`.

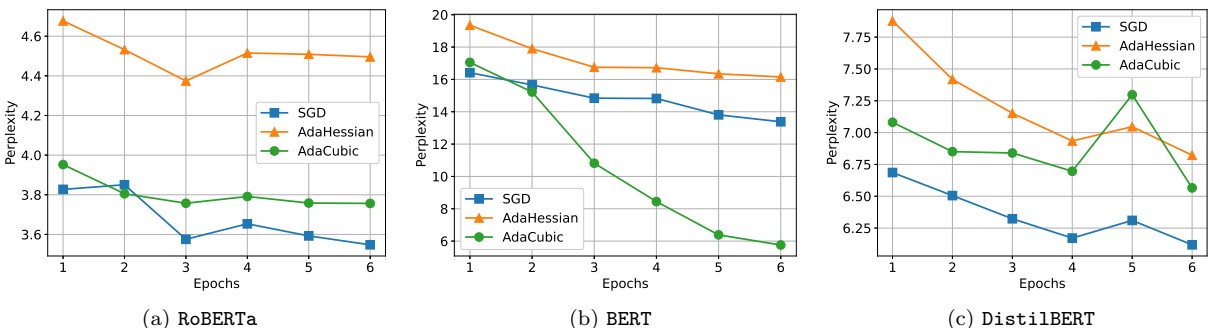

(a) `RoBERTa`  (b) `BERT`  (c) `DistilBERT`

Figure 2: Perplexity vs. epochs for `RoBERTa`, `BERT`, and `DistilBERT` models on `wikitext-2` dataset.

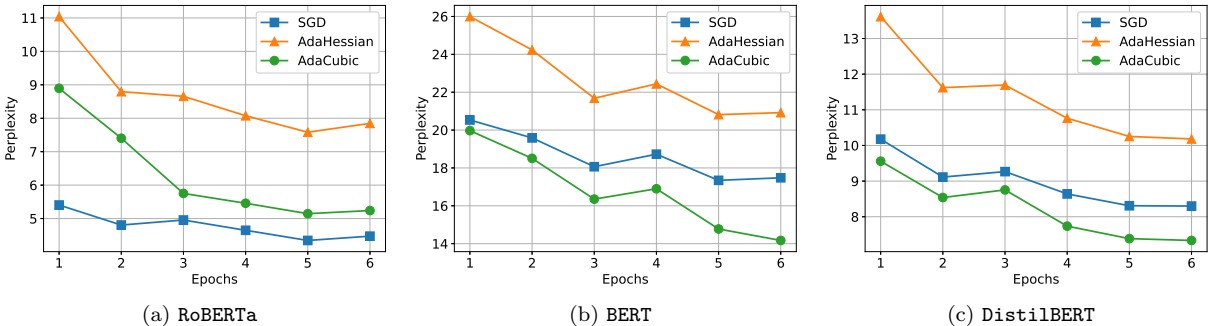

(a) `RoBERTa`  (b) `BERT`  (c) `DistilBERT`

Figure 3: Perplexity vs. epochs for `RoBERTa`, `BERT`, and `DistilBERT` models on `PTB` dataset.

First, the perplexity measurements gathered for the `wikitext-2` dataset in Table 6 are discussed. When `RoBERTa` is used, SGD outperforms ADACUBIC and ADAHESSIAN by 0.209 and 0.827, respectively. Next, when `BERT` is used, ADACUBIC outperforms SGD and ADAHESSIAN by 7.621 and 10.392, respectively. For the `DistilBERT` model, SGD outperforms ADACUBIC and ADAHESSIAN by 0.447 and 0.704, respectively. We observe that in all models, ADACUBIC outperforms ADAHESSIAN and performs better or competitively when compared to SGD. Figure 2 depicts the perplexity metric vs. epochs for all models and optimizers on the `wikitext-2` dataset.

Table 7 gathers perplexity measures on the `PTB` dataset. When `RoBERTa` is used, SGD outperforms ADACUBIC and ADAHESSIAN by 0.8 and 3.237, respectively. For the `BERT` model, ADACUBIC outperforms SGD and ADAHESSIAN by 3.174 and 6.681, respectively. For the `DistilBERT` model, ADACUBIC outperforms SGD and ADAHESSIAN by 0.965 and 2.848, respectively. Figure 3 depicts the perplexity metric vs. epochs for all models and optimizers on the `PTB` dataset.

On the NLU benchmark, Table 5, ADACUBIC consistently achieves either the best or the second-best performance across all tasks, with the performance gaps reported in the $\Delta$ column remaining small. The second-best performance of ADACUBIC on certain GLUE tasks can be understood in light of recent Hessian-based analyses of Transformers (Zhang et al., 2024). In particular, Zhang et al. (2024) shows that Transformer models

---

[2]`https://github.com/huggingface/transformers/tree/main/examples/pytorch/language-modeling`

exhibit block-wise heterogeneity in their Hessian structure, with strong curvature differences and interactions across parameter groups. While ADACUBIC explicitly leverages second-order information through diagonal Hessian approximations, such approximations may be insufficient to capture cross-parameter or block-level curvature interactions fully. This likely explains why ADACUBIC remains highly competitive but does not consistently outperform finely tuned baselines on Transformer-based tasks. Similar conclusions hold for the LM benchmark, where ADACUBIC consistently achieves either the best or second-best performance across all datasets.

Overall, it should be noted that ADACUBIC exhibits the best or second-best performance with a pre-fixed universal set of parameters, while SGD and ADAHESSIAN are fine-tuned w.r.t. the initial LR.

**Camera Model Identification (CMI).** The publicly available `VISION` dataset (Shullani et al., 2017) is utilized for camera model identification. `VISION` includes 648 `Native` videos, which remain unaltered post-capture by the camera. These `Native` videos were disseminated via social media platforms such as `YouTube` and `WhatsApp`, with corresponding versions included in the dataset. Of the 684 `Native` videos, 644 were shared via `YouTube` and 622 via `WhatsApp`. Additional details on `VISION` can be found in (Shullani et al., 2017). Taking into account the `VISION` dataset naming conventions outlined in Shullani et al. (2017), videos captured by devices D04, D12, D17, and D22 are excluded due to issues encountered during frame extraction or audio track retrieval.

Table 8: CMI accuracy (%) using `ResNet18`.

|  | ADACUBIC | | | ADAM | | |
|---|---|---|---|---|---|---|
|  | Native | WhatsApp | YouTube | Native | WhatsApp | YouTube |
| Fold 0 | 97.40 | 96.10 | 94.59 | 96.10 | 93.50 | 91.9 |
| Fold 1 | 93.51 | 93.51 | 93.24 | 94.80 | 90.90 | 93.24 |
| Fold 2 | 94.81 | 92.22 | 94.59 | 90.90 | 88.31 | 95.94 |
| Fold 3 | 93.42 | 93.43 | 91.89 | 93.42 | 94.73 | 82.43 |
| Fold 4 | 94.73 | 88.16 | 93.24 | 94.73 | 88.15 | 95.94 |
| Mean | 94.77 | 93.68 | 93.51 | 93.99 | 91.11 | 91.89 |
| ± std | ± 1.43 | ± 2.59 | ± 1.01 | ± 1.76 | ± 2.66 | ± 4.98 |

The videos are partitioned into training, testing, and validation sets to conduct a typical five-fold stratified cross-validation. The audio content from each video is extracted, and the log-Mel spectrogram of each extracted audio is computed using three distinct windows and hop sizes. This results in 3-channel log-Mel spectrograms that capture various frequency details of the audio content. The 3-channel log-Mel spectrograms are then fed into `ResNet18` to perform CMI. Furthermore, for ADAM, $\beta_1$ and $\beta_2$ are set to 0.9 and 0.999, respectively. The LR is decayed by a factor of 10 at epochs 80 and 120 with an initial value $10^{-4}$.

Table 8 summarizes the results when ADACUBIC and ADAM optimizers are used. The mean accuracy achieved using ADACUBIC in the `Native`, `WhatsApp`, and `YouTube` benchmarks is 94.77%, 93.68%, and 93.51%, respectively. In comparison, the mean accuracy with ADAM is 93.99% for `Native`, 91.11% for `WhatsApp`, and 91.89% for `YouTube`. This indicates that ADACUBIC is more accurate than ADAM by 0.78%, 2.57%, and 1.62% in the `Native`, `WhatsApp`, and `YouTube` benchmarks, respectively. In terms of std, ADACUBIC demonstrates greater consistency than ADAM by achieving lower std values of 0.33, 0.07, and 3.97 in the `Native`, `WhatsApp`, and `YouTube` benchmarks, respectively. Implementation details for the audio CMI task can be found in (Tsingalis et al., 2024).

## 6 Computational Complexity and Discussion

The performance and time complexity of the second-order methods depend on the approximation of the second-order information captured by the Hessian matrix. Similarly to ADAHESSIAN, ADACUBIC leverages the Hutchinson method (Bekas et al., 2007) to approximate the diagonal of the Hessian matrix. Figure 4 depicts the time complexity of SGD, ADAHESSIAN, and ADACUBIC when they are used to train `ResNet20`

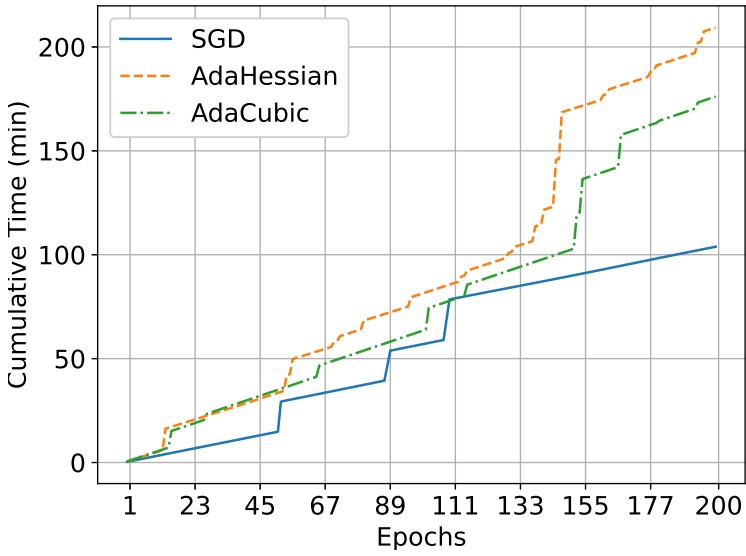

Figure 4: Cumulative time vs. epochs for SGD, ADAHESSIAN, and ADACUBIC for `ResNet20` on `CIFAR-10`.

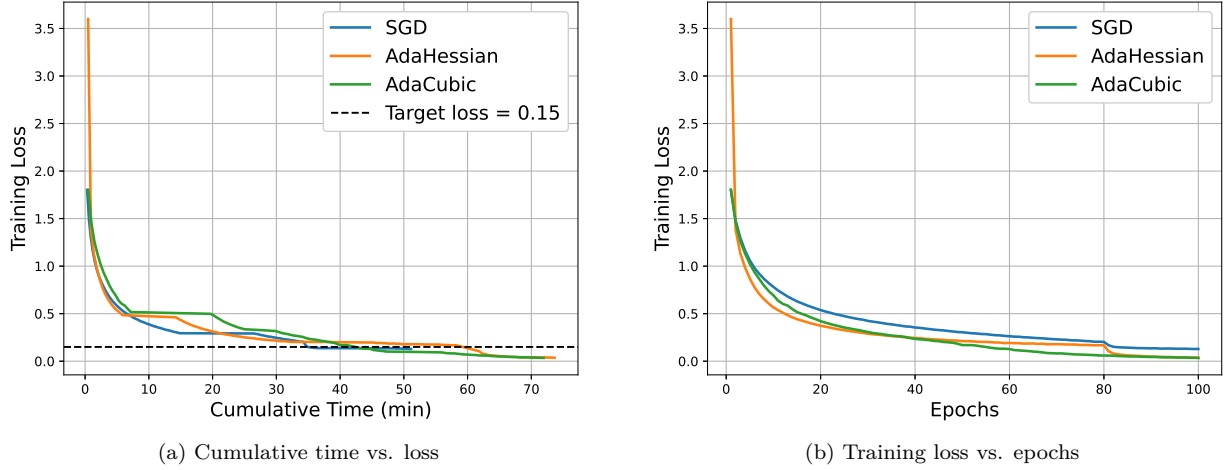

(a) Cumulative time vs. loss

(b) Training loss vs. epochs

Figure 5: Comparison of SGD, ADAHESSIAN, and ADACUBIC on `ResNet20` and `CIFAR-10`. Training loss vs. cumulative time over epochs (Left). Training loss vs. epochs (Right).

on `CIFAR-10`. As can be seen, the time complexity of the first-order optimizer SGD is smaller than that of the two second-order optimizers, with ADACUBIC having less time complexity than ADAHESSIAN.

Figure 5a shows the training loss vs. cumulative time for SGD, ADAHESSIAN, and ADACUBIC. Figure 5b shows the training loss vs. epochs for SGD, ADAHESSIAN, and ADACUBIC. The training loss in Figure 5b corresponds to that in Figure 5a. The horizontal dashed line in Figure 5a marks the target loss threshold of 0.15. ADACUBIC reaches this threshold after 55 epochs and 42.40 minutes. In comparison, SGD and ADAHESSIAN require 83 and 81 epochs, corresponding to 35.16 and 61.85 minutes. Table 9 summarizes the latter results. Although ADACUBIC needs more time than SGD due to computation of the second-order information, ADACUBIC reaches the desired loss in fewer epochs without any LR tuning. This highlights ADACUBIC as an efficient trade-off between computational cost and convergence quality.

Table 9: Execution time in minutes required to reach a target loss threshold when ResNet20 is trained on `CIFAR-10`.

|  | SGD | | ADAHESSIAN | | ADACUBIC | |
| --- | --- | --- | --- | --- | --- | --- |
|  | Epoch | Time | Epoch | Time | Epoch | Time |
| Result | 83 | 35.16 | 81 | 61.85 | 55 | 42.40 |

Additionally, storing the Hessian matrix increases the memory consumption of any second-order optimizer. Using Hutchinson's method for approximating the diagonal of the Hessian, the second-order information is represented by the diagonal approximation of the Hessian matrix, which leads to a $\mathcal{O}(d)$ memory complexity (Bekas et al., 2007). This additional memory cost is incurred by ADACUBIC relative to first-order methods such as SGD.

Furthermore, when utilizing Bekas et al. (2007), the approximation of the diagonal of the Hessian demands an additional gradient back-propagation. The additional gradient back-propagation step is also needed in ADAHESSIAN. When comparing ADACUBIC with ADAM, the latter shares similar memory consumption due to the requirement of the gradient momentum term, but it does not necessitate an additional gradient back-propagation.

Table 10: Comparison of optimization methods used in the experimental evaluation. Recall that $d$ denotes the number of model parameters and $\mathcal{S}$ the number of random vectors used in the diagonal Hessian approximation.

| Optimizer | Order | Sensitivity | Extra Backward Pass | Time Cost | Memory Footprint |
| --- | --- | --- | --- | --- | --- |
| SGD | First | High | No | $d$ | $d$ |
| ADAM | First | High | No | $d$ | $3d$ |
| ADAHESSIAN | Second | Medium | Yes | $\mathcal{S}\,d$ | $4d$ |
| ADACUBIC | Second | Low | Yes | $\mathcal{S}\,d$ | $2d$ |

Table 10 summarizes the optimization methods used in the experimental evaluation, highlighting their optimization order, sensitivity to hyperparameters, and computational overhead. The "Order" column indicates whether an optimizer relies on first- or second-order information. The sensitivity of the optimizers w.r.t. the LR is summarized in the "Sensitivity" column. The sensitivity of SGD, ADAM, and ADAHESSIAN w.r.t. the LR is discussed thoroughly in (Yao et al., 2021). ADACUBIC has low sensitivity, as it achieves competitive performance with a universal set of hyperparameters. The "Extra Backward Pass" column indicates whether additional back-propagation steps are required per optimization iteration, which directly relates to the use of second-order information. The reported time cost is dominated by the back-propagation procedure and is expressed as a function of the number of model parameters $d$. Recall that $\mathcal{S}$ is the number of random vectors used in the approximation of the diagonal Hessian matrix. Using $\mathcal{S}$ random vectors requires $\mathcal{S}$ backpropagation steps, increasing the time cost linearly. The "Memory Footprint" refers to the memory needed to store the gradient, the moment terms, and the approximated diagonal Hessian. As can be seen, the memory footprint of ADAM and ADAHESSIAN is $3d$ and $4d$, respectively, as the gradient and moments need memory relative to the number of parameters $d$. ADAHESSIAN needs an additional memory footprint of $d$ for the storage of the approximate diagonal Hessian. ADACUBIC shows a $2d$ memory overhead relative to SGD, since the gradient must be retained to compute the Hutchinson-based approximation of the diagonal Hessian, which is subsequently used by Algorithm 2.

However, according to Algorithm 2, ADACUBIC requires only the approximated diagonal Hessian for its updates, yielding a theoretical memory footprint of $\mathcal{O}(d)$. The gap between practical and theoretical memory costs comes from the design of modern deep-learning frameworks, such as PyTorch, which are optimized for first-order optimization methods. Thus, computing the diagonal Hessian approximation requires retaining intermediate gradient information. Developing a custom implementation that directly computes the diagonal Hessian without storing such intermediates is beyond the scope of this work.

## 7 Conclusions

ADACUBIC, a novel adaptive cubic regularized second-order optimizer, has been proposed. ADACUBIC leverages an approximate Hessian diagonal to reduce the computational cost induced by estimating curvature information. Although many cubically regularized methods have been proposed in the literature, none have been extensively tested in practical deep-learning applications. The effectiveness of the proposed optimizer has been demonstrated through experiments on computer vision, natural language processing, and signal processing tasks that utilize deep neural networks trained on various datasets. With a pre-fixed universal selection of parameters, ADACUBIC exhibits better or competitive performance when compared to other state-of-the-art fine-tuned optimizers. This fact makes ADACUBIC an attractive solution for optimizing deep neural networks.

## Acknowledgments

This work was supported by the Hellenic Foundation for Research and Innovation (HFRI) under the HFRI PhD Fellowship grant (Fellowship Number: 1376) and the "2nd Call for HFRI Research Projects to support Faculty Members & Researchers" (Project Number: 3888). The results were obtained using the High-Performance Computing Infrastructure and Resources of Aristotle University of Thessaloniki (AUTh). The authors would like to acknowledge the support provided by the IT Center of AUTh throughout the progress of this research work.

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

# A    Summary of Dependencies

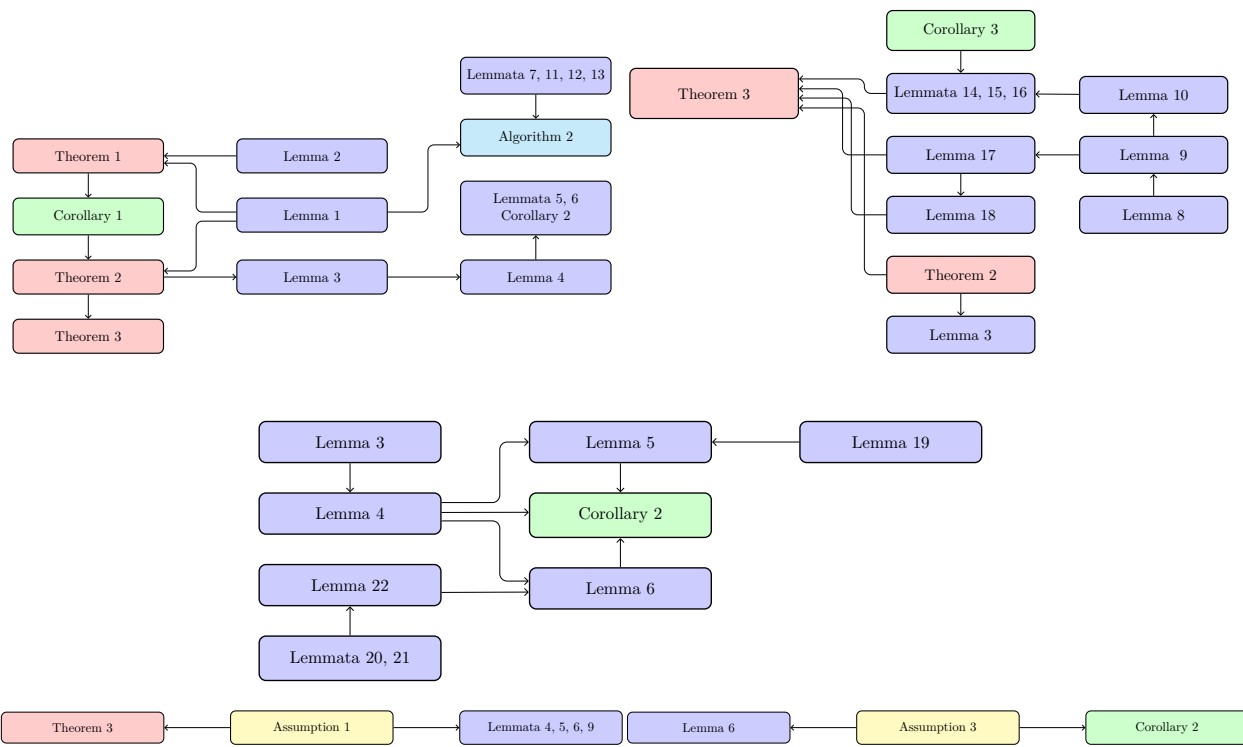

Figure 6: Logical connection between key lemmata, theorems, and corollaries throughout Sections 2 to 4 and Appendices B.1 to B.13.

# B    Supporting Proofs

## B.1    Proof of Lemma 1

Here, we follow the guidelines in (Conn et al., 2000, Theorem 7.2.1). Let us assume that $\mathbf{s}^*$ is a minimizer of $\hat{m}(\mathbf{s})$ subject to $\|\mathbf{s}^*\|_2^3 \leq \xi$. Then, there is a Lagrange multiplier $\nu^*$ such that

$$\nu^* \left( \|\mathbf{s}^*\|_2^3 - \xi \right) = 0 \Leftrightarrow \begin{cases} \nu^* = 0, & \text{inactive constraint} \\ \|\mathbf{s}^*\|_2^3 = \xi, & \text{active constraint.} \end{cases} \tag{52}$$

(52) is the unfolded *Complementary Slackness (CS)* condition Bertsekas (2017) for the constrained optimization problem (7). The active case occurs when $\mathbf{s}^*$ lies on the boundary of $\Omega$, i.e., $g_\xi(\mathbf{s}^*) = 0 \Leftrightarrow \|\mathbf{s}^*\|_2^3 = \xi$ and the inactive case occurs when $\mathbf{s}^*$ lies in the interior of $\Omega$, i.e. $g_\xi(\mathbf{s}^*) < 0 \Leftrightarrow \|\mathbf{s}^*\|_2^3 < \xi$.

▷ **Active constraint case.** We assume that $\mathbf{s}^*$ is a minimizer of $\hat{m}(\mathbf{s})$ subject to $\|\mathbf{s}^*\|_2^3 = \xi$. From the first-order optimality conditions Bertsekas (2017), there exists a Lagrange multiplier $\nu^*$, such that

$$\nabla_{\mathbf{s}} \mathcal{L}_\xi(\mathbf{s}^*, \nu^*) = \mathbf{0} \Leftrightarrow \nabla_{\mathbf{s}} \hat{m}(\mathbf{s}^*) + \nu^* \nabla_{\mathbf{s}} g_\xi(\mathbf{s}^*) = \mathbf{0}$$

$$\Leftrightarrow \underbrace{\nabla^2 f(\mathbf{x}_k)\, \mathbf{s}^* + \nabla f(\mathbf{x}_k)}_{\nabla_{\mathbf{s}} \hat{m}(\mathbf{s}^*)} + \frac{\nu^*}{2} \|\mathbf{s}^*\|_2\, \mathbf{s}^* = \mathbf{0} \Leftrightarrow$$

$$\left( \nabla^2 f(\mathbf{x}_k) + \frac{\nu^*}{2} \|\mathbf{s}^*\|_2\, \mathbf{I} \right) \mathbf{s}^* = -\nabla f(\mathbf{x}_k), \quad (53)$$

where the identity $\nabla_{\mathbf{s}} \|\mathbf{s}\|_2^3 = 3 \|\mathbf{s}\|_2 \mathbf{s}$ was used for some $\mathbf{s}$. Let $\mathbf{s}$ be a feasible point on the boundary of $\Omega$, i.e., $\|\mathbf{s}\|_2^3 = \xi$. The Taylor expansion of $\hat{m}(\mathbf{s})$ around the minimizer $\mathbf{s}^*$ is

$$\hat{m}(\mathbf{s}) = \hat{m}(\mathbf{s}^*) + (\mathbf{s} - \mathbf{s}^*)^T \nabla_{\mathbf{s}} \hat{m}(\mathbf{s}^*) + \frac{1}{2} (\mathbf{s} - \mathbf{s}^*)^T \nabla_{\mathbf{s}}^2 \hat{m}(\mathbf{s}^*) (\mathbf{s} - \mathbf{s}^*). \tag{54}$$

From the second line in (53), we also have

$$\nabla_{\mathbf{s}} \hat{m}(\mathbf{s}^*) = -\frac{\nu}{2} \|\mathbf{s}^*\|_2 \mathbf{s}^*. \tag{55}$$

Given (55) and the fact that $\mathbf{s}$ and $\mathbf{s}^*$ are feasible points on the boundary of $\Omega$, i.e., $\|\mathbf{s}^*\|_2^3 = \xi = \|\mathbf{s}\|_2^3$, we have

$$(\mathbf{s} - \mathbf{s}^*)^T \nabla_{\mathbf{s}} \hat{m}(\mathbf{s}^*) = -\frac{\nu^*}{2} \|\mathbf{s}^*\|_2 (\mathbf{s} - \mathbf{s}^*)^T \mathbf{s}^* = \frac{\nu^*}{2} \|\mathbf{s}^*\|_2 (\|\mathbf{s}^*\|_2^2 - \mathbf{s}^T \mathbf{s}^*)$$

$$= \frac{\nu^*}{2} \|\mathbf{s}^*\|_2 \left[ \frac{1}{2} \left( \xi^{2/3} + \xi^{2/3} \right) - \mathbf{s}^T \mathbf{s}^* \right] = \frac{\nu^*}{2} \|\mathbf{s}^*\|_2 \left[ \frac{1}{2} \left( \|\mathbf{s}^*\|_2^2 + \|\mathbf{s}\|_2^2 \right) - \mathbf{s}^T \mathbf{s}^* \right], \tag{56}$$

which implies

$$(\mathbf{s} - \mathbf{s}^*)^T \nabla_{\mathbf{s}} \hat{m}(\mathbf{s}^*) = \frac{\nu^*}{4} \|\mathbf{s}^*\|_2 (\mathbf{s} - \mathbf{s}^*)^T (\mathbf{s} - \mathbf{s}^*). \tag{57}$$

Combining (54), (57), and $\nabla_{\mathbf{s}}^2 \hat{m}(\mathbf{s}^*) = \nabla^2 f(\mathbf{x}_k)$ gives

$$\hat{m}(\mathbf{s}) = \hat{m}(\mathbf{s}^*) + \frac{1}{4} \nu^* \|\mathbf{s}^*\|_2 (\mathbf{s} - \mathbf{s}^*)^T (\mathbf{s} - \mathbf{s}^*) + \frac{1}{2} (\mathbf{s} - \mathbf{s}^*)^T \nabla^2 f(\mathbf{x}_k)(\mathbf{s} - \mathbf{s}^*)$$

$$= \hat{m}(\mathbf{s}^*) + \frac{1}{2} (\mathbf{s} - \mathbf{s}^*)^T \left( \nabla^2 f(\mathbf{x}_k) + \frac{\nu^*}{2} \|\mathbf{s}^*\| \mathbf{I} \right) (\mathbf{s} - \mathbf{s}^*). \tag{58}$$

The second-order optimality condition Bertsekas (2017, Proposition 4.3.1) for $\mathbf{z} \in \mathbb{R}^d$ yields

$$\mathbf{z}^T \left( \nabla_{\mathbf{s}}^2 \hat{m}(\mathbf{s}^*) + \nu^* \nabla_{\mathbf{s}}^2 g_\xi(\mathbf{s}^*) \right) \mathbf{z} \geq 0, \tag{59}$$

where

$$\underbrace{\nabla_{\mathbf{s}}^2 \hat{m}(\mathbf{s}^*)}_{\nabla^2 f(\mathbf{x}_k)} + \nu^* \nabla_{\mathbf{s}}^2 g_\xi(\mathbf{s}^*) = \left( \nabla^2 f(\mathbf{x}_k) + \frac{\nu^*}{2} \|\mathbf{s}^*\|_2 \mathbf{I} \right) + \frac{\nu^*}{2} \frac{\mathbf{s}^*(\mathbf{s}^*)^T}{\|\mathbf{s}^*\|_2} \tag{60}$$

such that $\mathbf{z}^T \nabla_{\mathbf{s}} g_\xi(\mathbf{s}^*) = \frac{1}{2} \|\mathbf{s}^*\|_2 \mathbf{z}^T \mathbf{s}^* = 0 \Leftrightarrow \mathbf{z}^T \mathbf{s}^* = 0$. Since $\mathbf{s}^* \neq \mathbf{0}$, we have

$$\mathbf{z}^T \left\{ \left( \nabla^2 f(\mathbf{x}_k) + \frac{\nu^*}{2} \|\mathbf{s}^*\|_2 \mathbf{I} \right) + \frac{\nu^*}{2} \frac{\mathbf{s}^*(\mathbf{s}^*)^T}{\|\mathbf{s}^*\|_2} \right\} \mathbf{z} \geq 0 \Leftrightarrow$$

$$\mathbf{z}^T \left( \nabla^2 f(\mathbf{x}_k) + \frac{\nu^*}{2} \|\mathbf{s}^*\|_2 \mathbf{I} \right) \mathbf{z} + \frac{\nu^*}{2} \frac{(\mathbf{z}^T \mathbf{s}^*)^2}{\|\mathbf{s}^*\|_2} \geq 0. \tag{61}$$

Using $\mathbf{z}^T \mathbf{s}^* = 0$ in (61) we get

$$\mathbf{z}^T \left( \nabla^2 f(\mathbf{x}_k) + \frac{\nu^*}{2} \|\mathbf{s}^*\|_2 \mathbf{I} \right) \mathbf{z} \geq 0. \tag{62}$$

This indicates that $\nabla^2 f(\mathbf{x}_k) + \frac{\nu^*}{2} \|\mathbf{s}^*\|_2 \mathbf{I}$ is positive semi-definite for vectors in the direction of the null-space of $\nabla_{\mathbf{s}} g_\xi(\mathbf{s}^*)$, i.e., perpendicular to $\nabla_{\mathbf{s}} g_\xi(\mathbf{s}^*)$.

It remains to consider vectors $\mathbf{w} \in \mathbb{R}^d$ that do not belong to the null-space of $\nabla_{\mathbf{s}} g_\xi(\mathbf{s}^*)$, i.e., $\mathbf{w}^T \nabla_{\mathbf{s}} g_\xi(\mathbf{s}^*) \neq 0$, and prove that $\nabla^2 f(\mathbf{x}_k) + \frac{\nu^*}{2} \|\mathbf{s}^*\|_2 \mathbf{I}$ is also positive semi-definite. To this end, define the line $\mathbf{s} = \mathbf{s}^* + \alpha \mathbf{w}$ as a function of $\alpha$. Because we are interested in $\mathbf{w}$, such that $\mathbf{w}^T \nabla_{\mathbf{s}} g_\xi(\mathbf{s}^*) \neq 0$, the line intersects the constraint $g_\xi(\mathbf{s}) = 0 \Leftrightarrow \|\mathbf{s}\|_2^3 = \xi$ in two values of $\alpha$. For $\alpha = 0$ we have $\mathbf{s} = \mathbf{s}^*$ and the aforementioned

discussion holds. For $\alpha \neq 0$, $\mathbf{s}$ satisfies $\|\mathbf{s}\|_2^3 = \xi$. In the latter case, we may write $\mathbf{s} - \mathbf{s}^* = \alpha \, \mathbf{w}$. From (58), we arrive at

$$\hat{m}(\mathbf{s}) = \hat{m}(\mathbf{s}^*) + \frac{\alpha^2}{2} \, \mathbf{w}^T \left( \nabla^2 f(\mathbf{x}_k) + \frac{\nu^*}{2} \|\mathbf{s}^*\|_2 \, \mathbf{I} \right) \mathbf{w}, \tag{63}$$

with $\alpha \neq 0$. Given the assumption that $\mathbf{s}^*$ is a minimizer, i.e., $\hat{m}(\mathbf{s}^*) \leq \hat{m}(\mathbf{s})$, (63) implies that $\nabla^2 f(\mathbf{x}_k) + \frac{\nu^*}{2} \|\mathbf{s}^*\|_2 \, \mathbf{I}$ is positive semi-definite. So far, we have shown that if $\mathbf{s}^*$ is a minimizer subject to $\|\mathbf{s}^*\|_2^3 = \xi$, then $\nabla^2 f(\mathbf{x}_k) + \frac{\nu^*}{2} \|\mathbf{s}^*\|_2 \, \mathbf{I}$ is positive semi-definite either in the direction of the null-space of $\nabla_\mathbf{s} g_\xi(\mathbf{s}^*)$ or not. Conversely, if $\nabla^2 f(\mathbf{x}_k) + \frac{\nu^*}{2} \|\mathbf{s}^*\|_2 \, \mathbf{I}$ is positive semi-definite, from (58) and (63), we arrive at $\hat{m}(\mathbf{s}^*) \leq \hat{m}(\mathbf{s})$, i.e., $\mathbf{s}^*$ is a minimizer subject to $\|\mathbf{s}^*\|_2^3 = \xi$.

Regarding the uniqueness of the solution, when $\nabla^2 f(\mathbf{x}_k) + \frac{\nu^*}{2} \|\mathbf{s}^*\|_2 \, \mathbf{I}$ is positive definite, from (58) and (63) we have that $\hat{m}(\mathbf{s}^*) < \hat{m}(\mathbf{s})$, which indicates that $\mathbf{s}^*$ is a unique minimizer subject to $\|\mathbf{s}^*\|_2^3 = \xi$.

▷ **Inactive constraint case.**

In this case, we assume that $\mathbf{s}^*$ is a minimizer of $\hat{m}(\mathbf{s})$ subject to $\|\mathbf{s}^*\|_2^3 < \xi$ when $\nu^* = 0$. From (53) we obtain

$$\nabla^2 f(\mathbf{x}_k)\mathbf{s}^* = -\nabla f(\mathbf{x}_k). \tag{64}$$

From the second-order optimality condition Bertsekas (2017, Proposition 4.3.1), it is implied that $\nabla_{\mathbf{ss}}^2 \mathcal{L}_\xi(\mathbf{s}^*, \nu^*)$ is positive semi-definite. Using the latter fact, along with the fact that $\nabla_{\mathbf{ss}}^2 \mathcal{L}_\xi(\mathbf{s}^*, \nu^*) = \nabla^2 f(\mathbf{x}_k)$ when $\nu^* = 0$, we get that $\nabla^2 f(\mathbf{x}_k)$ is positive semi-definite. This, in turn, implies that we are dealing with a convex problem.

Conversely, when $\nabla^2 f(\mathbf{x}_k)$ is positive semi-definite and $\nu^* = 0$, we can use the Taylor expansion of $\hat{m}(\mathbf{s})$ in (54) along with the fact that

$$\nabla_\mathbf{s} \mathcal{L}_\xi(\mathbf{s}^*, \nu^*) = \mathbf{0} \Leftrightarrow \nabla_\mathbf{s} \hat{m}(\mathbf{s}^*) + \underbrace{\nu^* \nabla_\mathbf{s} g_\xi(\mathbf{s}^*)}_{\text{as } \nu^* = 0}\overset{0}{=} \mathbf{0} \tag{65}$$

$$\Leftrightarrow \nabla_\mathbf{s} \hat{m}(\mathbf{s}^*) = \mathbf{0}$$

to show that $\hat{m}(\mathbf{s}^*) \leq \hat{m}(\mathbf{s})$. This implies that $\mathbf{s}^*$ is a minimizer subject to $\|\mathbf{s}^*\|_2^3 < \xi$. Regarding the uniqueness of the solution, when $\nabla^2 f(\mathbf{x}_k)$ positive definite and $\nu^* = 0$, we can solve (58) w.r.t. $\mathbf{s}^* = -\nabla^2 f(\mathbf{x}_k)^{-1} \nabla f(\mathbf{x}_k)$, which indicates that $\mathbf{s}^*$ is a unique minimizer subject to $\|\mathbf{s}^*\|_2^3 < \xi$.

Given that no assumption has been made on the structure of $\nabla f(\mathbf{x})$, we can repeat the aforementioned proof using $\mathrm{Diag}(\nabla^2 f(\mathbf{x}))$ instead of $\nabla f(\mathbf{x})$ to arrive at Corollary 3.

**Corollary 3.** *A vector $\mathbf{s}^*$ is a minimizer of $\hat{m}(\mathbf{s})$ subject to $\|\mathbf{s}^*\|_2^3 \leq \xi$ if and only if satisfies*

$$\left( \mathrm{Diag}(\nabla^2 f(\mathbf{x}_k)) + \frac{\nu^*}{2} \|\mathbf{s}^*\|_2 \, \mathbf{I} \right) \mathbf{s}^* = -\nabla f(\mathbf{x}_k), \tag{66}$$

$$\mathrm{Diag}(\nabla^2 f(\mathbf{x}_k)) + \frac{\nu^*}{2} \|\mathbf{s}^*\|_2 \, \mathbf{I} \succeq 0, \tag{67}$$

*and $\nu^* (\|\mathbf{s}^*\|_2^3 - \xi) = 0$, where $\nu^* \geq 0$. If $\nabla^2 f(\mathbf{x}_k) + \frac{\nu^*}{2} \|\mathbf{s}^*\|_2 \, \mathbf{I} \succ 0$, then the minimizer $\mathbf{s}^*$ is unique.*

Corollary 3 will be used in the proof of Theorem 3.

## B.2 Proof of Lemma 2

Starting from the primal optimization problem

$$\min_{\mathbf{s} \in \mathbb{R}^d} \mathcal{L}_\xi(\mathbf{s}, \nu) \overset{(8)}{=} \min_{\mathbf{s} \in \mathbb{R}^d, \, \|\mathbf{s}\|_2^2 = \tau} \nabla f(\mathbf{x}_k)^T \mathbf{s} + \frac{1}{2} \mathbf{s}^T \nabla^2 f(\mathbf{x}_k)\mathbf{s} + \frac{\nu}{6} \left( \tau^{3/2} - \xi \right), \tag{68}$$

where $\nu$ is the Lagrange multiplier, the optimal value of the primal problem can be expressed as

$$\min_{\mathbf{s} \in \mathbb{R}^d} \mathcal{L}_\xi(\mathbf{s}, \nu) \stackrel{(8)}{=} \min_{\mathbf{s} \in \mathbb{R}^d} \max_{\tau \geq 0} \max_{r \in \mathcal{D}_\nu} \left\{ \nabla f(\mathbf{x}_k)^T \mathbf{s} + \frac{1}{2} \mathbf{s}^T \nabla^2 f(\mathbf{x}_k) \mathbf{s} + \frac{\nu}{6} \left( \tau^{3/2} - \xi \right) + \frac{r\nu}{4} \left( \|\mathbf{s}\|_2^2 - \tau \right) \right\}, \quad (69)$$

where $r$ is the Lagrange multiplier associated to the constraint $\|\mathbf{s}\|_2^2 = \tau$ (Boyd & Vandenberghe, 2004, Section 5.4). It is essential to highlight that the optimality conditions outlined in Bertsekas (2017, Proposition 4.2.1) explicitly require $r$ to belong to $\mathbb{R}$. However, $r$ is restricted to $\mathcal{D}_\nu$ for reasons that become apparent as the proof unfolds. If the weak duality property is applied to the right-hand side (RHS) of (69), we arrive at

$$\min_{\mathbf{s} \in \mathbb{R}^d} \mathcal{L}_\xi(\mathbf{s}, \nu) \geq \max_{r \in \mathcal{D}_\nu} \min_{\mathbf{s} \in \mathbb{R}^d} \max_{\tau \geq 0} \left\{ \nabla f(\mathbf{x}_k)^T \mathbf{s} + \frac{1}{2} \mathbf{s}^T \nabla^2 f(\mathbf{x}_k) \mathbf{s} + \frac{\nu}{6} \left( \tau^{3/2} - \xi \right) + \frac{r\nu}{4} \left( \|\mathbf{s}\|_2^2 - \tau \right) \right\}. \quad (70)$$

From the first-order optimality condition Bertsekas (2017), the optimal value in the Left Hand Side (LHS) of (68) w.r.t. $\mathbf{s}$ is attained by $\mathbf{s}$ that satisfies $\nabla_\mathbf{s} \mathcal{L}_\xi(\mathbf{s}, \nu) = \mathbf{0}$, i.e.,

$$\left( \nabla^2 f(\mathbf{x}_k) + \frac{\nu}{2} \|\mathbf{s}\|_2 \mathbf{I} \right) \mathbf{s} = -\nabla f(\mathbf{x}_k), \quad \nu \geq 0. \quad (71)$$

At this point, we note that (71) differs from (9), because the stationarity of $\mathcal{L}_\xi(\mathbf{s}, \nu)$ is studied w.r.t. $\mathbf{s}$ only. Denote the RHS of (70) as

$$\begin{aligned} \mathscr{L}_\xi(\mathbf{s}, \nu, r, \tau) &= \nabla f(\mathbf{x}_k)^T \mathbf{s} + \frac{1}{2} \mathbf{s}^T \nabla^2 f(\mathbf{x}_k) \mathbf{s} + \frac{\nu}{6} \left( \tau^{3/2} - \xi \right) + \frac{r\nu}{4} \left( \|\mathbf{s}\|_2^2 - \tau \right) \\ &= \nabla f(\mathbf{x}_k)^T \mathbf{s} + \frac{1}{2} \mathbf{s}^T \left( \nabla^2 f(\mathbf{x}_k) + \frac{\nu r}{2} \mathbf{I} \right) \mathbf{s} + \frac{\nu}{6} \left( \tau^{3/2} - \xi \right) - \frac{r\nu}{4} \tau. \end{aligned} \quad (72)$$

We start with the case $\nu > 0$. Solving $\partial_\tau \mathscr{L}_\xi(\mathbf{s}, \nu, r, \tau) = 0$ w.r.t. $\tau$, we get

$$\tau^* = r^2, \quad (73)$$

where $r \in \mathcal{D}_\nu$. Restricting $r$ in $\mathcal{D}_\nu$, implies that $r > 0$, which in turn implies $\tau^* > 0$, as $\|\mathbf{s}\| = \tau^*$. If $\tau^* = 0$, we have $\|\mathbf{s}\| = 0$, which leads to the trivial solution, i.e., the zero vector. Solving $\nabla_\mathbf{s} \mathscr{L}_\xi(\mathbf{s}, \nu, r, \tau) = \mathbf{0}$ w.r.t. $\mathbf{s}$, we arrive at

$$\nabla f(\mathbf{x}_k) = - \left( \nabla^2 f(\mathbf{x}_k) + \frac{\nu r}{2} \mathbf{I} \right) \mathbf{s}. \quad (74)$$

For $r \in \mathcal{D}_\nu$, we get from (74)

$$\mathbf{s}(\nu, r) = - \left( \nabla^2 f(\mathbf{x}_k) + \frac{\nu r}{2} \mathbf{I} \right)^{-1} \nabla f(\mathbf{x}_k), \quad \nu > 0, \quad (75)$$

which implies the dependence of $\mathbf{s}$ on the variables $\nu$ and $r$. Restricting $r$ in $\mathcal{D}_\nu$, we achieve the invertibility in (75) when $\nu > 0$. Substituting (73) and (74) in (72), we get

$$\mathscr{L}_\xi(\mathbf{s}(\nu, r), \nu, r) = -\frac{1}{2} \mathbf{s}(\nu, r)^T \left( \nabla^2 f(\mathbf{x}_k) + \frac{\nu r}{2} \mathbf{I} \right) \mathbf{s}(\nu, r) - \frac{\nu}{6} \xi - \frac{\nu}{12} r^3. \quad (76)$$

Combining (70) and (76) we get for $\nu > 0$

$$\min_{\mathbf{s} \in \mathbb{R}^d} \mathcal{L}_\xi(\mathbf{s}, \nu) \geq \max_{r \in \mathcal{D}_\nu} \mathscr{L}_\xi(\mathbf{s}(\nu, r), \nu, r). \quad (77)$$

The derivative of (76) w.r.t. $r$ is

$$\partial_r \mathscr{L}_\xi(\mathbf{s}(\nu, r), \nu, r) = \frac{\nu}{4} \left( \|\mathbf{s}(\nu, r)\|_2^2 - r^2 \right). \quad (78)$$

Thus, for any $\nu > 0$, the optimal value in the RHS of (77) is attained for $r^* \in \mathcal{D}_\nu$ that solves

$$\left( \frac{\partial \mathscr{L}_\xi(\mathbf{s}(\nu, r), \nu, r)}{\partial r} \right)_{r=r^*} = 0. \quad (79)$$

Using (78) in (79), we have

$$r^* = \|\mathbf{s}(\nu, r^*)\|_2 \text{ for any } \nu > 0. \tag{80}$$

Restricting $r^*$ in $\mathcal{D}_\nu$, we avoid the trivial solution $\mathbf{s}(\nu, r^*) = \mathbf{0}$ for any $\nu > 0$. This restriction on $r$ in (69) is precisely due to its inclusion in $\mathcal{D}_\nu$.

Using (74) in (8) we attain

$$\mathcal{L}_\xi(\mathbf{s}(\nu, r), \nu) = -\mathbf{s}(\nu, r)^T \left( \nabla^2 f(\mathbf{x}_k) + \frac{\nu\, r}{2}\, \mathbf{I} \right)\, \mathbf{s}(\nu, r) + \frac{1}{2}\, \mathbf{s}(\nu, r)^T\, \nabla^2 f(\mathbf{x}_k)\, \mathbf{s}(\nu, r) + \frac{\nu}{6}\, \|\mathbf{s}(\nu, r)\|_2^3 - \frac{\nu}{6}\xi. \tag{81}$$

Adding and subtracting the terms $\frac{\nu}{12}r^3$ and $\frac{\nu\, r}{4}\, \|\mathbf{s}(\nu, r)\|_2^2$, we get

$$\mathcal{L}_\xi(\mathbf{s}(\nu, r), \nu) = -\mathbf{s}(\nu, r)^T \left( \nabla^2 f(\mathbf{x}_k) + \frac{\nu\, r}{2}\, \mathbf{I} \right)\, \mathbf{s}(\nu, r) + \frac{1}{2}\, \mathbf{s}(\nu, r)^T\, \nabla^2 f(\mathbf{x}_k)\, \mathbf{s}(\nu, r) + \frac{\nu}{6}\, \|\mathbf{s}(\nu, r)\|_2^3$$
$$- \frac{\nu}{6}\xi + \underbrace{\left( \frac{\nu}{12}r^3 - \frac{\nu}{12}r^3 \right)}_{0} + \underbrace{\left( \frac{\nu\, r}{4}\, \|\mathbf{s}(\nu, r)\|_2^2 - \frac{\nu\, r}{4}\, \|\mathbf{s}(\nu, r)\|_2^2 \right)}_{0}. \tag{82}$$

Next, using (76), with appropriate rearrangements we arrive at

$$\mathcal{L}_\xi(\mathbf{s}(\nu, r), \nu) = \mathscr{L}_\xi(\mathbf{s}(\nu, r), \nu, r) + \frac{\nu}{12}r^3 + \frac{\nu}{6}\, \|\mathbf{s}(\nu, r)\|_2^3 - \frac{\nu\, r}{4}\, \|\mathbf{s}(\nu, r)\|_2^2$$
$$= \mathscr{L}_\xi(\mathbf{s}(\nu, r), \nu, r) + \frac{\nu}{12}\left( r^3 + 2\, \|\mathbf{s}(\nu, r)\|_2^3 - 3r\, \|\mathbf{s}(\nu, r)\|_2^2 \right)$$
$$= \mathscr{L}_\xi(\mathbf{s}(\nu, r), \nu, r) + \frac{\nu}{12}\, (\|\mathbf{s}(\nu, r)\|_2 - r)^2\, (r + 2\, \|\mathbf{s}(\nu, r)\|_2). \tag{83}$$

Then, using (78) for $r \in \mathcal{D}_\nu$ and $\nu > 0$ we obtain

$$\mathcal{L}_\xi(\mathbf{s}(\nu, r), \nu) = \mathscr{L}_\xi(\mathbf{s}(\nu, r), \nu, r) + \frac{4}{3\nu}\, \frac{(r + 2\, \|\mathbf{s}(\nu, r)\|_2)}{(r + \|\mathbf{s}(\nu, r)\|_2)^2}\, \left( \partial_r \mathscr{L}_\xi(\mathbf{s}(\nu, r), \nu, r) \right)^2. \tag{84}$$

When (79) is satisfied for some $\nu > 0$, $\mathscr{L}_\xi(\mathbf{s}(\nu, r), \nu, r)$ given by (84) is maximized w.r.t. $r \in \mathcal{D}_\nu$. From (84) we have

$$\mathcal{L}_\xi(\mathbf{s}(\nu, r^*), \nu) = \max_{r \in \mathcal{D}_\nu} \mathscr{L}_\xi(\mathbf{s}(\nu, r), \nu, r). \tag{85}$$

In order to obtain (12), we need to show

$$\mathcal{L}_\xi(\mathbf{s}(\nu, r^*), \nu) = \min_{\mathbf{s} \in \mathbb{R}^d} \mathcal{L}_\xi(\mathbf{s}, \nu), \quad \nu > 0. \tag{86}$$

When $r^* \in \mathcal{D}_\nu$ in (75) and using $r^* = \|\mathbf{s}(\nu, r^*)\|_2$ for some $\nu > 0$ in (80), we get

$$\left( \nabla^2 f(\mathbf{x}_k) + \frac{\nu}{2}\, \|\mathbf{s}(\nu, r^*)\|\, \mathbf{I} \right)\, \mathbf{s}(\nu, r^*) = -\nabla f(\mathbf{x}_k) \tag{87}$$

which implies that $\mathbf{s}(\nu, r^*)$ minimizes $\mathcal{L}_\xi(\mathbf{s}, \nu)$.

We conclude with the case $\nu = 0$. In this case, we observe that (12) is easily attained by applying (70) when equality holds, which concludes the proof.

### B.3 Proof of Theorem 1

From the *weak duality* in (7) and (12) we have

$$\min_{\mathbf{s} \in \mathbb{R}^d} \max_{\nu \geq 0} \mathcal{L}_\xi(\mathbf{s}, \nu) \geq \max_{\nu \geq 0} \min_{\mathbf{s} \in \mathbb{R}^d} \mathcal{L}_\xi(\mathbf{s}, \nu) \overset{(12)}{=} \max_{\nu \geq 0} \max_{r \in \mathcal{D}_\nu} \mathscr{L}_\xi(\mathbf{s}(\nu, r), \nu, r) = \max_{r \in \mathcal{D}_\nu, \nu \geq 0} \mathscr{L}_\xi(\mathbf{s}(\nu, r), \nu, r), \tag{88}$$

where the last term in (88) refers to a joint optimization problem. We start with the case $\nu > 0$. The derivative of (76) w.r.t. $\nu$ is

$$\partial_\nu \mathscr{L}_\xi\big(\mathbf{s}(\nu, r), \nu, r\big) = \frac{r}{4}\, \|\mathbf{s}(\nu, r)\|_2^2 - \frac{\xi}{6} - \frac{r^3}{12}, \tag{89}$$

where $r \in \mathcal{D}_\nu$.

To prove that (88) holds with equality and subsequently prove (17), we study the optimality conditions that maximize the RHS of (88). The optimal value in the RHS of (88) w.r.t $\nu > 0$ is achieved by some $\nu^* > 0$ that solves

$$\left( \frac{\partial \mathscr{L}_\xi\big(\mathbf{s}(\nu, r), \nu, r\big)}{\partial \nu} \right)_{\nu = \nu^*} = 0, \tag{90}$$

for $r \in \mathcal{D}_\nu$. In addition, the optimal value in the RHS of (88) w.r.t $r \in \mathcal{D}_\nu$ is achieved if (79) or equivalently (80) holds. Given that (80) holds for any $\nu > 0$, without loss of generality, we assume that (80) also holds for $\nu^* > 0$, i.e.,

$$r^* = \|\mathbf{s}(\nu^*, r^*)\|_2 \text{ for any } \nu^* > 0. \tag{91}$$

When (91) holds, from (90) we get

$$\xi = \frac{3r^*}{2} \|\mathbf{s}(\nu^*, r^*)\|_2^2 - \frac{r^{*3}}{2}. \tag{92}$$

Solving (14) w.r.t. $\nabla f(\mathbf{x}_k)$ and substituting in (8) we get

$$\mathcal{L}_\xi(\mathbf{s}(\nu, r), \nu) = -\mathbf{s}(\nu, r)^T \left( \nabla^2 f(\mathbf{x}_k) + \frac{\nu\, r}{2}\, \mathbf{I} \right) \mathbf{s}(\nu, r) +$$
$$\frac{1}{2}\, \mathbf{s}(\nu, r)^T\, \nabla^2 f(\mathbf{x}_k)\, \mathbf{s}(\nu, r) + \frac{\nu}{6} \left( \|\mathbf{s}(\nu, r)\|_2^3 - \xi \right). \tag{93}$$

Then, applying (92) for $r \in \mathcal{D}_\nu$, we have

$$\mathcal{L}_\xi(\mathbf{s}(\nu, r), \nu) = -\mathbf{s}(\nu, r)^T \left( \nabla^2 f(\mathbf{x}_k) + \frac{\nu\, r}{2}\, \mathbf{I} \right) \mathbf{s}(\nu, r) + \frac{1}{2}\, \mathbf{s}(\nu, r)^T\, \nabla^2 f(\mathbf{x}_k)\, \mathbf{s}(\nu, r) +$$
$$\frac{\nu}{6} \|\mathbf{s}(\nu, r)\|_2^3 - \frac{\nu\, r}{4} \|\mathbf{s}(\nu, r)\|_2^2 + \frac{\nu\, r^3}{12}. \tag{94}$$

Adding and subtracting $\frac{\nu\, r}{4} \|\mathbf{s}(\nu, r)\|_2^2$ we obtain

$$\mathcal{L}_\xi(\mathbf{s}(\nu, r), \nu) = -\frac{1}{2}\, \mathbf{s}(\nu, r)^T \left( \nabla^2 f(\mathbf{x}_k) + \frac{\nu\, r}{2}\, \mathbf{I} \right) \mathbf{s}(\nu, r) - \frac{\nu\, r}{4} \|\mathbf{s}(\nu, r)\|_2^2 + \frac{\nu}{6} \|\mathbf{s}(\nu, r)\|_2^3 -$$
$$\frac{\nu\, r}{4} \|\mathbf{s}(\nu, r)\|_2^2 + \frac{\nu\, r^3}{12}. \tag{95}$$

Similarly adding and subtracting $\frac{\nu\xi}{6}$ and $\frac{\nu\, r^3}{12}$ reveals the term $\mathscr{L}_\xi(\mathbf{s}(\nu, r), \nu, r)$ yielding

$$\mathcal{L}_\xi(\mathbf{s}(\nu, r), \nu) \overset{(13)}{=} \mathscr{L}_\xi(\mathbf{s}(\nu, r), \nu, r) + \nu \left( \frac{r^3}{12} + \frac{\xi}{6} - \frac{r}{4} \|\mathbf{s}(\nu, r)\|_2^2 \right) + \frac{\nu}{12} \left( r^3 + 2 \|\mathbf{s}(\nu, r)\|_2^3 - 3r \|\mathbf{s}(\nu, r)\|_2^2 \right). \tag{96}$$

The terms inside the first bracket of (96) are identified as $-\partial_\nu \mathscr{L}_\xi\big(\mathbf{s}(\nu, r), \nu, r\big)$, yielding

$$\mathcal{L}_\xi(\mathbf{s}(\nu, r), \nu) = \mathscr{L}_\xi(\mathbf{s}(\nu, r), \nu, r) - \nu\, \partial_\nu \mathscr{L}_\xi\big(\mathbf{s}(\nu, r), \nu, r\big) + \frac{\nu}{12} \left( \|\mathbf{s}(\nu, r)\|_2 - r \right)^2 (r + 2 \|\mathbf{s}(\nu, r)\|_2)$$
$$\overset{(89)}{=} \mathscr{L}_\xi\big(\mathbf{s}(\nu, r), \nu, r\big) + \frac{4}{3\nu} \frac{(r + 2 \|\mathbf{s}(\nu, r)\|_2)}{(r + \|\mathbf{s}(\nu, r)\|_2)^2} \left( \partial_r \mathscr{L}_\xi(\mathbf{s}(\nu, r), \nu, r) \right)^2 - \nu\, \partial_\nu \mathscr{L}_\xi\big(\mathbf{s}(\nu, r), \nu, r\big) \tag{97}$$

and by rearranging terms, we arrive at

$$\mathcal{L}_\xi(\mathbf{s}(\nu, r), \nu) = \mathscr{L}_\xi\big(\mathbf{s}(\nu, r), \nu, r\big) - \nu\, \partial_\nu \mathscr{L}_\xi\big(\mathbf{s}(\nu, r), \nu, r\big) + \frac{4}{3\nu} \frac{(r + 2 \|\mathbf{s}(\nu, r)\|_2)}{(r + \|\mathbf{s}(\nu, r)\|_2)^2} \left( \partial_r \mathscr{L}_\xi(\mathbf{s}(\nu, r), \nu, r) \right)^2. \tag{98}$$

When (79) and (90) hold, $\mathscr{L}_\xi\big(\mathbf{s}(\nu, r), \nu, r\big)$ is maximized and from (98), we have

$$\mathcal{L}_\xi\big(\mathbf{s}(\nu^*, r^*), \nu^*, r^*\big) = \max_{\nu \geq 0, r \in \mathcal{D}_\nu} \mathscr{L}_\xi(\mathbf{s}(\nu, r), \nu, r), \tag{99}$$

where $r^*$ and $\nu^*$ optimize the RHS of (88). Given $r^*$, $\nu^*$, and (99), to show (17), we need to prove

$$\mathcal{L}_\xi\big(\mathbf{s}(\nu^*, r^*), \nu^*, r^*\big) = \min_{\mathbf{s}\in\mathbb{R}^d} \max_{\nu\geq 0} \mathcal{L}_\xi(\mathbf{s}, \nu). \tag{100}$$

To do so, we need to show that the optimal $\mathbf{s}$ in the RHS of (100) equals $\mathbf{s}(\nu^*, r^*)$ in the LHS of (100). The optimal $\mathbf{s}$ in the RHS of (100) satisfies Lemma 1. Thus, by Lemma 1, if $\mathbf{s}(\nu^*, r^*)$ satisfies the CS condition

$$\nu^* \left(\|\mathbf{s}(\nu^*, r^*)\|_2^3 - \xi\right) = 0 \tag{101}$$

and the system of equations

$$\left(\nabla^2 f(\mathbf{x}_k) + \frac{\nu^*}{2} \|\mathbf{s}(\nu^*, r^*)\| \, \mathbf{I}\right) \mathbf{s}(\nu^*, r^*) = -\nabla f(\mathbf{x}_k), \tag{102}$$

then (100) holds. (101) implies $\|\mathbf{s}(\nu^*, r^*)\|_2^3 = \xi$ for $\nu^* > 0$, which is true because of (91). To prove (102), we apply (87), where without loss of generality we replace $\nu > 0$ with $\nu^* > 0$, and the proof is complete. Corollary 1, in the paper's main body, summarizes this proof's main result.

### B.4 Proof of Theorem 2

The proof has two parts. The first part deals with the RHS of (22), while the second part deals with the LHS of (22). A similar procedure is followed to that in Kloft et al. (2009, Proposition 1) to prove Theorem 2.

▷ **First part**.

Let $\mathbf{s}^*$ be the minimizer of (7) which satisfies the feasibility condition $g_\xi(\mathbf{s}^*) \leq 0$. We want to show that when $M = \nu^*$, $\mathbf{s}^*$ is also a minimizer of (5). From Lemma 1, we recall that $\nu^* (\|\mathbf{s}^*\|_2^3 - \xi) = 0$. Consequently,

$$\min_{\mathbf{s}\in\mathbb{R}^d} \max_{\nu\geq 0} \mathcal{L}_\xi(\mathbf{s}, \nu) = \mathcal{L}_\xi(\mathbf{s}^*, \nu^*) = \hat{m}(\mathbf{s}^*) + \underbrace{\frac{\nu^*}{6}\left(\|\mathbf{s}^*\|_2^3 - \xi\right)}_{\text{0 from CS condition}}{}^{\nearrow 0} = \hat{m}(\mathbf{s}^*). \tag{103}$$

From Corollary 1, we have

$$\min_{\mathbf{s}\in\mathbb{R}^d} \max_{\nu\geq 0} \mathcal{L}_\xi(\mathbf{s}, \nu) = \max_{\nu\geq 0} \underbrace{\min_{\mathbf{s}\in\mathbb{R}^d} \mathcal{L}_\xi(\mathbf{s}, \nu)}_{\psi(\nu)} = \max_{\nu\geq 0} \psi(\nu) = \psi(\nu^*), \tag{104}$$

where $\psi(\nu)$ is the dual function of the constrained optimization problem (7). From (103) and (104) we have

$$\hat{m}(\mathbf{s}^*) = \psi(\nu^*) = \min_{\mathbf{s}\in\mathbb{R}^d} \mathcal{L}_\xi(\mathbf{s}, \nu^*) = \min_{\mathbf{s}\in\mathbb{R}^d} \left\{\hat{m}(\mathbf{s}) + \frac{\nu^*}{6} \overbrace{\left(\|\mathbf{s}\|_2^3 - \xi\right)}^{\leq 0,\text{ by feasibility}}\right\}$$

$$\leq \min_{\mathbf{s}\in\mathbb{R}^d} \hat{m}(\mathbf{s}) = \hat{m}(\mathbf{s}^*) + \frac{1}{6} \underbrace{\nu^*\left(\|\mathbf{s}^*\|_2^3 - \xi\right)}_{\text{0 from CS condition}}{}^{\nearrow 0} = \hat{m}(\mathbf{s}^*). \tag{105}$$

Since the first and the last term in (105) are equal, due to the CS condition, the in-between inequalities hold with equality, i.e.,

$$\min_{\mathbf{s}\in\mathbb{R}^d} \left\{\hat{m}(\mathbf{s}) + \frac{\nu^*}{6}\left(\|\mathbf{s}\|_2^3 - \xi\right)\right\} = \hat{m}(\mathbf{s}^*) + \frac{\nu^*}{6}\left(\|\mathbf{s}^*\|_2^3 - \xi\right). \tag{106}$$

Removing the constant term $-\frac{\nu^*}{6}\xi$ from both sides of (106), we obtain

$$\min_{\mathbf{s}\in\mathbb{R}^d} m_{\nu^*}(\mathbf{s}) = \hat{m}(\mathbf{s}^*) + \frac{\nu^*}{6}\|\mathbf{s}^*\|_2^3 \overset{(7)}{=} f(\mathbf{x}_k) + \nabla f(\mathbf{x}_k)^T\mathbf{s}_k^* + \frac{1}{2}\mathbf{s}^{*T}\nabla^2 f(\mathbf{x}_k)\mathbf{s}^* + \frac{\nu^*}{6}\|\mathbf{s}^*\|_2^3 \overset{(6)}{=} m_{\nu^*}(\mathbf{s}^*), \tag{107}$$

which implies that $\mathbf{s}^*$ is also a minimizer of (5) with $M = \nu^*$ and the first part of the proof is complete.

▷ **Second part**.

Let $\mathbf{s}^*$ be a minimizer of (5). We should prove that $\mathbf{s}^*$ is also a minimizer of (7) when $\xi = \|\mathbf{s}^*\|_2^3$. For such $\xi$, $g_\xi(\mathbf{s}^*) = 0$. We prove the second part by contradiction. Suppose, $\mathbf{s}^*$ is not optimal in (7), i.e., there is a feasible point $\mathbf{s}$ such that $\hat{m}(\mathbf{s}) \leq \hat{m}(\mathbf{s}^*)$. For this feasible point we also have $g_\xi(\mathbf{s}) \leq 0$ and $g_\xi(\mathbf{s}) \leq g_\xi(\mathbf{s}^*)$. Then, we get,

$$\hat{m}(\mathbf{s}) \leq \hat{m}(\mathbf{s}^*) \Leftrightarrow \hat{m}(\mathbf{s}) + g_\xi(\mathbf{s}) \leq \hat{m}(\mathbf{s}^*) + g_\xi(\mathbf{s}^*)$$

$$\Leftrightarrow \hat{m}(\mathbf{s}) + \frac{\nu}{6}\left(\|\mathbf{s}\|_2^3 - \xi\right) \leq \hat{m}(\mathbf{s}^*) + \frac{\nu}{6}\left(\|\mathbf{s}^*\|_2^3 - \xi\right). \quad (108)$$

Adding $\frac{\nu}{6}\xi$ in both sides of the last inequality in (108), using the definition of $m_M(\mathbf{s})$ (6) with $M = \nu$, and applying the definition of $\hat{m}(\mathbf{s})$ (7), we get

$$m_\nu(\mathbf{s}) \leq m_\nu(\mathbf{s}^*). \quad (109)$$

This is a contradiction, because $\mathbf{s}^*$ is a minimizer of (5). Hence, $\mathbf{s}^*$ is also a minimizer of (7), when $g_\xi(\mathbf{s}^*) = 0 \Leftrightarrow \xi = \|\mathbf{s}^*\|_2^3$, which concludes the second part of the proof.

## B.5 Proof of Lemma 5

We follow similar lines to the proof of (Kohler & Lucchi, 2017, Lemma 6 and Theorem 7). Note that $\mathcal{B}_k^g$ is used instead of $\mathcal{B}_k$ to emphasize that the deviation bound in (44) and the sampling scheme in (45) are specifically derived using information associated with $\mathbf{g}_k$.

The proof resorts to Vector Bernstein's inequality in Lemma 19 (discussed in Appendix B.13). Let us define the centered gradient

$$\mathbf{z}_{i,k}^s = \nabla f_i(\mathbf{x}_k) - \nabla f(\mathbf{x}_k), \quad (110)$$

where $i = 1, \ldots, n = |\mathcal{B}_k^g|$. First, we show

$$\left\|\mathbf{z}_{i,k}^s\right\|_2 \leq \|\nabla f_i(\mathbf{x}_k)\| + \|\nabla f(\mathbf{x}_k)\|_2 \leq 2L_f, \quad (111)$$

which implies $\left\|\mathbf{z}_{i,k}^s\right\|_2^2 \leq 4L_f^2$. Accordingly, $\sigma^2 \triangleq 4L_f^2$ in Lemma 19. In (111), we have used

$$\|\nabla f(\mathbf{x}_k)\|_2 \leq \frac{1}{n}\sum_{i=1}^n \|\nabla f_i(\mathbf{x})\|_2 \leq \frac{1}{n}\sum_{i=1}^n L_f = L_f, \quad (112)$$

where the triangle inequality and Assumption 1 have been applied. Then, we have

$$\mathbf{z}_k = \frac{1}{|\mathcal{B}_k^g|}\sum_{i=1}^n \mathbf{z}_{i,k}^s = \mathbf{g}_k - \nabla f(\mathbf{x}_k). \quad (113)$$

Using (113) in Lemma 19 for $n = |\mathcal{B}_k^g|$ and $\sigma^2 = 4L_f^2$ yields

$$\Pr(\|\mathbf{g}_k - \nabla f(\mathbf{x}_k)\|_2 \geq \epsilon) \leq \exp\left(-|\mathcal{B}_k^g|\frac{\epsilon^2}{32L_f^2} + \frac{1}{4}\right). \quad (114)$$

Next, we require that the probability of the gradient deviation $\Pr(\|\mathbf{g}_k - \nabla f(\mathbf{x}_k)\|_2 \geq \epsilon)$ is less than some $\delta \in (0, 1]$, i.e.,

$$\exp\left(-|\mathcal{B}_k^g|\frac{\epsilon^2}{32L_f^2} + \frac{1}{4}\right) \leq \delta \Leftrightarrow \epsilon \geq 4\sqrt{2}L_f\sqrt{\frac{\ln\frac{1}{\delta} + \frac{1}{4}}{|\mathcal{B}_k^g|}}. \quad (115)$$

To derive (44), we use (115) in $\|\mathbf{g}_k - \nabla f(\mathbf{x}_k)\|_2 \geq \epsilon$, along with Assumption 2 to get

$$\epsilon \leq \|\mathbf{g}_k - \nabla f(\mathbf{x}_k)\|_2 \geq C_g\|\mathbf{s}_k\|_2^2 \Leftrightarrow 4\sqrt{2}L_f\sqrt{\frac{\ln\frac{1}{\delta} + \frac{1}{4}}{|\mathcal{B}_k^g|}} \leq C_g\|\mathbf{s}_k\|_2^2, \quad (116)$$

which yields (45). Using the complementary probability

with $\delta \in (0, 1]$, it is implied that

$$\|\mathbf{g}_k - \nabla f(\mathbf{x}_k)\|_2 \leq \epsilon$$

is fulfilled with high probability $1 - \delta$ when (115) holds. The latter derives (44), and the proof is complete.

### B.6 Proof of Lemma 6

Following similar lines to Kohler & Lucchi (2017, Lemma 8 and Theorem 9) and using $\mathcal{B}_k^H$ instead of $\mathcal{B}_k$ to emphasize that the deviation bound in (46) and the sampling scheme in (47) are obtained using information related to $\mathbf{B}_k$, we get

$$\mathbf{B}_k \overset{(30)}{=} \frac{1}{\mathcal{S}} \sum_{s=1}^{\mathcal{S}} \mathrm{Diag}\left(\mathbf{H}_k \mathbf{v}_s \odot \mathbf{v}_s\right) = \frac{1}{\mathcal{S}} \sum_{s=1}^{\mathcal{S}} \mathrm{Diag}\left(\left(\frac{1}{|\mathcal{B}_k^g|} \sum_{i \in \mathcal{B}_k^g} \nabla^2 f_i(\mathbf{x}_k)\right) \mathbf{v}_s \odot \mathbf{v}_s\right)$$

$$= \frac{1}{\mathcal{S}} \sum_{s=1}^{\mathcal{S}} \frac{1}{|\mathcal{B}_k^g|} \sum_{i \in \mathcal{B}_k^g} \mathrm{Diag}\left(\nabla^2 f_i(\mathbf{x}_k) \mathbf{v}_s \odot \mathbf{v}_s\right). \quad (117)$$

Let

$$\mathbf{B}_{i,k}^s = \mathrm{Diag}\left(\nabla^2 f_i(\mathbf{x}_k) \mathbf{v}_s \odot \mathbf{v}_s\right). \quad (118)$$

For $\mathbf{A} \in \mathbb{R}^{d \times d}$ it is known that $\|\mathbf{A}\|_2 \leq \|\mathbf{A}\|_F$ (Golub & Van Loan, 2012). Accordingly, for $\mathbf{B}_{i,k}^s$ we obtain

$$\left\|\mathbf{B}_{i,k}^s\right\|_2 = \left\|\mathrm{Diag}\left(\nabla^2 f_i(\mathbf{x}_k) \mathbf{v}_s \odot \mathbf{v}_s\right)\right\|_2 \leq \left\|\mathrm{Diag}\left(\nabla^2 f_i(\mathbf{x}_k) \mathbf{v}_s \odot \mathbf{v}_s\right)\right\|_F$$

$$= \sqrt{\sum_{j=1}^{d} \left([\nabla^2 f_i(\mathbf{x}_k) \mathbf{v}_s]_j [\mathbf{v}_s]_j\right)^2} = \sqrt{\sum_{j=1}^{d} \left([\nabla^2 f_i(\mathbf{x}_k) \mathbf{v}_s]_j\right)^2}, \quad (119)$$

where $[\mathbf{v}_s]_j = \pm 1$, yielding

$$\left\|\mathbf{B}_{i,k}^s\right\|_2 \leq \left\|\nabla^2 f_i(\mathbf{x}_k) \mathbf{v}_s\right\|_2 \leq \left\|\nabla^2 f_i(\mathbf{x}_k)\right\|_2 \|\mathbf{v}_s\|_2. \quad (120)$$

For $\mathbf{v}_s \in \mathbb{R}^d$, $\|\mathbf{v}_s\|_2 \leq \sqrt{d} \|\mathbf{v}_s\|_\infty$ Gould et al. (1999), where $\|\mathbf{v}_s\|_\infty = \max_{1 \leq i \leq d} |[\mathbf{v}_s]_i| = 1$. This allows us to rewrite (120) as

$$\left\|\mathbf{B}_{i,k}^s\right\|_2 \leq \sqrt{d} \left\|\nabla^2 f_i(\mathbf{x}_k)\right\|_2 \|\mathbf{v}_s\|_\infty \leq \sqrt{d} \left\|\nabla^2 f_i(\mathbf{x}_k)\right\|_2. \quad (121)$$

As a result

$$\left\|\mathbf{B}_{i,k}^s\right\|_2 \leq \sqrt{d} L_g, \quad (122)$$

because $\left\|\nabla^2 f_i(\mathbf{x}_k)\right\|_2 \leq L_g$ due to Assumption 1.

To apply the Matrix Bernstein's inequality in Lemma 22, define the centred Hessian matrix

$$\mathbf{Z}_{i,k}^s = \mathbf{B}_{i,k}^s - \mathrm{Diag}(\nabla^2 f(\mathbf{x}_k)), \quad (123)$$

where $i = 1, \ldots, |\mathcal{B}_k^H|$.

Let $n' = |\mathcal{B}_k^H|$. From Lemma 8, using Assumption 1, and applying the triangle inequality we have

$$\left\|\mathrm{Diag}(\nabla^2 f(\mathbf{x}))\right\|_2 \leq \left\|\nabla^2 f(\mathbf{x})\right\|_2 \leq \frac{1}{n'} \sum_{i=1}^{n'} \left\|\nabla^2 f_i(\mathbf{x})\right\|_2 \leq \frac{1}{n'} \sum_{i=1}^{n'} L_g \leq L_g, \quad (124)$$

Using (122), (124), and applying triangle inequality yields

$$\left\|\mathbf{Z}_{i,k}^s\right\|_2 = \left\|\mathbf{B}_{i,k}^s - \mathrm{Diag}(\nabla^2 f(\mathbf{x}))\right\|_2 \leq \left\|\mathbf{B}_{i,k}^s\right\|_2 + \left\|\mathrm{Diag}(\nabla^2 f(\mathbf{x}))\right\|_2 \leq (\sqrt{d} + 1) L_g, \quad (125)$$

which implies

$$\left\|\mathbf{Z}_{i,k}^s\right\|_2 \le \sqrt{d}L_g, \tag{126}$$

as $d \gg 1$. Let

$$\mathbf{Z}_k \triangleq \frac{1}{\mathcal{S}}\sum_{s=1}^{\mathcal{S}}\frac{1}{|\mathcal{B}_k^H|}\sum_{i\in\mathcal{B}_k^H}\mathbf{Z}_{i,k}^s \overset{(123)}{\underset{(118)}{=}} \mathbf{B}_k - \mathrm{Diag}(\nabla^2 f(\mathbf{x}_k)). \tag{127}$$

Let also

$$\sigma^2 \triangleq \left\|\sum_{s=1}^{\mathcal{S}}\sum_{i\in\mathcal{B}_k^H}\mathbb{E}\left[\mathbf{Z}_{i,k}^s{}^2\right]\right\|_2 \le \sum_{s=1}^{\mathcal{S}}\sum_{i\in\mathcal{B}_k^H}\left\|\mathbb{E}\left[\mathbf{Z}_{i,k}^s{}^2\right]\right\|_2 \le \sum_{s=1}^{\mathcal{S}}\sum_{i\in\mathcal{B}_k^H}\mathbb{E}\left[\left\|\mathbf{Z}_{i,k}^s{}^2\right\|_2\right]$$

$$\le \sum_{s=1}^{\mathcal{S}}\sum_{i\in\mathcal{B}_k^H}\mathbb{E}\left[\left\|\mathbf{Z}_{i,k}^s\right\|_2\left\|\mathbf{Z}_{i,k}^s\right\|_2\right] \le d\,\mathcal{S}\,|\mathcal{B}_k^H|\,L_g^2, \tag{128}$$

which implies $\sigma^2 \le d\,\mathcal{S}\,|\mathcal{B}_k^H|\,L_g^2$. Also let $K \triangleq \sqrt{d}\,L_g$. Using the latter in Lemma 22 implies

$$P\left(\left\|\sum_{s=1}^{\mathcal{S}}\sum_{i\in\mathcal{B}_k^H}\mathbf{Z}_{i,k}^s\right\|_2 \ge t\right) \le \begin{cases} 2d\exp\left(\frac{3}{8}\frac{-t^2}{d\mathcal{S}|\mathcal{B}_k^H|L_g^2}\right), & t \le \frac{\sigma^2}{\sqrt{d}L_g} \\ 2d\exp\left(\frac{3}{8}\frac{-t}{\sqrt{d}L_g}\right), & t > \frac{\sigma^2}{\sqrt{d}L_g}. \end{cases} \tag{129}$$

Using (127) in (129) for $t \le \frac{\sigma^2}{\sqrt{d}L_g}$ and setting $\epsilon = \frac{t}{|\mathcal{B}_k^H|\mathcal{S}}$ yields

$$\Pr(\left\|\mathbf{B}_k - \mathrm{Diag}(\nabla^2 f(\mathbf{x}_k))\right\|_2 \ge \epsilon) \le 2d\exp\left(-\frac{3}{8}\mathcal{S}|\mathcal{B}_k^H|\left(\frac{\epsilon}{\sqrt{d}L_g}\right)^2\right). \tag{130}$$

For some $\delta \in (0,1]$, we are interested in the upper bound

$$\Pr(\left\|\mathbf{B}_k - \mathrm{Diag}(\nabla^2 f(\mathbf{x}_k))\right\|_2 \ge \epsilon) \le \delta, \tag{131}$$

which implies

$$2d\exp\left(-\frac{3}{8}\mathcal{S}|\mathcal{B}_k^H|\frac{\epsilon^2}{dL_g^2}\right) \le \delta \Leftrightarrow \epsilon \ge \sqrt{d}L_g\sqrt{\frac{\ln\frac{2d}{\delta}}{\mathcal{S}|\mathcal{B}_k^H|}}. \tag{132}$$

Similarly, using (127) in (129) for $t > \frac{\sigma^2}{\sqrt{d}L_g}$ and setting $\epsilon = \frac{t}{|\mathcal{B}_k^H|\mathcal{S}}$ yields

$$\Pr(\left\|\mathbf{B}_k - \mathrm{Diag}(\nabla^2 f(\mathbf{x}_k))\right\|_2 \ge \epsilon) \le 2d\exp\left(-\frac{3}{8}\mathcal{S}|\mathcal{B}_k^H|\frac{\epsilon}{\sqrt{d}L_g}\right). \tag{133}$$

Again, we are interested in the probability $\Pr(\left\|\mathbf{B}_k - \mathrm{Diag}(\nabla^2 f(\mathbf{x}_k))\right\|_2) \ge \epsilon)$ is less than some $\delta \in (0,1]$, i.e.,

$$2d\exp\left(-\frac{3}{8}\mathcal{S}|\mathcal{B}_k^H|\frac{\epsilon}{\sqrt{d}L_g}\right) \le \delta \Leftrightarrow \epsilon \ge \sqrt{d}L_g\frac{\ln\frac{2d}{\delta}}{\mathcal{S}|\mathcal{B}_k^H|}. \tag{134}$$

We have that for $x \le 1$ that $e^{-x} \le e^{-x^2}$. Thus, for $d \gg 1$ we have $\frac{\epsilon}{\sqrt{d}L_g} < 1$ and the tightest upper bound of $\Pr(\left\|\mathbf{B}_k - \mathrm{Diag}(\nabla^2 f(\mathbf{x}_k))\right\|_2 \ge \epsilon)$ is (133). (134) indicates how large $\epsilon$ must be for the probability of a deviation in (133) to be at most $\delta$, depending on the number of Hutchinson samples $\mathcal{S}$, the mini-batch size $|\mathcal{B}_k^H|$, the parameter dimension $d$, and the smoothness constant $L_g$. Next, using (134) in Assumption 3, we get

$$\epsilon \le \left\|\mathbf{B}_k - \mathrm{Diag}(\nabla^2 f(\mathbf{x}_k))\right\|_2 \le C_B\,\left\|\mathbf{s}_k\right\|_2 \Leftrightarrow \sqrt{d}L_g\frac{\ln\frac{2d}{\delta}}{\mathcal{S}|\mathcal{B}_k^H|} \le C_B\,\left\|\mathbf{s}_k\right\|_2 \Leftrightarrow |\mathcal{B}_k^H| \ge \sqrt{d}L_g\frac{\ln\frac{2d}{\delta}}{\mathcal{S}\,\left\|\mathbf{s}_k\right\|_2 C_B}, \tag{135}$$

which yields (47).

Using the complementary bound

$$\Pr(\left\|\mathbf{B}_k - \mathrm{Diag}(\nabla^2 f(\mathbf{x}_k))\right\|_2 \le \epsilon) \ge 1 - \delta,$$

it is implied that

$$\left\|\mathbf{B}_k - \mathrm{Diag}(\nabla^2 f(\mathbf{x}_k))\right\|_2 \le \epsilon$$

is fulfilled with high probability $1 - \delta$ when (134) holds. The latter is (46) and the proof is complete.

### B.7 Proof of Lemma 7

In the following, $\mathbf{g}_k$ and $\mathbf{B}_k$ are utilized instead of $\nabla f(\mathbf{x}_k)$ and $\nabla^2 f(\mathbf{x}_k)$. This substitution is performed because $\mathbf{g}_k$ and $\mathbf{B}_k$ are directly used in Algorithms 1 and 2, both of which operate on data batches. In this manner, $\mathbf{s}(\nu, r)$ is now defined in terms of $\mathbf{B}_k$ and $\mathbf{g}_k$, rather than $\mathrm{Diag}(\nabla^2 f(\mathbf{x}_k))$ and $\nabla f(\mathbf{x}_k)$, respectively.

To obtain the minimizer $\mathbf{s}(\nu, r)$ in (20), we need to solve w.r.t. $\nu$ and $r$ the system of equations

$$\partial_r \mathscr{L}_\xi(\nu, r) = 0 \quad \text{and} \quad \partial_\nu \mathscr{L}_\xi(\nu, r) = 0. \tag{136}$$

The solution $(\nu, r)$ in (136) can also be computed sequentially. We start with the case $\nu > 0$. In this case, we can first solve $\partial_r \mathscr{L}_\xi(\nu, r) = 0$ w.r.t. $r \in \mathcal{D}_\nu$. Then, the optimal $r$, can be used to solve $\partial_\nu \mathscr{L}_\xi(\nu, r) = 0$ w.r.t. $\nu$ to obtain the optimal $\nu > 0$. Solving $\partial_r \mathscr{L}_\xi(\nu, r) = 0$ w.r.t. $r$, yields

$$\frac{\nu}{4}\left\{ \left[ -\mathbf{g}_k^T \left( \mathbf{B}_k + \frac{\nu\, r}{2}\mathbf{I}\right)^{-1}\right]\left[ -\left(\mathbf{B}_k + \frac{\nu\, r}{2}\mathbf{I}\right)^{-1}\mathbf{g}_k\right]\right\} - \frac{\nu\, r^2}{4} = 0 \stackrel{(18)}{\Leftrightarrow} \nu\left(\|\mathbf{s}(\nu,r)\|_2^2 - r^2\right) = 0. \tag{137}$$

For $\mathbf{s}(\nu, r) \ne \mathbf{0}$, $r \in \mathcal{D}_\nu$, and $\nu > 0$. Fom (137), we obtain the root

$$r = \|\mathbf{s}(\nu, r)\|_2. \tag{138}$$

Then, the optimal $\nu$ can be computed by solving $\partial_\nu \mathscr{L}_\xi(\nu, r) \stackrel{(13)}{=} 0$ w.r.t. $\nu$, i.e.,

$$\frac{r}{4}\left\{ \left[ -\mathbf{g}_k^T \left( \mathbf{B}_k + \frac{\nu\, r}{2}\mathbf{I}\right)^{-1}\right]\left[ -\left(\mathbf{B}_k + \frac{\nu\, r}{2}\mathbf{I}\right)^{-1}\mathbf{g}_k\right]\right\} - \frac{\xi}{6} - \frac{r^3}{12} = 0, \tag{139}$$

which is rewritten as

$$\frac{r}{4}\|\mathbf{s}(\nu, r)\|_2^2 - \frac{\xi}{6} - \frac{r^3}{12} = 0 \tag{140}$$

for some $\xi > 0$. Substituting (138) in (140), the optimal $r$ is given by

$$r = \sqrt[3]{\xi}. \tag{141}$$

Using (141) in (140), the optimal $\nu$ can be computed by solving

$$\omega(\nu, r) = \|\mathbf{s}(\nu, r)\|_2 - \sqrt[3]{\xi} = 0, \tag{142}$$

w.r.t. $\nu$ for $r$ fixed. It is shown in Conn et al. (2000, Section 7.3.3) that instead of solving (142), it is more preferable to solve

$$\phi(\nu, r) = \frac{1}{\|\mathbf{s}(\nu, r)\|_2} - \frac{1}{\sqrt[3]{\xi}} = 0. \tag{143}$$

We conclude with the case $\nu = 0$. In this case, the minimizer $\mathbf{s}(\nu, r)$ in (20) is handled by Algorithm 2, which concludes the proof.

### B.8 Preliminaries for Lemmata 6 and 9

**Lemma 8.** *For* $\mathbf{M} \in \mathbb{R}^{d \times d}$ *and* $\mathrm{Diag}(\mathbf{M})$ *we have*

$$\|\mathbf{M}\|_2 \geq \|\mathrm{Diag}(\mathbf{M})\|_2. \tag{144}$$

*Proof.* First, we prove

$$\|\mathrm{Diag}(\mathbf{M})\|_2 = \max_k |\mathrm{Diag}(\mathbf{M})_{kk}|, \tag{145}$$

Let $d^* = \max_k \mathrm{Diag}(\mathbf{M})_{kk}$. Then, from the definition of the spectral norm, we have

$$\|\mathrm{Diag}(\mathbf{M})\|_2 = \max_{\|\mathbf{x}\|_2=1} \|\mathrm{Diag}(\mathbf{M})\mathbf{x}\|_2 \leq \max_{\|\mathbf{x}\|_2=1} \sqrt{\sum_k (\mathrm{Diag}(\mathbf{M})_{kk}\mathbf{x}_k)^2} \leq |d^*| \max_{\|\mathbf{x}\|_2=1} \sqrt{\sum_k \mathbf{x}_k^2}, \tag{146}$$

which leads to

$$\|\mathrm{Diag}(\mathbf{M})\|_2 \leq |d^*|. \tag{147}$$

Let $\mathbf{e}_m$ be the vector of all zeros except a 1 in the $m$th position, where $m = \arg\max_i \mathrm{Diag}(\mathbf{M})_{ii}$. Then,

$$\|\mathrm{Diag}(\mathbf{M})\|_2 = \max_{\|\mathbf{x}\|_2=1} \|\mathrm{Diag}(\mathbf{M})\mathbf{x}\|_2 \geq \|\mathrm{Diag}(\mathbf{M})\mathbf{e}_m\|_2, \tag{148}$$

which leads to

$$\|\mathrm{Diag}(\mathbf{M})\|_2 \geq |d^*|. \tag{149}$$

(147) and (149) imply (145). From the definition of the spectral norm, we have

$$\|\mathbf{M}\|_2 = \max_{\|\mathbf{x}\|_2=1} \sqrt{\frac{\mathbf{x}^T\mathbf{M}^T\mathbf{M}\mathbf{x}}{\|\mathbf{x}\|_2^2}} = \max_{\|\mathbf{x}\|_2=1} \frac{\|\mathbf{M}\mathbf{x}\|_2}{\|\mathbf{x}\|_2} = \max_{\mathbf{x},\mathbf{y}\neq\mathbf{0}} \frac{|\mathbf{x}^T\mathbf{M}\mathbf{y}|}{\|\mathbf{x}\|_2 \|\mathbf{y}\|_2} \geq |\mathbf{e}_j^T\mathbf{M}\mathbf{e}_i| = |\mathbf{M}_{ij}|. \tag{150}$$

Restricting (150) in the diagonal elements of $\mathbf{M}$ gives

$$\|\mathbf{M}\|_2 \geq \max_i |\mathrm{Diag}(\mathbf{M})_{ii}|, \tag{151}$$

which using (145) leads to

$$\|\mathbf{M}\|_2 \geq \|\mathrm{Diag}(\mathbf{M})\|_2, \tag{152}$$

which is (144). This inequality becomes an equality when $\mathbf{M}$ is a diagonal matrix. Therefore, (144) provides the tightest possible bound in this case. $\qquad\square$

### B.9 Lemmata 9 and 10

Given the Lipschitz continuity assumption of $\nabla^2 f(\mathbf{x})$, Lemma 9 introduces the Lipschitz continuity of $\mathrm{Diag}(\nabla^2 f(\mathbf{x}))$. Lemma 9 is used in Lemmata 10 and 17. Lemma 10 is an adaptation of Nesterov & Polyak (2006, Lemma 1) tailored to fit the context of this analysis. Lemma 10 is used in Lemmata 14, 15, and 16.

**Lemma 9.** *If* $\nabla^2 f(\mathbf{x})$ *is Lipschitz continuous in* $\mathcal{F}$, $\mathrm{Diag}(\nabla^2 f(\mathbf{x}))$ *is also Lipschitz continuous, i.e.,*

$$\left\|\mathrm{Diag}(\nabla^2 f(\mathbf{x})) - \mathrm{Diag}(\nabla^2 f(\mathbf{y}))\right\|_2 \leq L_H \|\mathbf{x} - \mathbf{y}\|_2. \tag{153}$$

*Proof.* The proof of the lemma is easily obtained by combining the Lipschitz continuity of the Hessian matrix in Assumption 1 (see also Remark 1) with Lemma 8 in Appendix B.8. In Lemma 8, we use $\mathbf{M} = \nabla^2 f(\mathbf{x}) - \nabla^2 f(\mathbf{y})$. $\qquad\square$

**Lemma 10.** *For any* $\mathbf{x}$ *and* $\mathbf{y}$ *in* $\mathcal{F}$*, we have*

$$\left\|\nabla f(\mathbf{y}) - \nabla f(\mathbf{x}) - \mathrm{Diag}(\nabla^2 f(\mathbf{x}))(\mathbf{y} - \mathbf{x})\right\|_2 \leq \frac{L_H}{2} \|\mathbf{y} - \mathbf{x}\|_2 \tag{154}$$

*and*

$$\left| f(\mathbf{y}) - f(\mathbf{x}) - \nabla f(\mathbf{x})^T (\mathbf{y} - \mathbf{x}) - \frac{1}{2}(\mathbf{y} - \mathbf{x})^T \mathrm{Diag}(\nabla^2 f(\mathbf{x}))(\mathbf{y} - \mathbf{x}) \right| \leq \frac{L_H}{6} \|\mathbf{y} - \mathbf{x}\|_2^3. \tag{155}$$

*Proof.* Nesterov & Polyak (2006, Lemma 1) does not make any assumption on the structure of $\nabla^2 f(\mathbf{x})$. The only assumption to derive Nesterov & Polyak (2006, Lemma 1) is the Lipschitz continuity of $\nabla^2 f(\mathbf{x})$. Thus, given Lemma 9 and following the proof guidelines in Nesterov & Polyak (2006, Lemma 1), (154) and (155) are easily derived, which concludes the proof. □

### B.10 Details for Algorithm 2

Here, Algorithm 2 is discussed when $\mathbf{B}_k$ and $\mathbf{g}_k$ are utilized instead of $\nabla f(\mathbf{x}_k)$ and $\nabla^2 f(\mathbf{x}_k)$, respectively. This is done because $\mathbf{B}_k$ and $\mathbf{g}_k$ are directly involved in the application of Algorithm 1, which operates on data batches. In this manner, $\mathbf{s}(\nu, r)$ and $\phi(\nu, r)$ are now defined with respect to $\mathbf{B}_k$ and $\mathbf{g}_k$, rather than $\mathrm{Diag}(\nabla^2 f(\mathbf{x}_k))$ and $\nabla f(\mathbf{x}_k)$, respectively.

Lemma 11 provides some useful properties of $\phi(\nu, r)$ exploited in line 18 of Algorithm 2. Lemma 11 is used in Lemma 12 which shows that for some $r \in \mathcal{D}_\nu$, Newton-Raphson updates in line 18 of Algorithm 2 converge to the roots of $\phi(\nu, r) = 0$ w.r.t. $\nu > 0$. However, the Newton-Raphson method may diverge on its own, and appropriate safeguards are necessary to prevent this. These safeguards are adopted from (Conn et al., 2000, Algorithm 7.3.6) in Algorithm 2. Lemma 13 stems from Conn et al. (2000, Lemma 7.3.5) adapted to the analysis here. Lemma 13 provides the termination rule used in Algorithm 2.

**Lemma 11.** *Let*

$$\tilde{\mathbf{H}}_k(\nu, r) \overset{\triangle}{=} \mathbf{B}_k + \frac{\nu\, r}{2}\, \mathbf{I}. \tag{156}$$

*Suppose* $\mathbf{g}_k \neq \mathbf{0}$ *and* $\nu\, r > \max\{0, -2\,\lambda_d(\mathbf{B}_k)\}$ *for some* $\nu > 0$ *and* $r > 0$*. Then, the function* $\phi(\nu, r)$ *is strictly increasing and concave w.r.t.* $\nu$ *and fixed* $r$*. The first- and second-order partial derivatives of* $\phi(\nu, r)$ *w.r.t.* $\nu$ *are*

$$\partial_\nu \phi(\nu, r) = -\frac{\partial_\nu \mathbf{s}(\nu, r)^T\, \mathbf{s}(\nu, r)}{\|\mathbf{s}(\nu, r)\|_2^3} > 0 \tag{157}$$

*and*

$$\partial_\nu^2 \phi(\nu, r) = 3\left\{ \frac{\left(\partial_\nu \mathbf{s}(\nu, r)^T\, \mathbf{s}(\nu, r)\right)^2}{\|\mathbf{s}(\nu, r)\|_2^5} - \frac{\|\partial_\nu \mathbf{s}(\nu, r)\|_2^2\, \|\mathbf{s}(\nu, r)\|_2^2}{\|\mathbf{s}(\nu, r)\|_2^5} \right\} \leq 0, \tag{158}$$

*respectively, with*

$$\partial_\nu \mathbf{s}(\nu, r) = -\frac{r}{2}\, \tilde{\mathbf{H}}_k^{-1}(\nu, r)\, \mathbf{s}(\nu, r). \tag{159}$$

*Proof.* Following similar lines to Conn et al. (2000, Lemma 7.3.1), the first-order partial derivative of $\phi(\nu, r)$ (51) w.r.t. $\nu$ is

$$\partial_\nu \phi(\nu, r) = \partial_\nu \left(\mathbf{s}(\nu, r)^T \mathbf{s}(\nu, r)\right)^{-\frac{1}{2}} - \partial_\nu \underbrace{\frac{1}{\sqrt[3]{\xi}}}_{0}$$

$$= -\frac{1}{2} \|\mathbf{s}(\nu, r)\|_2^{-3} \sum_{\ell=1}^d 2\, \partial_\nu (\mathbf{s}(\nu, r))_\ell\, (\mathbf{s}(\nu, r))_\ell = -\frac{\partial_\nu \mathbf{s}(\nu, r)^T\, \mathbf{s}(\nu, r)}{\|\mathbf{s}(\nu, r)\|_2^3}, \tag{160}$$

which is (157). The second-order partial derivative of $\phi(\nu, r)$ w.r.t. $\nu$ reads

$$\partial_\nu^2 \phi(\nu, r) = -\partial_\nu \left\{ \left( \mathbf{s}(\nu, r)^T \, \mathbf{s}(\nu, r) \right)^{-\frac{3}{2}} \left( \partial_\nu \mathbf{s}(\nu, r)^T \, \mathbf{s}(\nu, r) \right) \right\}$$

$$= \left\{ 3 \frac{\left( \partial_\nu \mathbf{s}(\nu, r)^T \, \mathbf{s}(\nu, r) \right)^2}{\|\mathbf{s}(\nu, r)\|_2^5} - \frac{\partial_\nu^2 \mathbf{s}(\nu, r)^T \, \mathbf{s}(\nu, r) + \|\partial_\nu \mathbf{s}(\nu, r)\|_2^2}{\|\mathbf{s}(\nu, r)\|_2^3} \right\}. \quad (161)$$

The first-order partial derivative of

$$\mathbf{s}(\nu, r) \overset{(18)}{=} -\tilde{\mathbf{H}}_k^{-1}(\nu, r) \, \mathbf{g}_k \quad (162)$$

w.r.t. $\nu$ is

$$\partial_\nu \mathbf{s}(\nu, r) = \frac{r}{2} \, \tilde{\mathbf{H}}_k^{-2}(\nu, r) \, \mathbf{g}_k \overset{(162)}{=} -\frac{r}{2} \, \tilde{\mathbf{H}}_k^{-1}(\nu, r) \, \mathbf{s}(\nu, r), \quad (163)$$

which is (159). From (163) and the assumptions about $\nu$ and $r$, i.e., $\nu \, r > \max\{0, -2\,\lambda_d(\mathbf{B}_k)\}$ with $\nu > 0$ and $r > 0$, we obtain

$$\partial_\nu \mathbf{s}(\nu, r)^T \, \mathbf{s}(\nu, r) = -\frac{r}{2} \, \mathbf{s}(\nu, r)^T \, \tilde{\mathbf{H}}_k^{-1}(\nu, r) \, \mathbf{s}(\nu, r) < 0. \quad (164)$$

Using (164) in (160), we infer that $\phi(\nu, r)$ is strictly increasing w.r.t. $\nu$. The second derivative of (162) w.r.t. $\nu$ is given by

$$\partial_\nu^2 \mathbf{s}(\nu, r) = \frac{r^2}{2} \, \tilde{\mathbf{H}}_k^{-2}(\nu, r) \, \mathbf{s}(\nu, r). \quad (165)$$

From (162) and (165) we get

$$\partial_\nu^2 \mathbf{s}(\nu, r)^T \, \mathbf{s}(\nu, r) = 2 \, \|\partial_\nu \mathbf{s}(\nu, r)\|_2^2. \quad (166)$$

The substitution of (166) in (161) yields

$$\partial_\nu^2 \phi(\nu, r) = 3 \left\{ \frac{\left( \partial_\nu \mathbf{s}(\nu, r)^T \, \mathbf{s}(\nu, r) \right)^2}{\|\mathbf{s}(\nu, r)\|_2^5} - \frac{\|\partial_\nu \mathbf{s}(\nu, r)\|_2^2 \, \|\mathbf{s}(\nu, r)\|_2^2}{\|\mathbf{s}(\nu, r)\|_2^5} \right\}, \quad (167)$$

which is (158). The concavity of $\phi(\nu, r)$, w.r.t. $\nu$, i.e., $\partial_\nu^2 \phi(\nu, r) \leq 0$, follows by applying the Cauchy-Schwartz inequality, i.e., $\left( \partial_\nu \mathbf{s}(\nu, r)^T \, \mathbf{s}(\nu, r) \right)^2 \leq \|\partial_\nu \mathbf{s}(\nu, r)\|_2^2 \, \|\mathbf{s}(\nu, r)\|_2^2$, in (167), which completes the proof. $\qquad \square$

**Lemma 12.** *Let $\phi(\nu, r)$ satisfy Lemma 11. Suppose that for some $\nu > 0$ and $r > 0$ we have $\nu \, r > \max\left\{0, -2\,\lambda_d(\mathbf{B}_k)\right\}$ and $\phi(\nu, r) < 0$. Then for a fixed $r$, the Newton iterates*

$$\nu^+ \leftarrow \nu - \frac{\phi(\nu, r)}{\partial_\nu \phi(\nu, r)}, \quad (168)$$

*will still satisfy $\phi(\nu^+, r) < 0$ and convergence monotonically toward the root $\nu^*$ of $\phi(\nu, r) = 0$ w.r.t. $\nu$. The convergence of the Newton iterations w.r.t. $\nu$ is at least linear and ultimately quadratic.*

*Proof.* Following similar lines to Conn et al. (2000, Lemma 7.3.2), we study the convergence of the Newton iterations in (168) w.r.t. $\nu$ when $r$ is fixed. Suppose that $\phi(\nu, r)$ satisfies Lemma 11, which implies $\partial_\nu \phi(\nu, r) > 0$. Then, from the Newton iteration w.r.t. $\nu$ in (168), we have

$$\phi(\nu, r) + (\nu^+ - \nu) \, \partial_\nu \phi(\nu, r) = 0. \quad (169)$$

According to Lemma 11, $\phi(\nu, r)$ is concave, i.e., $\partial_\nu^2 \phi(\nu, r) \leq 0$. Combining the concavity of $\phi(\nu, r)$ with (169) we get

$$\phi(\nu^+, r) < \phi(\nu, r) + (\nu^+ - \nu) \, \partial_\nu \phi(\nu, r) = 0$$

which proves that $\phi(\nu, r) < 0$ is inherited by all Newton iterations w.r.t. $\nu$. Let $(\nu^*, r)$ be the root of $\phi(\nu, r)$. In addition, let $(\nu^I, r)$ be an intermediate point between points $(\nu, r)$ and $(\nu^*, r)$, i.e., $(\nu^I, r) = \alpha \, (\nu, r) + (1 - \alpha) \, (\nu^*, r)$ with $\alpha \in (0, 1)$. Then, the Taylor expansion about $\nu^*, r$ reads as

$$\phi(\nu, r) = \underbrace{\phi(\nu^*, r)}_{0} + \partial_\nu \phi(\nu^I, r)(\nu - \nu^*) + \frac{1}{2} \, \partial_\nu^2 \phi(\nu^I, r)(\nu - \nu^*)^2 + \mathcal{O}\left((\nu - \nu^*)^3\right). \quad (170)$$

We assume that $(\nu, r)$ is close to $(\nu^*, r)$. This proximity implies that the last term in equation (170) becomes even closer to zero, and consequently, it is omitted from the subsequent analysis. The assumption that $(\nu, r)$ is in proximity to $(\nu^*, r)$ ensures the convergence of the Newton method (Bertsekas, 2017). This proximity is achieved by utilizing the safeguarded Newton Algorithm 2. A comprehensive analysis of the safeguarded Newton methodology can be found in (Conn et al., 2000).

Subtracting $\nu^*$ from both sides of (168), i.e.,

$$\nu^+ - \nu^* = (\nu - \nu^*) - \frac{\phi(\nu, r)}{\partial_\nu \phi(\nu, r)} \tag{171}$$

and substituting (170) in (171), we arrive at

$$\nu^+ - \nu^* = \left(1 - \frac{\partial_\nu \phi(\nu^I, r)}{\partial_\nu \phi(\nu, r)}\right)(\nu - \nu^*) - \frac{1}{2}\frac{\partial_\nu^2 \phi(\nu^I, r)}{\partial_\nu \phi(\nu, r)}(\nu - \nu^*)^2. \tag{172}$$

We examine the following cases:

1. If $\left|1 - \frac{\partial_\nu \phi(\nu^I, r)}{\partial_\nu \phi(\nu, r)}\right| > 1$, (172) diverges.

2. If $\left|1 - \frac{\partial_\nu \phi(\nu^I, r)}{\partial_\nu \phi(\nu, r)}\right| < 1$, we have at least linear convergence in (172).

3. If $\left|1 - \frac{\partial_\nu \phi(\nu^I, r)}{\partial_\nu \phi(\nu, r)}\right| = 0$, we have quadratic convergence as the linear term vanishes in (172).

From the concavity of $\phi(\nu, r)$ w.r.t. $\nu$, we have that $\partial_\nu \phi(\nu, r)$ is decreasing, which implies that $\left|1 - \frac{\partial_\nu \phi(\nu^I, r)}{\partial_\nu \phi(\nu, r)}\right| < 1$. Thus, the Newton iterations w.r.t. $\nu$ convergence in (168) is at least linear and ultimately quadratic, which completes the proof. $\qquad\square$

**Remark 3.** *Lemma 11 implies that $\partial_\nu \phi(\nu, r) > 0$. Suppose that for some $\nu$ and $r$ we have $\phi(\nu, r) < 0$, i.e., Lemma 12 holds. Then, by (168) we have that $\nu^+ > \nu$. Given $\nu^+ > \nu$ and the initial values of $r$ and $\nu$ in lines 2, 4, and 6 of Algorithm 2, the optimal $r$ and $\nu$ always satisfy $\mathbf{B}_k + \frac{\nu r}{2}\mathbf{I} \succ \mathbf{0}$.*

**Lemma 13.** *For some $\nu > 0$ and $r > 0$, suppose $\nu r > \max\{0, -2\lambda_d(\mathbf{B}_k)\}$ and*

$$\left|\|\mathbf{s}(\nu, r)\|_2 - \xi^{\frac{1}{3}}\right| \le \kappa_{easy}\,\xi^{\frac{1}{3}} \tag{173}$$

*with $\kappa_{easy} \in (0, 1)$. Then*

$$\hat{\mathrm{m}}(\mathbf{s}(\nu, r)) \le (1 - \kappa_{easy})^2\,\hat{\mathrm{m}}(\mathbf{s}_k^*), \tag{174}$$

*where $\mathbf{s}_k^*$ is the minimizer of (7) that satisfies Lemma 1, and*

$$\hat{\mathrm{m}}(\mathbf{s}) \overset{\triangle}{=} F(\mathbf{x}_k) + \mathbf{g}_k^T \mathbf{s} + \frac{1}{2}\mathbf{s}^T \mathbf{B}_k \mathbf{s}. \tag{175}$$

*Proof.* A similar proof to that in Conn et al. (2000, Lemma 7.3.5) can be devised for the constraint $\|\mathbf{s}\|_2^3 \le \xi$. $\qquad\square$

## B.11 Preliminaries for Theorem 3

Here, Nesterov & Polyak (2006, Lemma 1), Nesterov & Polyak (2006, Lemma 2), Nesterov & Polyak (2006, Lemma 3), Nesterov & Polyak (2006, Lemma 4), and Nesterov & Polyak (2006, Lemma 5), correspond to Lemmata 10, 14, 15, 16 and 18, respectively, proven for $\mathrm{Diag}(\nabla^2 f(\mathbf{x}))$ in place of $\nabla^2 f(\mathbf{x})$. In addition, Lemma 17 is proven providing details not included in (Nesterov & Polyak, 2006, Section 2).

Let the level set

$$\mathfrak{L}(c) = \{\mathbf{x} \in \mathbb{R}^d \colon f(\mathbf{x}) \leq c\}, \tag{176}$$

and assume that $\mathcal{F} \subseteq \mathfrak{L}(f(\mathbf{x}_0))$. Let

$$\hat{f}(\mathbf{x}, \mathbf{y}) = f(\mathbf{x}) + \nabla f(\mathbf{x})^T(\mathbf{y} - \mathbf{x}) + \frac{1}{2}(\mathbf{y} - \mathbf{x})^T \operatorname{Diag}(\nabla^2 f(\mathbf{x}))(\mathbf{y} - \mathbf{x}) + \frac{M}{6} \|\mathbf{y} - \mathbf{x}\|_2^3, \tag{177}$$

$$T_M(\mathbf{x}) = \arg\min_{\mathbf{y}} \hat{f}(\mathbf{x}, \mathbf{y}), \tag{178}$$

and

$$\bar{f}_M(\mathbf{x}) = \min_{\mathbf{y}} \hat{f}(\mathbf{x}, \mathbf{y}). \tag{179}$$

That is,

$$\bar{f}_M(\mathbf{x}) = \hat{f}(\mathbf{x}, T_M(\mathbf{x})). \tag{180}$$

To compute $T_M(\mathbf{x})$ in (178), we solve $\nabla_{\mathbf{y}} \hat{f}(\mathbf{x}, \mathbf{y}) = \mathbf{0}$, i.e.,

$$\nabla f(\mathbf{x}) + \operatorname{Diag}(\nabla^2 f(\mathbf{x}))(\mathbf{y} - \mathbf{x}) + \frac{M}{2} \|\mathbf{y} - \mathbf{x}\|_2 (\mathbf{y} - \mathbf{x}) = \mathbf{0}. \tag{181}$$

Let $r_M(\mathbf{x}) = \|\mathbf{x} - T_M(\mathbf{x})\|_2$. For $\mathbf{y} = T_M(\mathbf{x})$ in (181), if we multiply both sides of (181) by $T_M(\mathbf{x}) - \mathbf{x}$ we arrive at

$$\nabla f(\mathbf{x})^T(T_M(\mathbf{x}) - \mathbf{x}) + (T_M(\mathbf{x}) - \mathbf{x})^T \operatorname{Diag}(\nabla^2 f(\mathbf{x}))(T_M(\mathbf{x}) - \mathbf{x}) + \frac{M}{2} \|(T_M(\mathbf{x}) - \mathbf{x})\|_2^3 = 0. \tag{182}$$

**Lemma 14.** *For any* $\mathbf{x} \in \mathcal{F}$ *with* $f(\mathbf{x}) \leq f(\mathbf{x}_0)$, *we have*

$$\nabla f(\mathbf{x})^T(\mathbf{x} - T_M(\mathbf{x})) \geq 0. \tag{183}$$

*Moreover, if* $M \geq \frac{2}{3} L_H$ *and* $\mathbf{x} \in \operatorname{int} \mathcal{F}$, *then*

$$T_M(\mathbf{x}) \in \mathfrak{L}(f(\mathbf{x})). \tag{184}$$

*Proof.* Using Corollary 3, we obtain

$$\operatorname{Diag}(\nabla^2 f(\mathbf{x})) + \frac{M}{2} \|\mathbf{x} - \mathbf{y}\|_2 \ \mathbf{I} \succeq 0, \tag{185}$$

which when pre-multiplied by $(T_M(\mathbf{x}) - \mathbf{x})^T$ and post-multiplied by $(T_M(\mathbf{x}) - \mathbf{x})$ yields

$$(T_M(\mathbf{x}) - \mathbf{x})^T \operatorname{Diag}(\nabla^2 f(\mathbf{x}))(T_M(\mathbf{x}) - \mathbf{x}) + \frac{M}{2} \|T_M(\mathbf{x}) - \mathbf{x}\|_2^3 \geq 0. \tag{186}$$

Then, combining (182) with (186) we arrive at (183).

**Assumption and Contradiction.** We now show that $T_M(\mathbf{x}) \in \mathfrak{L}(f(\mathbf{x}))$. Following the approach of Nesterov & Polyak (2006, Lemma 2), we proceed by contradiction by assuming $T_M(\mathbf{x}) \notin \mathfrak{L}(f(\mathbf{x}))$ for $M \geq \frac{2}{3} L_H$. We then show that this assumption cannot hold, leading to a contradiction. Thus, we conclude that $T_M(\mathbf{x}) \in \mathfrak{L}(f(\mathbf{x}))$.

By assuming $T_M(\mathbf{x}) \notin \mathfrak{L}(f(\mathbf{x}))$, there exist

$$\mathbf{y}_\alpha = (1 - \alpha)\,\mathbf{x} + \alpha\,T_M(\mathbf{x}), \tag{187}$$

with $\alpha \in [0, 1]$, such that

$$f(\mathbf{y}_\alpha) > f(\mathbf{x}). \tag{188}$$

Using the upper bound of (155) with $\mathbf{y} = \mathbf{y}_\alpha$ we obtain

$$f(\mathbf{y}_\alpha) \leq f(\mathbf{x}) + \nabla f(\mathbf{x})^T (\mathbf{y}_\alpha - \mathbf{x}) + \frac{1}{2}(\mathbf{y}_\alpha - \mathbf{x})^T \operatorname{Diag}(\nabla^2 f(\mathbf{x}))(\mathbf{y}_\alpha - \mathbf{x}) + \frac{L_H}{6} \|\mathbf{y}_\alpha - \mathbf{x}\|_2^3 \stackrel{(187)}{=}$$

$$f(\mathbf{x}) + \alpha \nabla f(\mathbf{x})^T (T_M(\mathbf{x}) - \mathbf{x}) + \frac{\alpha^2}{2}(T_M(\mathbf{x}) - \mathbf{x})^T \operatorname{Diag}(\nabla^2 f(\mathbf{x}))(T_M(\mathbf{x}) - \mathbf{x}) + \frac{\alpha^3 L_H}{6} \|T_M(\mathbf{x}) - \mathbf{x}\|_2^3, \tag{189}$$

which using (182) implies

$$f(\mathbf{y}_\alpha) - f(\mathbf{x}) \leq \underbrace{-\left(\alpha - \frac{\alpha^2}{2}\right) \nabla f(\mathbf{x})^T (\mathbf{x} - T_M(\mathbf{x}))}_{\geq 0 \text{ as } \alpha \in [0,1] \text{ and } (195)} - \frac{\alpha^2}{2}\left(\frac{M}{2} - \frac{\alpha L_H}{3}\right) \|T_M(\mathbf{x}) - \mathbf{x}\|_2^3. \tag{190}$$

For $\alpha \leq 1$, we have

$$\frac{M}{2} - \frac{\alpha L_H}{3} \geq \frac{M}{2} - \frac{L_H}{3}, \tag{191}$$

which by using our assumption $M \geq \frac{2}{3}L_H$ it is implied that $f(\mathbf{y}_\alpha) \leq f(\mathbf{x})$ in (190). However, $f(\mathbf{y}_\alpha) \leq f(\mathbf{x})$ contradicts (188) for $M \geq \frac{2}{3}L_H$, which in turn leads to (184), and the proof is complete. □

**Lemma 15.** *If $T_M(\mathbf{x}) \in \mathcal{F}$ then*

$$\|\nabla f(T_M(\mathbf{x}))\|_2 \leq \frac{L_H + M}{2} r_M^2(\mathbf{x}). \tag{192}$$

*Proof.* Setting $\mathbf{y} = T_M(\mathbf{x})$ in (154) and (181) we get

$$\left\|\nabla f(T_M(\mathbf{x})) - \nabla f(\mathbf{x}) - \operatorname{Diag}(\nabla^2 f(\mathbf{x}))(T_M(\mathbf{x}) - \mathbf{x})\right\|_2 \leq \frac{L_H}{2} \|T_M(\mathbf{x}) - \mathbf{x}\|_2 \tag{193}$$

and

$$\left\|\nabla f(\mathbf{x}) + \operatorname{Diag}(\nabla^2 f(\mathbf{x}))(T_M(\mathbf{x}) - \mathbf{x})\right\|_2 = \frac{M}{2} \|T_M(\mathbf{x}) - \mathbf{x}\|_2^2 = \frac{M}{2} r_M^2(\mathbf{x}), \tag{194}$$

respectively. Then, combining (193) with (194) and the definition of the reverse triangle inequality, we arrive at (192) and the proof is complete. □

**Lemma 16.** *For any $\mathbf{x} \in \mathcal{F}$ we have*

$$\bar{f}_M(\mathbf{x}) \leq \min_{\mathbf{y}} \left(\frac{M + L_H}{6} \|\mathbf{y} - \mathbf{x}\|_2^3 + f(\mathbf{y})\right) \tag{195}$$

*and*

$$f(\mathbf{x}) - \bar{f}_M(\mathbf{x}) \geq \frac{M}{12} r_M^3(\mathbf{x}). \tag{196}$$

*Moreover, if $M \geq L_H$, then $T_M(\mathbf{x}) \in \mathcal{F}$ and*

$$f(T_M(\mathbf{x})) \leq \bar{f}_M(\mathbf{x}). \tag{197}$$

*Proof.* In the following, we have used the relaxed condition $M \geq L_H$ as $M \geq L_H > \frac{2}{3}L_H$. Note that the relaxed condition $M \geq L_H$ also satisfies Lemma 14. From the lower and upper bound of (155) we have

$$\hat{f}(\mathbf{x}, \mathbf{y}) \leq \frac{M + L_H}{6} \|\mathbf{y} - \mathbf{x}\|_2^3 + f(\mathbf{y}) \tag{198}$$

and

$$f(\mathbf{y}) \leq \hat{f}(\mathbf{x}, \mathbf{y}), \tag{199}$$

respectively. Thus, we have

$$f(\mathbf{y}) \leq \hat{f}(\mathbf{x}, \mathbf{y}) \leq \frac{M + L_H}{6} \|\mathbf{y} - \mathbf{x}\|_2^3 + f(\mathbf{y}). \tag{200}$$

Minimizing (200) all sides w.r.t. $\mathbf{y}$ yields

$$\min_{\mathbf{y}} f(\mathbf{y}) \leq \min_{\mathbf{y}} \hat{f}(\mathbf{x}, \mathbf{y}) \leq \min_{\mathbf{y}} \left( \frac{M + L_H}{6} \|\mathbf{y} - \mathbf{x}\|_2^3 + f(\mathbf{y}) \right). \tag{201}$$

which in turn using $\mathbf{y} = T_M(\mathbf{x})$ yields

$$f(T_M(\mathbf{x})) \leq \hat{f}(\mathbf{x}, T_M(\mathbf{x})) \leq \min_{\mathbf{y}} \left( \frac{M + L_H}{6} \|\mathbf{y} - \mathbf{x}\|_2^3 + f(\mathbf{y}) \right). \tag{202}$$

Additionally, using (180) in (202) we obtain (195). Note that (195) aligns with the results presented in Nesterov & Polyak (2006, Lemma 4), with the distinction that $\mathrm{Diag}(\nabla^2 f(\mathbf{x}))$ is used in place of $\nabla^2 f(\mathbf{x})$. From the LHS of (202) and (177) we obtain

$$f(\mathbf{x}) - f(T_M(\mathbf{x})) \geq f(\mathbf{x}) - \hat{f}(\mathbf{x}, T_M(\mathbf{x}))$$
$$= -\nabla f(\mathbf{x})^T (T_M(\mathbf{x}) - \mathbf{x}) - \frac{1}{2}(T_M(\mathbf{x}) - \mathbf{x})^T \mathrm{Diag}(\nabla^2 f(\mathbf{x}))(T_M(\mathbf{x}) - \mathbf{x}) - \frac{M}{6} \|T_M(\mathbf{x}) - \mathbf{x}\|_2^3. \tag{203}$$

In addition, from (182) we have

$$-\frac{1}{2}(T_M(\mathbf{x}) - \mathbf{x})^T \mathrm{Diag}(\nabla^2 f(\mathbf{x}))(T_M(\mathbf{x}) - \mathbf{x}) = \frac{1}{2}\nabla f(\mathbf{x})^T (T_M(\mathbf{x}) - \mathbf{x}) + \frac{M}{4} \|T_M(\mathbf{x}) - \mathbf{x}\|_2^3, \tag{204}$$

which combined with (203) and (180) gives

$$f(\mathbf{x}) - \bar{f}_M(\mathbf{x}) \geq -\frac{1}{2}\nabla f(\mathbf{x})^T (T_M(\mathbf{x}) - \mathbf{x}) + \frac{M}{12}r_M^3(\mathbf{x}), \tag{205}$$

which in turn combined with (183) yields (196). To conclude the proof, setting $\mathbf{y} = T_M(\mathbf{x})$ in the LHS of (200) and using (180) we obtain (197). $\qquad\square$

**Lemma 17.** *If $\mathbf{x} \in \mathcal{F}$ then*

$$\mu_{M_i}(\mathbf{x}_{i+1}) \stackrel{\Delta}{=} \max\left\{ \sqrt{\frac{2}{L_H + M_i}} \|\nabla f(\mathbf{x}_{i+1})\|_2, -\frac{2}{2L_H + M_i}\lambda_{min}(\mathrm{Diag}(\nabla^2 f(\mathbf{x}_{i+1}))) \right\}. \tag{206}$$

*Proof.* From (192) we have

$$\sqrt{\frac{2}{L_H + M}} \|\nabla f(\mathbf{x}_{i+1})\|_2 \leq \|\mathbf{x}_{i+1} - \mathbf{x}_i\|_2. \tag{207}$$

(185) implies

$$\mathrm{Diag}(\nabla^2 f(\mathbf{x})) - L_H \|\mathbf{x} - \mathbf{y}\|_2 \, \mathbf{I} + \frac{M}{2} \|\mathbf{x} - \mathbf{y}\|_2 \, \mathbf{I} \succeq -L_H \|\mathbf{x} - \mathbf{y}\|_2 \, \mathbf{I} \Leftrightarrow$$
$$\lambda_{\min}\left(\mathrm{Diag}(\nabla^2 f(\mathbf{x}))\right) - L_H \|\mathbf{x} - \mathbf{y}\|_2 \geq -\left(\frac{M}{2} + L_H\right) \|\mathbf{x} - \mathbf{y}\|_2, \tag{208}$$

which by setting $\mathbf{y} = \mathbf{x}_{t+1}$ and $\mathbf{x} = \mathbf{x}_t$ yields

$$\lambda_{\min}\left(\mathrm{Diag}(\nabla^2 f(\mathbf{x}_t))\right) - L_H \|\mathbf{x}_t - \mathbf{x}_{t+1}\|_2^2 \geq -\left(\frac{M}{2} + L_H\right) \|\mathbf{x}_t - \mathbf{x}_{t+1}\|_2. \tag{209}$$

Combining Nesterov (2018, Corrolary 1.2.3) with Lemma 9 for $\mathbf{y} = \mathbf{x}_{t+1}$ and $\mathbf{x} = \mathbf{x}_t$ yields

$$\lambda_{\min}\left(\mathrm{Diag}(\nabla^2 f(\mathbf{x}_{t+1}))\right) \geq \lambda_{\min}\left(\mathrm{Diag}(\nabla^2 f(\mathbf{x}_t))\right) - L_H \|\mathbf{x}_t - \mathbf{x}_{t+1}\|_2, \tag{210}$$

which when combined with (209) gives

$$\lambda_{\min}\left(\mathrm{Diag}(\nabla^2 f(\mathbf{x}_{t+1}))\right) \geq -\left(\frac{M}{2} + L_H\right) \|\mathbf{x}_t - \mathbf{x}_{t+1}\|_2, \tag{211}$$

and, in turn, implies

$$-\frac{2}{2L_H + M}\lambda_{\min}\left(\mathrm{Diag}(\nabla^2 f(\mathbf{x}_{t+1}))\right) \leq \|\mathbf{x}_t - \mathbf{x}_{t+1}\|_2,\tag{212}$$

In order to obtain an $(\epsilon_g, \epsilon_H)$-stationary point, we need $\|\mathbf{x}_t - \mathbf{x}_{t+1}\|_2 \leq \epsilon_g$ and $\|\mathbf{x}_t - \mathbf{x}_{t+1}\|_2 \leq \epsilon_H$ in (207) and (212), respectively, which yields (206), and the proof is complete. $\qquad\square$

**Lemma 18.** *For any* $\mathbf{x} \in \mathcal{F}$ *we have*

$$\mu_M(T_M(\mathbf{x})) \leq r_M(\mathbf{x}).\tag{213}$$

*Proof.* Adapting Nesterov (2018, Corollary 1.2.2) in our context yields

$$\mathrm{Diag}(\nabla^2 f(T_M(\mathbf{x}))) \succeq \mathrm{Diag}(\nabla^2 f(\mathbf{x})) - r_M(\mathbf{x})L_H\mathbf{I}.\tag{214}$$

Combining (214) and (185) gives

$$\mathrm{Diag}(\nabla^2 f(T_M(\mathbf{x}))) \succeq -\left(\frac{1}{2}M + L_H\right)r_M(\mathbf{x})\mathbf{I},\tag{215}$$

which when combined with Lemma 17 yields (213) and the proof is complete. $\qquad\square$

## B.12   Proof of Theorem 3

Let $(\mathbf{s}_{i+1}, \nu_{i+1})$ be the output of Algorithm 2 for $\mathbf{B} = \mathrm{Diag}(\nabla^2 f(\mathbf{x}_i))$, $\mathbf{g} = \nabla f(\mathbf{x}_i)$. Recall that $(\mathbf{s}_{i+1}, \nu_{i+1})$ is a minimizer of (7) and according to Theorem 2 it is also a minimizer of problem (5) where $M = \nu_{i+1}$. Recall also that $\mathbf{x}_{i+1} = \mathbf{x}_i + \mathbf{s}_{i+1}$ and let the sequence $\{\mathbf{x}_i\}_{i\geq 1}$ be generated by Algorithm 1.

Next, suppose that Assumption 1 holds, i.e., the objective function $f(\mathbf{x})$ is bounded from bellow, $f(\mathbf{x}) \geq f^{\mathrm{low}}$ for all $\mathbf{x} \in \mathcal{F}$. Then, we continue with the proof of the main result in Theorem 3. From (196), we have

$$\begin{aligned}
f(\mathbf{x}_0) - \bar{f}_{M_0}(\mathbf{x}_0) &\geq \frac{M_0}{12}r_{M_0}^3(\mathbf{x}_0)\\
f(\mathbf{x}_1) - \bar{f}_{M_1}(\mathbf{x}_1) &\geq \frac{M_1}{12}r_{M_1}^3(\mathbf{x}_1)\\
&\vdots\\
f(\mathbf{x}_{k-1}) - \bar{f}_{M_{k-1}}(\mathbf{x}_{k-1}) &\geq \frac{M_{k-1}}{12}r_{M_{k-1}}^3(\mathbf{x}_{k-1}),
\end{aligned}\tag{216}$$

where $r_{M_i}(\mathbf{x}_i) = \|\mathbf{x}_i - \mathbf{x}_{i+1}\|_2$ and

$$\bar{f}_{M_i}(\mathbf{x}) \triangleq \min_{\mathbf{s}\in\mathbb{R}^d} m_{M_i}(\mathbf{s}).\tag{217}$$

Summing over (216) we get

$$\sum_{i=0}^{k-1}\left(f(\mathbf{x}_i) - \bar{f}_{M_i}(\mathbf{x}_i)\right) \geq \sum_{i=0}^{k-1}\frac{M_i}{12}r_{M_i}^3(\mathbf{x}_i).\tag{218}$$

Then applying $f(\mathbf{x}_{i+1}) \leq \bar{f}_{M_i}(\mathbf{x}_i)$ (Lemma 16), we get

$$\sum_{i=0}^{k-1}\left(f(\mathbf{x}_i) - f(\mathbf{x}_{i+1})\right) \geq \sum_{i=0}^{k-1}\frac{M_i}{12}r_{M_i}^3(\mathbf{x}_i),\tag{219}$$

which, by applying the telescoping sum, yields

$$f(\mathbf{x}_0) - f(\mathbf{x}_k) \geq \sum_{i=0}^{k-1}\frac{M_i}{12}r_{M_i}^3(\mathbf{x}_i).\tag{220}$$

Next, using the lower bound of $f(\mathbf{x}_k)$, i.e., $f^{\text{low}}$ by Assumption 1 and Remark 1, we obtain

$$f(\mathbf{x}_0) - f^{\text{low}} \geq \sum_{i=0}^{k-1} \frac{M_i}{12} r_{M_i}^3(\mathbf{x}_i), \tag{221}$$

which implies

$$f(\mathbf{x}_0) - f^{\text{low}} \geq k\frac{L_0}{12} r_{L_0}^3(\mathbf{x}_i) \Leftrightarrow \mu_{M_i}(\mathbf{x}_{i+1}) \leq r_{M_i}(\mathbf{x}_i) \leq 12^{1/3} \left(\frac{f(\mathbf{x}_0) - f^{\text{low}}}{k\,M_i}\right)^{1/3}, \tag{222}$$

where Lemma 18 (Appendix B.11) is applied to get $\mu_{M_i}(\mathbf{x}_{i+1}) \leq r_{M_i}(\mathbf{x}_i)$. For $L_H = M_i$ and applying the trick $12 = 3 \cdot 4 \Leftrightarrow 12^{1/3} = 3^{1/3} \cdot (8/2)^{1/3} \Leftrightarrow 12^{1/3} = (3/2)^{1/3} \cdot 8^{1/3} \Leftrightarrow 12^{1/3} = (3/2)^{1/3} \cdot 2 \Leftrightarrow 12^{1/3} = (3/2)^{1/3} \cdot 8/4 \leq (3/2)^{1/3} \cdot 8/3$, we arrive at

$$\mu_{L_H}(\mathbf{x}_{i+1}) \leq \frac{8}{3}\left(\frac{3}{2}\frac{f(\mathbf{x}_0) - f^{\text{low}}}{L_H k}\right)^{1/3}. \tag{223}$$

As in Nesterov & Polyak (2006, Theorem 3), here it is assumed that $\nabla^2 f(\mathbf{x}_i)$ is positive definite for some $i \geq 0$. The latter assumption implies that $\text{Diag}(\nabla^2 f(\mathbf{x}_i))$ is also positive definite. Then for some $i \geq 0$, from (206), we restrict our study to

$$\mu_{L_H}(\mathbf{x}_{i+1}) = \sqrt{\frac{1}{L_H}} \|\nabla f(\mathbf{x}_{i+1})\|_2, \tag{224}$$

which combined with (223), yields

$$\min_{0 \leq i \leq k-1} \|\nabla f(\mathbf{x}_{i+1})\|_2 \leq L_H^{1/3}\left(\frac{8}{3}\right)^2 \left(\frac{3}{2}\frac{f(\mathbf{x}_0) - f^{\text{low}}}{k}\right)^{2/3}, \tag{225}$$

which implies (23). The convergence rate in (23) is used to establish the local convergence rate when the approximate $\mathbf{g}_i$ and $\mathbf{B}_i$ are used instead of $\nabla f(\mathbf{x}_i)$ and $\nabla^2 f(\mathbf{x}_i)$, respectively. The latter argument is strengthened by Corollary 2, and the proof is complete.

### B.13 Vector and Matrix Bernstein Inequalities

For completeness, we restate Kohler & Lucchi (2017, Lemma 18), incorporating corrections for minor typographical errors. Lemma 19 is utilized by Lemma 5. Next, Lemma 20 is introduced and utilized by Lemma 22, which in turn is utilized by Lemma 6. Lemma 21 is also used by Lemma 22.

**Lemma 19** (Vector Bernstein Inequality). *Let $\mathbf{x}_1, \mathbf{x}_2, \ldots \mathbf{x}_n$ be independent random vectors of common dimension $d$ and assume that each one is centered, uniformly bounded, and also the variance is bounded from above, i.e.,*

$$\mathbb{E}[\mathbf{x}_i] = \mathbf{0} \text{ and } \|\mathbf{x}_i\|_2 \leq \vartheta \text{ as well as } \mathbb{E}[\|\mathbf{x}_i\|_2^2] \leq \sigma^2. \tag{226}$$

*Let $\mathbf{z} = \frac{1}{n}\sum_{i=1}^n \mathbf{x}_i$. Then we have*

$$\Pr(\|\mathbf{z}\|_2 \geq \epsilon) \leq \exp\left(-n\frac{\epsilon^2}{8\sigma^2} + \frac{1}{4}\right), \tag{227}$$

*with $0 < \epsilon < \sigma^2/\vartheta + \sigma$.*

*Proof.* A proof can be found in (Kohler & Lucchi, 2017, Lemma 18). However, some typographical errors were identified, leading us to reproduce the proof for clarity.

The Vector Bernstein inequality for independent, zero-mean random vectors Gross (2011, Theorem 12) states

$$\Pr\left(\frac{1}{n}\left\|\sum_{i=1}^n \mathbf{x}_i\right\|_2 \geq \frac{1}{n}(t + \sqrt{V})\right) \leq \exp\left(-\frac{t^2}{4V}\right), \tag{228}$$

where $V = \sum_{i=1}^{n} \mathbb{E}[\|\mathbf{x}_i\|_2^2]$ is the sum of the traces of the covariance matrices of the centered vectors $\mathbf{x}_i$. Using $\mathbb{E}[\|\mathbf{x}_i\|_2^2] \leq \sigma^2$ yields $V \leq n\sigma^2$.

Note that in (228), the probability condition is scaled by a factor of $1/n$ to align with the subsequent analysis involving $\mathbf{z}$. Let $\epsilon = (t + \sqrt{V})/n \Leftrightarrow t = n\epsilon - \sqrt{V}$. Using (228) we get

$$\Pr\left(\|\mathbf{z}\|_2 \geq \epsilon\right) \leq \exp\left(-\frac{1}{4}\left(\frac{n\epsilon - \sqrt{V}}{\sqrt{V}}\right)^2\right) = \exp\left(-\frac{1}{4}\left(\frac{n\epsilon}{\sqrt{V}} - 1\right)^2\right). \tag{229}$$

We claim that

$$-\frac{1}{4}\left(\frac{n\epsilon}{\sqrt{V}} - 1\right)^2 \leq -\frac{1}{4}\frac{n^2\epsilon^2}{2V} + \frac{1}{4} \tag{230}$$

Indeed, if (230) holds we arrive at a valid inequality

$$-\frac{n^2\epsilon^2}{V} + 2\frac{n\epsilon}{\sqrt{V}} - 1 \leq -\frac{1}{2}\frac{n^2\epsilon^2}{V} + 1$$
$$\Leftrightarrow \left(\frac{n\epsilon}{\sqrt{2V}} - \sqrt{2}\right)^2 \geq 0. \tag{231}$$

Using (230) in (229) gives

$$\Pr\left(\|\mathbf{z}\|_2 \geq \epsilon\right) \leq \exp\left(-n\frac{\epsilon^2}{8\sigma^2} + \frac{1}{4}\right), \tag{232}$$

where $V \leq n\sigma^2$ is used. According to Gross (2011, Theorem 12), $t < V/\max_i \|\mathbf{x}_i\|_2$. For $V \leq n\sigma^2$ and $\|\mathbf{x}_i\|_2 \leq \vartheta$ gives $t < n\sigma^2/\vartheta$. Given $V \leq n\sigma^2$ and $t < n\sigma^2/\vartheta$, we arrive at

$$n\epsilon = t + \sqrt{V} \leq \frac{n\sigma^2}{\vartheta} + \sqrt{n}\sigma \Leftrightarrow \epsilon \leq \frac{\sigma^2}{\vartheta} + \sigma, \tag{233}$$

where $\sqrt{x} < x$ with $x > 1$ is used. In addition, it can be shown that $\mathrm{Var}(\mathbf{z}) \leq \sigma^2/n$ Gross (2011, Theorem 12) establishing (227), which concluded the proof. $\qquad\square$

**Lemma 20.** *Let $\mathbf{u}_i : \Omega_{\mathbf{u}_i} \to \mathbb{R}^n$ and $\mathbf{v}_j : \Omega_{\mathbf{v}_j} \to \mathbb{R}^m$ be independent random vectors for each $i, j$. Let $g : \mathbb{R}^n \times \mathbb{R}^m \to \mathbb{R}^{d \times d}$ be a function that produces random matrices. Then, for any indices $(i, j) \neq (k, l)$, the matrices $g(\mathbf{u}_i, \mathbf{v}_j)$ and $g(\mathbf{u}_k, \mathbf{v}_l)$ are independent, regardless of whether $\mathbf{u}_i$ and $\mathbf{v}_j$ come from the same or different distributions.*

*Proof.* Let two threshold matrices $\mathbf{M}$ and $\mathbf{M}'$ (which are symmetric $d \times d$ matrices), and consider the probability

$$P(g(\mathbf{u}_i, \mathbf{v}_j) \preceq \mathbf{M} \cap g(\mathbf{u}_k, \mathbf{v}_l) \preceq \mathbf{M}'), \tag{234}$$

using the element-wise comparison operator $\preceq$. Given that $\mathbf{u}_i : \Omega_{\mathbf{u}_i} \to \mathbb{R}^n$ and $\mathbf{v}_j : \Omega_{\mathbf{v}_j} \to \mathbb{R}^m$ are independent random vectors, and $g : \mathbb{R}^n \times \mathbb{R}^m \to \mathbb{R}^{d \times d}$ is a function generating random matrices, we rewrite the event as

$$\{g(\mathbf{u}_i, \mathbf{v}_j) \preceq \mathbf{M}\} \equiv \{(\omega_{\mathbf{u}_i}, \omega_{\mathbf{v}_j}) \in \Omega_{\mathbf{u}_i} \times \Omega_{\mathbf{v}_j} : g(\mathbf{u}_i(\omega_{\mathbf{u}_i}), \mathbf{v}_j(\omega_{\mathbf{v}_j})) \preceq \mathbf{M}\}$$
$$\equiv \{(\mathbf{u}_i, \mathbf{v}_j) \in \mathbb{R}^n \times \mathbb{R}^m : g(\mathbf{u}_i, \mathbf{v}_j) \preceq \mathbf{M}\}. \tag{235}$$

Similarly, we have

$$\{g(\mathbf{u}_k, \mathbf{v}_l) \preceq \mathbf{M}'\} \equiv \{(\omega_{\mathbf{u}_k}, \omega_{\mathbf{v}_l}) \in \Omega_{\mathbf{u}_k} \times \Omega_{\mathbf{v}_l} : g(\mathbf{u}_k(\omega_{\mathbf{u}_k}), \mathbf{v}_l(\omega_{\mathbf{v}_l})) \preceq \mathbf{M}'\}$$
$$\equiv \{(\mathbf{u}_k, \mathbf{v}_l) \in \mathbb{R}^n \times \mathbb{R}^m : g(\mathbf{u}_k, \mathbf{v}_l) \preceq \mathbf{M}'\}. \tag{236}$$

Let

$$A = \{(\mathbf{u}_i, \mathbf{v}_j) \in \mathbb{R}^n \times \mathbb{R}^m : g(\mathbf{u}_i, \mathbf{v}_j) \preceq \mathbf{M}\} \tag{237}$$

and

$$B = \{(\mathbf{u}_k, \mathbf{v}_l) \in \mathbb{R}^n \times \mathbb{R}^m : g(\mathbf{u}_k, \mathbf{v}_l) \preceq \mathbf{M}'\}. \tag{238}$$

Then using (235) and (236) in (234), we write

$$P(\{g(\mathbf{u}_i, \mathbf{v}_j) \preceq \mathbf{M}\} \cap \{g(\mathbf{u}_k, \mathbf{v}_l) \preceq \mathbf{M}'\}) = P((\mathbf{u}_i, \mathbf{v}_j) \in A \cap (\mathbf{u}_k, \mathbf{v}_l) \in B). \tag{239}$$

Recall that the sequences $\{\mathbf{u}_i\}$ and $\{\mathbf{v}_j\}$ are independent families of random variables which implies that the pairs $(\mathbf{u}_i, \mathbf{v}_j)$ are formed by drawing independently from these families. Thus, since $\mathbf{u}_i$ and $\mathbf{v}_j$ are independent for each $(i, j)$, and $(\mathbf{u}_k, \mathbf{v}_l)$ are also independent, we have

$$P(\{g(\mathbf{u}_i, \mathbf{v}_j) \preceq \mathbf{M}\} \cap \{g(\mathbf{u}_k, \mathbf{v}_l) \preceq \mathbf{M}'\}) = P((\mathbf{u}_i, \mathbf{v}_j) \in A)P((\mathbf{u}_k, \mathbf{v}_l) \in B), \tag{240}$$

which implies

$$P(\{g(\mathbf{u}_i, \mathbf{v}_j) \preceq \mathbf{M}\} \cap \{g(\mathbf{u}_k, \mathbf{v}_l) \preceq \mathbf{M}'\}) = P(g(\mathbf{u}_i, \mathbf{v}_j) \preceq \mathbf{M})P(g(\mathbf{u}_k, \mathbf{v}_l) \preceq \mathbf{M}'), \tag{241}$$

where (235) and (236) are used. Since the joint probability factorizes, this proves that $g(\mathbf{u}_i, \mathbf{v}_j)$ and $g(\mathbf{u}_k, \mathbf{v}_l)$ are independent whenever $(i, j) \neq (k, l)$, regardless of whether $\mathbf{u}_i$ and $\mathbf{v}_j$ come from the same or different distributions. □

**Lemma 21.** *Let $\mathbf{X} \in \mathbb{R}^{d \times d}$ be a symmetric mean-zero matrix with $\|\mathbf{X}\| \leq 1$ almost surely. Then,*

$$\mathbb{E}\left[\exp(\lambda \mathbf{X})\right] \preceq \exp\left(g(\lambda)\mathbb{E}[\mathbf{X}^2]\right), \tag{242}$$

*where $g(\lambda) = e^\lambda - \lambda - 1$.*

*Proof.* We refer the reader to (Vershynin, 2018). □

**Lemma 22** (Matrix Bernstein Inequality). *Let $\mathbf{X}_{ij} \overset{\Delta}{=} g(\mathbf{u}_i, \mathbf{v}_j)$ be $d \times d$ zero-mean random matrices with two independent sources of randomness, $\mathbf{u}_i$ and $\mathbf{v}_j$. Also, let $\{\mathbf{X}_{ij}\}_{i,j=1}^{N,M}$ be a set of independent random matrices of common dimension $d \times d$, such that $\|\mathbf{X}_{ij}\|_2 \leq K$ almost surely for all $i, j$. Then, for every $t \geq 0$, we have*

$$\Pr\left(\left\|\sum_{i=1}^{N}\sum_{j=1}^{M}\mathbf{X}_{ij}\right\|_2 \geq t\right) \leq 2d \exp\left(-\frac{t^2/2}{\sigma^2 + Kt/3}\right). \tag{243}$$

*Here, the matrix variance is given by*

$$\sigma^2 = \left\|\sum_{i=1}^{N}\sum_{j=1}^{M}\mathbb{E}\left[\mathbf{X}_{ij}^2\right]\right\|_2. \tag{244}$$

*In particular, we can express this bound as a mixture of sub-Gaussian and sub-exponential tails, just like in the scalar Bernstein's inequality:*

$$\Pr\left(\left\|\sum_{i=1}^{N}\sum_{j=1}^{M}\mathbf{X}_{ij}\right\|_2 \geq t\right) \leq 2d \exp\left(-\frac{3}{8}\min\left\{\frac{t^2}{\sigma^2}, \frac{t}{K}\right\}\right). \tag{245}$$

*Proof.* The following analysis is based on (Vershynin, 2018, Theorem 5.4.1).

**Reduction of MGF.** To bound the norm of the sum

$$\mathbf{S} \overset{\Delta}{=} \sum_{i=1}^{N}\sum_{j=1}^{M}\mathbf{X}_{ij}, \tag{246}$$

we need to control the largest and smallest eigenvalues of $\mathbf{S}$. We can do this separately. To put this formally, consider the largest eigenvalue

$$\lambda_{\max}(\mathbf{S}) \stackrel{\Delta}{=} \max_i \lambda_i(\mathbf{S}) \tag{247}$$

and note that

$$\|\mathbf{S}\|_2 = \max |\lambda_i(\mathbf{S})| = \max\{\lambda_{\max}(\mathbf{S}), \lambda_{\max}(-\mathbf{S})\} \tag{248}$$

and

$$\Pr(|\lambda_{\max}(\mathbf{S})| \geq t) = \Pr(\lambda_{\max}(\mathbf{S}) \geq t) + \Pr(\lambda_{\max}(-\mathbf{S}) \geq t) - \Pr(\lambda_{\max}(\mathbf{S}) \geq t \text{ and } \lambda_{\max}(-\mathbf{S}) \geq t), \tag{249}$$

which implies

$$\Pr(|\lambda_{\max}(\mathbf{S})| \geq t) \leq \Pr(\lambda_{\max}(\mathbf{S}) \geq t) + \Pr(\lambda_{\max}(-\mathbf{S}) \geq t). \tag{250}$$

To bound $\lambda_{\max}(\mathbf{S})$, we proceed with computing the moment generating function. We fix $\lambda \geq 0$ and use Markov's inequality to obtain

$$\Pr(\lambda_{\max}(\mathbf{S}) \geq t) = \Pr(e^{\lambda \lambda_{\max}(\mathbf{S})} \geq e^{\lambda t}) \leq e^{-\lambda t}\, \mathbb{E}[e^{\lambda \lambda_{\max}(\mathbf{S})}]. \tag{251}$$

Since by Vershynin (2018, Definition 5.4.2) the eigenvalues of $e^{\lambda \mathbf{S}}$ are $e^{\lambda \lambda_i(\mathbf{S})}$, we have

$$E \stackrel{\Delta}{=} \mathbb{E}[e^{\lambda \lambda_{\max}(\mathbf{S})}] = \mathbb{E}[\lambda_{\max}(e^{\lambda \mathbf{S}})]. \tag{252}$$

Since the eigenvalues of $e^{\lambda \mathbf{S}}$ are all positive, the maximum eigenvalue of $e^{\lambda \mathbf{S}}$ is bounded by the sum of all eigenvalues, the trace of $e^{\lambda \mathbf{S}}$, which leads to

$$E \leq \mathbb{E}[\mathrm{Tr}(e^{\lambda \mathbf{S}})]. \tag{253}$$

**Application of Lieb's inequality.** First note that

$$\mathbf{S} = \sum_{i=1}^{N-1} \sum_{j=1}^{M-1} \mathbf{X}_{ij} + \sum_{i=1}^{N-1} \mathbf{X}_{iM} + \sum_{j=1}^{M-1} \mathbf{X}_{Nj} + \mathbf{X}_{NM}. \tag{254}$$

To prepare the application of Lieb's inequality in Vershynin (2018, Lemma 5.4.9), let us separate the last term from the sum $\mathbf{S}$

$$E \left[ \leq \mathbb{E}\ \mathrm{Tr}\left( \exp\left( \sum_{i=1}^{N-1} \sum_{j=1}^{M-1} \lambda \mathbf{X}_{ij} + \sum_{i=1}^{N-1} \lambda \mathbf{X}_{iM} + \sum_{j=1}^{M-1} \lambda \mathbf{X}_{Nj} + \lambda \mathbf{X}_{NM} \right) \right) \right]. \tag{255}$$

Conditioning on $\{\mathbf{X}_{ij}\}_{i,j=1}^{N-1,M-1}$ and applying Vershynin (2018, Lemma 5.4.9) for the fixed matrix

$$\mathbf{H} \stackrel{\Delta}{=} \sum_{i=1}^{N-1} \sum_{j=1}^{M-1} \lambda \mathbf{X}_{ij} + \sum_{i=1}^{N-1} \lambda \mathbf{X}_{iM} + \sum_{j=1}^{M-1} \lambda \mathbf{X}_{Nj} \tag{256}$$

and the random matrix $\mathbf{Z} \stackrel{\Delta}{=} \lambda \mathbf{X}_{NM}$, we obtain

$$E \leq \mathbb{E}_{\{\mathbf{X}_{ij}\}_{i,j=1}^{N,M}} \left[ \mathrm{Tr}\left( \exp\left( \sum_{i=1}^{N-1} \sum_{j=1}^{M-1} \lambda \mathbf{X}_{ij} + \sum_{i=1}^{N-1} \lambda \mathbf{X}_{iM} + \sum_{j=1}^{M-1} \lambda \mathbf{X}_{Nj} + \lambda \mathbf{X}_{NM} \right) \right) \right]$$

$$\leq \mathbb{E}_{\{\mathbf{X}_{ij}\}_{i,j=1}^{N-1,M-1}} \left[ \mathbb{E}_{\mathbf{X}_{NM}} \left[ \mathrm{Tr}\left( \exp\left( \sum_{i=1}^{N-1} \sum_{j=1}^{M-1} \lambda \mathbf{X}_{ij} + \sum_{i=1}^{N-1} \lambda \mathbf{X}_{iM} + \sum_{j=1}^{M-1} \lambda \mathbf{X}_{Nj} + \lambda \mathbf{X}_{NM} \right) \right) \right] \right]$$

$$\leq \mathbb{E}_{\{\mathbf{X}_{ij}\}_{i,j=1}^{N-1,M-1}} \left[ \mathrm{Tr}\left( \exp\left( \sum_{i=1}^{N-1} \sum_{j=1}^{M-1} \lambda \mathbf{X}_{ij} + \sum_{i=1}^{N-1} \lambda \mathbf{X}_{iM} + \sum_{j=1}^{M-1} \lambda \mathbf{X}_{Nj} + \log \mathbb{E}_{\mathbf{X}_{NM}} e^{\lambda \mathbf{X}_{NM}} \right) \right) \right]. \tag{257}$$

We continue similarly: separate the next term $\lambda \mathbf{X}_{N-1,M-1}$ from the remaining sum and apply Vershynin (2018, Lemma 5.4.9) again for $\mathbf{Z} = \lambda \mathbf{X}_{N-1,M-1}$. Repeating this process $NM$ times, we obtain

$$\Pr(\lambda_{\max}(\mathbf{S}) \geq t) \leq \mathrm{Tr}\left(e^{-\lambda t} \exp\left(\sum_{i=1}^{N} \sum_{j=1}^{M} \log \mathbb{E} \exp \lambda \mathbf{X}_{ij}\right)\right). \tag{258}$$

**MGF of the individual terms.** It remains to bound the matrix-valued moment generating function $\mathbb{E} e^{\lambda \mathbf{X}_{ij}}$ for each term $\mathbf{X}_{ij}$. We now use Lemma 21.

**Completion of the proof.** Using Lemma 21, we obtain

$$\mathbb{E}\left[\exp\left(\lambda \mathbf{X}_{ij}/K\right)\right] \preceq \exp\left(g(\lambda)\,\mathbb{E}[\mathbf{X}_{ij}^2]/K^2\right) \Leftrightarrow \prod_{i,j=1}^{N,M} \mathbb{E}\left[\exp\left(\lambda \mathbf{X}_{ij}/K\right)\right] \preceq \prod_{i,j=1}^{N,M} \exp\left(g(\lambda)\,\mathbb{E}[\mathbf{X}_{ij}^2]/K^2\right)$$

$$\Leftrightarrow \prod_{i,j=1}^{N,M} \mathbb{E}\left[\exp\left(\lambda \mathbf{X}_{ij}/K\right)\right] \preceq \exp\left(g(\lambda) \sum_{i,j=1}^{N,M} \mathbb{E}[\mathbf{X}_{ij}^2]/K^2\right), \quad (259)$$

which implies

$$\prod_{i,j=1}^{N,M} \mathbb{E}\left[\exp\left(\lambda \mathbf{X}_{ij}/K\right)\right] \preceq \exp\left(g(\lambda) \sum_{i,j=1}^{N,M} \mathbb{E}[\mathbf{X}_{ij}^2]/K^2\right). \tag{260}$$

Also, given that $\mathbf{X}_{ij}$ are independent, we have

$$\prod_{i,j=1}^{N,M} \mathbb{E}\left[\exp\left(\lambda \mathbf{X}_{ij}/K\right)\right] = \exp\left(\log\left(\prod_{i,j=1}^{N,M} \mathbb{E} \exp\left(\lambda \mathbf{X}_{ij}/K\right)\right)\right) = \exp\left(\sum_{i,j=1}^{N,M} \log \mathbb{E} \exp\left(\lambda \mathbf{X}_{ij}/K\right)\right), \quad (261)$$

which combined with (260)

$$\exp\left(\sum_{i,j=1}^{N,M} \log \mathbb{E} \exp\left(\lambda \mathbf{X}_{ij}/K\right)\right) \preceq \exp\left(g(\lambda) \sum_{i,j=1}^{N,M} \mathbb{E}[\mathbf{X}_{ij}^2]/K^2\right), \tag{262}$$

and applying the trace to both sides yields

$$\mathrm{Tr}\left(\exp\left(\sum_{i,j=1}^{N,M} \log \mathbb{E} \exp\left(\lambda \mathbf{X}_{ij}/K\right)\right)\right) \leq \mathrm{Tr}\left(\exp\left(g(\lambda)\,\tilde{\mathbf{Z}}/K^2\right)\right), \tag{263}$$

where $\tilde{\mathbf{Z}} \triangleq \mathbb{E}\left[\sum_{i=1}^{N} \sum_{j=1}^{M} \mathbf{X}_{ij}^2\right]$. Since the trace of $\exp(g(\lambda)\tilde{\mathbf{Z}}/K^2)$ is a sum of $d$ positive eigenvalues, it is bounded by $d$ times the maximum eigenvalue and using Vershynin (2018, Definition 5.4.2), we obtain

$$\mathrm{Tr}\left(\exp\left(g(\lambda)\,\tilde{\mathbf{Z}}/K^2\right)\right) \leq d\,\lambda_{\max}\left(\exp\left(g(\lambda)\,\tilde{\mathbf{Z}}/K^2\right)\right) = d\,\exp\left(g(\lambda)\,\lambda_{\max}(\tilde{\mathbf{Z}}/K^2)\right)$$
$$= d\,\exp\left(g(\lambda)\,\|\tilde{\mathbf{Z}}\|/K^2\right) = d\,\exp\left(g(\lambda)\,\sigma^2/K^2\right). \quad (264)$$

Combining (258) and (264) we get

$$\Pr(\lambda_{\max}(\mathbf{S}) \geq Kt) \leq e^{-\lambda t}\,\mathrm{Tr}\left(\exp\left(\sum_{i=1}^{N} \sum_{j=1}^{M} \log \mathbb{E} \exp \lambda \mathbf{X}_{ij}/K\right)\right) \leq d\,\exp\left(-\lambda t + g(\lambda)\,\sigma^2/K^2\right), \quad (265)$$

which implies

$$\Pr(\lambda_{\max}(\mathbf{S}) \geq t) \leq d\,\exp\left(-\frac{\lambda}{K}t + \frac{g(\lambda)}{K^2}\sigma^2\right). \tag{266}$$

Minimizing over $\lambda > 0$, the minimum occurs at

$$\lambda = \log\left(1 + \frac{Kt}{\sigma^2}\right), \quad t \geq 0. \tag{267}$$

Plugging this into the bound, we get

$$\Pr\left(\lambda_{\max}\left(\mathbf{S}\right) \geq t\right) \leq d \, \exp\left(-\frac{\sigma^2}{K^2} \, h\left(\frac{Kt}{\sigma^2}\right)\right), \tag{268}$$

where

$$h(u) = (1+u)\log(1+u) - u, \quad \text{for } u > 0. \tag{269}$$

We know that Boucheron et al. (2013, Exercise 2.8)

$$h(u) \geq \frac{u^2}{2(1+u/3)}, \tag{270}$$

with $u > 0$ and thus

$$\Pr\left(\lambda_{\max}\left(\mathbf{S}\right) \geq t\right) \leq d \, \exp\left(-\frac{\sigma^2}{K^2} \, \frac{u^2}{2(1+u/3)}\right), \tag{271}$$

where $u = \frac{Kt}{\sigma^2}$. Substituting $u$ in (271), we obtain

$$\Pr\left(\lambda_{\max}\left(\mathbf{S}\right) \geq t\right) \leq d \, \exp\left(-\frac{t^2/2}{\sigma^2 + Kt/3}\right). \tag{272}$$

Following similar steps with $-\mathbf{S}$ instead of $\mathbf{S}$ and using (250), yields

$$P(|\lambda_{\max}(\mathbf{S})| \geq t) \leq \begin{cases} 2d\exp\left(\frac{-3t^2}{8\sigma^2}\right), & t \leq \sigma^2/K \\ 2d\exp\left(\frac{-3t}{8K}\right), & t > \sigma^2/K. \end{cases} \tag{273}$$

Intuitively, for small $t$, i.e., $t \leq \sigma^2/K$, we have a sub-Gaussian bound, while for large $t$, i.e., $t > \sigma^2/K$, we have a sub-exponential bound. Looking for the tightest bound, we may write

$$P(|\lambda_{\max}(\mathbf{S})| \geq t) \leq 2d\exp\left(-\frac{3}{8}\min\left\{\frac{t^2}{\sigma^2}, \frac{t}{K}\right\}\right). \tag{274}$$

$\square$

