# OpenReview forum: "AdaCubic: An Adaptive Cubic Regularization Optimizer for Deep Learning"
_TMLR — Accepted by TMLR_

### Review · Reviewer_mfxj · 2025-12-13

**Summary Of Contributions:**

The paper introduces AdaCubic, a second-order optimization method that dynamically adapts the regularization parameter ($M$) of the cubic term in Newton’s cubic regularized method. By solving an auxiliary optimization problem with cubic constraints. the method avoids saddle points and ensures convergence. To maintain efficiency, it employs Hutchinson’s method to approximate the Hessian diagonal, reducing memory complexity to $O(d)$

**Additional Comments:**

Lower Confidence. I am not an expert in optimization or second-order methods. Reviews are more from the application point of view.

**Audience:**

Yes

**Audience Explanation:**

The paper addresses two major points of second-order methods: computational cost (addressed via diagonal approximation) and hyperparameter sensitivity (addressed via the adaptive cubic mechanism). The promise of a "tuning-free" optimizer that performs competitively with Adam/SGD is of high interest to practitioners who spend significant resources on hyperparameter search.

**Claims And Evidence:**

Yes

**Claims Explanation:**

Theoretical: The paper claims to introduce a novel method that dynamically adapts the cubic regularization parameter ($M$) to avoid saddle points. This is well-supported by the theoretical derivation in Section 2, specifically the formulation of the auxiliary constrained optimization problem and the proofs of strong duality (Theorem 1 and Corollary 1) and equivalence (Theorem 2). The convergence analysis (Theorem 3) accurately establishes the $O(1/\epsilon^{3/2})$ rate, consistent with standard Cubic Regularization literature.

Efficiency: The claim of low memory complexity ($O(d)$) is accurately supported by the use of Hutchinson’s method to approximate the Hessian diagonal (Eq. 30)

Robustness: The claim that AdaCubic works well with a "pre-fixed set of hyperparameters" (sourced from Trust Region literature) without fine-tuning is supported by the experimental setup.

Performance: The claim that AdaCubic "outperforms or competes" is supported by Tables 2 and 5 (GLUE benchmark and Signal Processing tasks).

**Requested Changes:**

Wall-Clock Time Analysis: While Figure 4 mentions "Cumulative time," the text heavily emphasizes per-epoch performance. It will be the best to add a specific Table comparing the wall-clock time required for AdaCubic vs. SGD/Adam to reach a specific validation accuracy or loss threshold. This is critical to assess if the extra computation per step (Hutchinson's estimation) is worth the improved convergence rate.

Clarification of "Large-Scale": The abstract claims applicability to "large-scale applications." The experiments use ResNet on CIFAR and BERT-base, which are arguably medium-scale by modern standards. Please either qualify this claim (e.g., "medium-to-large scale") or include a discussion/experiment on a truly large-scale setting (e.g., a larger LLM) to demonstrate memory scaling.

Performance Gaps: Explicitly discuss cases where the method did not outperform the baseline (e.g., specific GLUE tasks or CIFAR-100 if applicable). Analyzing why the diagonal Hessian approximation might fail in these specific instances would add valuable insight for the reader.

---

> ### Author Response · Authors · 2026-01-06
> **Response to reviewer mfxj**
>
> We thank the reviewer for the comments. Changes in the main body of the manuscript related to these comments are highlighted in cyan, unless stated otherwise. In addition, all answers provided below are also included at the end of the revised manuscript, with quoted text and cross-references for easy navigation of the main text.
>
> **Reviewer Comment #1**
> *Wall-Clock Time Analysis: While Figure 4 mentions “Cumulative time”, the text heavily emphasizes per-epoch performance. It would be best to add a Table comparing the wall-clock time required by AdaCubic vs. SGD/Adam to reach a specific validation accuracy or loss threshold. This is critical to assess if the extra computation per step (Hutchinson’s estimation) is worth the improved convergence rate.*
>
>
> **Author response:** We thank the reviewer for raising the issue of wall-clock efficiency. To address this comment, changes have been made in the revised manuscript marked with cyan color in Section 6. For completeness, we quote the changes below.
>
> >Figure 5a shows the training loss vs. cumulative time for SGD, AdaHessian, and AdaCubic. Figure 5b shows the training loss vs. epochs for SGD, AdaHessian, and AdaCubic. The training loss in Figure 5b corresponds to that in Figure 5a. The horizontal dashed line in Figure 5a marks the target loss threshold of 0.15. AdaCubic reaches this threshold after 55 epochs and 42.40 minutes. In comparison, SGD and AdaHessian require 83 and 81 epochs, corresponding to 35.16 and 61.85 minutes. Table 9 summarizes the latter results. Although AdaCubic needs more time than SGD due to second-order information, it reaches the desired loss in fewer epochs without any LR tuning. This highlights AdaCubic as an efficient trade-off between computational cost and convergence quality.
>
> **Reviewer Comment #2** *Clarification of “Large-Scale”: The abstract claims applicability to “large-scale applications”. The experiments use ResNet on CIFAR and BERT-base, which are arguably medium-scale by modern standards.
> Please either qualify this claim (e.g., “medium-to-large scale”) or include a discussion/experiment on a truly
> large-scale setting (e.g., a larger LLM) to demonstrate memory scaling.*
>
> **Author response:** We thank the reviewer for pointing this out. We agree that our previous wording (“large-scale applications”) could be interpreted as referring to training at the scale of multi-billion-parameter LLMs. Our intention was instead to emphasize that AdaCubic is designed to scale efficiently with model size and curvature estimation and has been validated on practical deep-learning workloads beyond the small-model settings typically used in prior cubic-regularization studies (Huang et al., 2022; Wang et al., 2019; Carmon & Duchi, 2019; Tripuraneni et al., 2018; Zhou & Gu, 2020; Wang et al., 2020b; Cartis et al., 2011a;b; Kohler & Lucchi, 2017; Park et al., 2020; Kamzolov et al., 2023). To avoid ambiguity, we have revised the wording in the abstract and conclusions. The abstract now states
> that AdaCubic “leverages cubic regularization in scalable deep-learning applications,” and the conclusions refer to “practical deep-learning applications” rather than “large-scale.” We believe this more accurately reflects the scope of our experiments. Full-scale LLM experiments are beyond our current computational resources. The changes are highlighted with cyan color in the Abstract and Conclusions of the revised manuscript.

---

> ### Author Response · Authors · 2026-01-06
> **Response to reviewer mfxj**
>
> **Reviewer Comment #3** *Performance Gaps: Explicitly discuss cases where the method did not outperform the baseline (e.g., specific GLUE tasks or CIFAR-100 if applicable). Analyzing why the diagonal Hessian approximation might fail in
> these specific instances would add valuable insight for the reader.*
>
> **Author response:** We thank the reviewer for highlighting the importance of discussing performance gaps. We recall that AdaCubic achieves the best and second-best performance on the CV and NLP benchmarks. On the CMI benchmark, AdaCubic outperforms ADAM. To address this comment, changes have been made in the revised manuscript related to the CV and NLP tasks, which are marked with cyan color in Section 5 of the revised manuscript. For completeness, we quote the changes below. The first and second quotes, highlighted with cyan color, can be found below the Figures 1 and 3, respectively.
>
> >On CIFAR-10, AdaCubic consistently outperforms first-order methods (SGD, Adam) and ranks second to AdaHessian, with very small gaps of 0.15% and 0.5% for ResNet20 and ResNet32, respectively, as summarized in Table 4. On CIFAR-100 without spatial averaging, AdaCubic trails the best-performing optimizer by at most 0.81%. Due to its larger number of classes and increased classification difficulty, CIFAR-100 will possibly lead to optimization regimes with stronger parameter interactions. Since AdaCubic, like AdaHessian, relies on a diagonal approximation of the Hessian, it does not explicitly capture such off-diagonal curvature effects, which may partially explain the observed gap. Importantly, when spatial averaging is applied, the performance of AdaCubic improves and becomes closer to that of AdaHessian and SGD, confirming that part of the gap is related to high-variance curvature estimation.
>
> >On the NLU benchmark, Table 5, AdaCubic consistently achieves either the best or the second-best performance across all tasks, with the performance gaps reported in the ∆ column remaining small. The second-best performance of AdaCubic on certain GLUE tasks can be understood in light of recent Hessian-based analyses of Transformer Zhang et al. (2024). In particular, Zhang et al. (2024) shows thatTransformer models exhibit block-wise heterogeneity in their Hessian structure, with strong curvature differences and interactions across parameter groups. While AdaCubic explicitly leverages second-order information through diagonal Hessian approximations, such approximations may be insufficient to capture cross-parameter or block-level curvature interactions fully. This likely explains why AdaCubic remains highly competitive but does not consistently outperform finely tuned baselines on transformer-based tasks. Similar conclusions are drawn for the LM benchmark, where AdaCubic consistently achieves either the best or the second-best performance across all datasets. Overall, it should be noted that AdaCubic exhibits the best or the second-best performance with a pre-fixed universal set of parameters, while SGD and AdaHessian are fine-tuned w.r.t. the initial
> learning rate.

---

### Review · Reviewer_d2MB · 2025-12-19

**Summary Of Contributions:**

This paper proposed a new optimizer for neural networks called AdaCubic.
The method relies on a second order method with a cubic regularization term. Unlike previous methods, which require finetuning the hyperparameters, the regularization term is adaptive and thus user friendly.
The paper proved convergence of the optimizer and exhibited strong emprical results on several tasks of wide machine learning domains.

**Audience:**

Yes

**Audience Explanation:**

A study in optimizers is always helpful to machine learning community. With the development of large models and hardware design, a fast, convergent optimizer is always an important topic.

**Claims And Evidence:**

Yes

**Claims Explanation:**

Partly yes.
The motivation and the problem statement is quite clear.
The paper is theorectically strong with rigorous proofs for convergence.
The experiment section of the paper exhibited performance of the proposed optimizer comparing with other optimizers, including experiments on neural network task specific performance, curves of convergence and cummulative timing. However, even though on some tasks the AdaCubic shows slightly better scores, e.g., on CIFAR, the convergence rate and the cummulative timing don't show significant improvement, which hinders the wide application of AdaCubic.
The authors claim other optimizers need finetuning of hyperparameters while AdaCubic does not. However, AdaCubic also have a set of hyperparameters, and it is unclear how these hyperparameters influence the training dynamic.

**Requested Changes:**

Despite the strong mathematical results, I think there should be text narrative part in the paper.
For example, why (3) is a form of non-convex optimization, why such sum decomposition?
Maybe add some text before lemma 2 so that it is more clear to reader what is going on there, it is not quite smooth transition.
For the self-containedness, some details e.g. $\lambda^+_d(\mathbf{B})$ in algorithm 2 should also be explicitly explained.
In figure 1, why the loss curve of AdaCubic does not drop a lot with the learning rate decay?
Since AdaCubic is a second order cubic regularized method, it should be compared with more related cubic regularized methods.
Is there a clear application field where AdaCubic dominates all the major optimizers?

---

> ### Author Response · Authors · 2026-01-06
> **Response to reviewer d2MB**
>
> We thank the reviewer for the careful assessment. We would like to clarify two key points regarding convergence behavior and hyperparameter usage.
>
> **Reviewer Comment.** *The motivation and the problem statement is quite clear. The paper is theoretically strong with rigorous proofs for convergence. The experiment section of the paper exhibited performance of the proposed optimizer comparing with other optimizers, including experiments on neural network task specific performance, curves of convergence and cumulative timing. However, even though on some tasksthe AdaCubic shows slightly better scores, e.g., on CIFAR, the convergence rate and the cumulative timing don’t show significant improvement, which hinders the wide application of AdaCubic. The authors claim other optimizers need finetuning of hyperparameters while AdaCubic does not. However, AdaCubic also have a set of hyperparameters, and it is unclear how these hyperparameters influence the training dynamic.*
>
> **Author response:**
>
> *Convergence and cumulative timing.* To better assess the trade-off between convergence and wall-clock time, we include Figures 5a and 5b in the revised manuscript. Figure 5a depicts the training loss vs. cumulative time over epochs, while Figure 5b depicts the training loss vs. epochs. Both Figures 5a and 5b depict comparisons between SGD, AdaHessian, and AdaCubic on ResNet20 and CIFAR-10. The horizontal dashed line in Figure 5a marks the target loss threshold of 0.15. AdaCubic reaches this threshold after 55 epochs and 42.40 minutes. In comparison, SGD and AdaHessian require 83 and 81 epochs, corresponding to 35.16 and 61.85 minutes. Table 9 summarizes the latter results. Although AdaCubic needs more time than SGD due to computation of second-order information, AdaCubic reaches the desired loss in fewer epochs without any LR tuning. In addition, according to Algorithm 2, AdaCubic relies only on the approximated diagonal Hessian for its updates, which implies a theoretical (algorithmic) memory footprint of O(d). The gap between practical and theoretical memory costs comes from the design of modern deep-learning frameworks, such as PyTorch, which are optimized for first-order optimization methods. Thus, computing the diagonal Hessian approximation requires retaining intermediate gradient information. Developing a custom implementation that directly computes the diagonal Hessian without storing such intermediates is beyond the scope of this work. A more detailed discussion of the computational complexity of AdaCubic is provided in Section 6, where both the time complexity and the loss reduction (convergence behavior) are analyzed. The portions of the text added in Section 6 during the revision are highlighted in color. We hope this analysis provides a clearer picture of the trade-off between computational cost and convergence behavior for AdaCubic.
>
> *Hyperparameter tuning.* In Algorithm 1, the parameters $\eta_1$ and $\eta_2$ govern step acceptance, the parameters $\alpha_1$ and $\alpha_2$ regulate the update of the trust-region boundary $\xi_k$, and the parameter $\kappa_{\text{easy}}$ balances accuracy and efficiency in the inner solver presented in Algorithm 2. AdaCubic adaptively computes the dual parameter $\nu_{k+1}$ in Algorithm 2, which determines the acceptance ratio $\rho_k$, the step $\mathbf{s}_{k+1}$, and, in turn the trusts-region parameter $\xi_k$.
>
> This relationship enables the optimization dynamics to adapt automatically while keeping the remaining hyperparameters fixed. As a result, AdaCubic can respond effectively to the local geometry of the loss landscape through the automatic adjustment of the trust-region parameter $\xi_k$ and perform competitively across the benchmarks in Section 5.
>
> The intuition behind these hyperparameters, as well as the rationale for why AdaCubic performs robustly with fixed hyperparameter values, is now discussed immediately after Algorithm 2 and highlighted in green. In addition, all answers provided below are also included at the end of the revised manuscript, with quoted text and cross-references for easy navigation of the main text.

---

> ### Author Response · Authors · 2026-01-06
> **Response to reviewer d2MB**
>
> **Reviewer Comment #1** *Despite the strong mathematical results, I think there should be a text narrative part in the paper. For example, why (3) is a form of non-convex optimization, and why such a sum decomposition? Maybe add some text before Lemma 2 so that it is clearer to the reader what is going on there; the transition is not quite smooth.*
>
>
> **Author response:** We thank the reviewer for this comment. The function $f(x)$ in (3) is non-convex by definition, as is the sum of non-convex functions. We have marked that in the revised manuscript by placing the symbol $\Delta$ above $=$. We have added information clarifying the role of Lemma 2, marked in red, immediately above its declaration. Additionally, the roles of Lemma 2 and the remaining theoretical results in Section 2 are now explicitly highlighted at the beginning of this section. Further illustration of the relationships and dependencies among the theoretical results is provided in Figure 6.
>
> **Reviewer Comment #2** *For the self-containedness, some details e.g. $\lambda_d^+ (\mathbf{B})$ in Algorithm 2 should also be explicitly explained.*
>
> **Author response:** We thank the reviewer for noticing this detail. We have added a description of $\lambda_d^+(\mathbf{B})$ just above (7) marked with red. The remaining parameters of Algorithms 1 and 2 are mapped to the parameters of the manuscript. A relation of the main theoretical results with Algorithms 1 and 2 also exists at the beginning of Section 4. In addition, comments integrated into Algorithms 1 and 2 provide a more self-contained description. An intuitive explanation of Algorithm 2 has also been added to the revised manuscript immediately following its appearance highlighted with green color.
>
> **Reviewer Comment #3** *In Figure 1, why does the loss curve of AdaCubic not drop a lot with the LR decay?*
>
> **Author response:** We thank the reviewer for noticing this detail. Note that AdaCubic does not have an LR to fine-tune. Thus, a drop in loss is not expected. The only parameter that is fine-tuned is $M$ , and this is done automatically by leveraging Theorems 1 and 2, and Algorithms 1 and 2. A description relevant to the only parameter that is fine-tuned automatically, i.e., $M$ , can be found at the beginning of Section 4.
>
> **Reviewer Comment #4** *Since AdaCubic is a second-order cubic regularized method, it should be compared with more related cubic regularized methods.*
>
> **Author response:** Cubic regularization has been used in previous papers (Huang et al., 2022; Wang et al., 2019; Carmon & Duchi, 2019; Tripuraneni et al., 2018; Zhou & Gu, 2020; Wang et al., 2020b; Cartis et al., 2011a;b; Kohler & Lucchi, 2017; Park et al., 2020; Kamzolov et al., 2023). However, in all these papers, the experimental evaluation was mostly theoretical and accompanied by small-scale experiments where the full Hessian matrix is required. Thus, their application to deep neural networks, and therefore their comparison with AdaCubic, is not yet possible. An interesting direction for future work is to extend AdaCubic to incorporate acceleration terms, such as those proposed in (Zhou & Gu, 2020; Wang et al., 2020b), in combination with approximate Hessian computations, and to study their effect on the optimization procedure.
>
> **Reviewer Comment #5** *Is there a clear application field where AdaCubic dominates all the major optimizers?*
>
> **Author response:** A clear application field in which AdaCubic excels is CMI, as shown in Table 8. According to Table 8, AdaCubic improves mean accuracy over Adam by approximately 1.7% on averageacross the Native, WhatsApp, and YouTube benchmarks. In addition, it should be noted that, in the natural language understanding tasks in Table 5, AdaCubic remains highly competitive, achieving the best or second-best results without fine-tuning any parameters. The column $\Delta$ in Table 5 is added in the revised manuscript to make these differences explicit. All related changes are highlighted in red.

---

### Review · Reviewer_1HGb · 2025-12-20

**Summary Of Contributions:**

**Summary:**

The paper proposes AdaCubic as a new second-order optimization algorithm suited for training deep neural networks. Compared to existing solutions that approximate the full Newton method, such as AdaHessian, the AdaCubic algorithm explicitly addresses the issue of saddle points in non-convex loss landscapes by leveraging cubic regularization. Concretely, the general cubic regularized Newton method is reformulated as a constrained optimization problem, whose batch-wise solutions involve automatically adapting the cubic regularization hyperparameter. Extensive theoretical results demonstrate the equivalence between the original and reformulated optimization problems, while also providing convergence guarantees for AdaCubic.
Finally, an empirical study across various learning tasks and data modalities demonstrates the effectiveness and robustness of AdaCubic with fixed hyperparameters compared to other optimization algorithms whose learning rates have been tuned.

**Strengths:**

- The theoretical results demonstrate AdaCubic's convergence under reasonable assumptions, taking into account the use of a diagonal Hessian matrix, as well as the noise inherent in the batch-wise computation of the gradient and the diagonal elements of the Hessian.
- The empirical study follows established benchmarks and demonstrates that AdaCubic can be applied with a single set of hyperparameters across various learning tasks, data modalities, neural network architectures, and other training hyperparameters, including the number of training epochs.
- The computational and memory complexity of AdaCubic is well-suited for practical large-scale deep learning applications, thanks to the use of Hutchinson's method for approximating the Hessian matrix.
- The paper provides a structured overview of existing works that target the issue of escaping from saddle points. Moreover, detailing their shortcomings provides a clear motivation for the development of AdaCubic.
- The paper clearly indicates that the code will be publicly available upon acceptance, reducing the barrier to using AdaCube in practice.

**Weaknesses:**
- (A) While a main contribution of AdaCubic is its effectiveness under fixed hyperparameter values, providing more details on how these values are selected and how different choices affect its performance would make AdaCubic more accessible for practitioners. Currently, the reader is referred to another paper to understand the physical meanings of the hyperparameters.
- (B) The presentation of the training hyperparameters and model architectures is difficult to follow. For example, the computer vision experiments clearly note the mini-batch size of 256, whereas this information is missing for the other learning tasks. A condensed and standardized tabular overview could resolve such issues.
- (C) There is no clear (potential tabular) overview of the differences and commonalities of AdaCubic to other established optimizers, such as SGD, Adagrad, Adam, and AdaHessian. Such an overview would make it easier to understand the advantages, e.g., cubic regularization, and the disadvantages, e.g., higher computational complexity due to the extra backward pass for Hutchinson's method, of AdaCubic.
- (D) Although the paper notes that the code will be made publicly available upon acceptance, providing it as supplementary material for review would have been insightful.

**Additional Comments:**

**Questions:**

- (I) In its principal design, the empirical evaluation study seems to be related to the one performed in the AdaHessian paper. For example, both report results for ResNet20 and ResNet32 on $\\texttt{CIFAR-10}$ and rely on the $\\texttt{GLUE}$ benchmark. However, specific hyperparameter values differ. Concretely, 500 as the number of training epochs for $\\texttt{CIFAR-10}$ is quite large compared to 160 training epochs used in the AdaHessian paper. Figure 1 also demonstrates that AdaCubic requires more epochs until convergence. What was the reason to increase the number of training epochs?
- (II) How are the (initial) learning rates of the optimizers SGD, Adam, and AdaHessian tuned? Most likely, these learning rates stem from the respective benchmark papers. However, making this process more transparent would be beneficial.
- (III) What are the effects of $\\kappa_{\mathrm{easy}} \in (0, 1)$ and how is its value defined?
- (IV) In the sentence above Corollary 3, you state that you repeat the proof with the diagonal Hessian matrix $\\mathrm{Diag} (\\nabla f(\\mathbf{x}))$ instead of the gradient $\\nabla f(\\mathbf{x})$. Do you rather mean $\\mathrm{Diag} (\\nabla^2 f(\\mathbf{x}))$ instead of the full Hessian matrix $\\nabla^2 f(\\mathbf{x})$?

**Minor Comments:**
- The paper uses `\citet` and `\citep` inconsistently:
  - When the citation is grammatically part of the sentence, e.g., "Authors et al. proposed …", `\citet` is often the right choice.
  - When the citation is only a parenthetical reference, e.g.,  "... as established previously (Authors et al., 2019) ...", `\citep` is typically the correct choice.
- There is a missing space in "subjectto" in Eq. (22).
- The `itemize` environment would be more suitable to describe the four groups of datasets as part of the $\\texttt{GLUE}$ benchmark.

**Audience:**

Yes

**Audience Explanation:**

Optimization algorithms with strong robustness for fixed hyperparameter values across diverse deep learning applications are of particular **interest for a large audience**. In addition to the presented empirical study, the theoretical results can also provide further insights into research on second-order optimization and the avoidance of saddle points.

**Claims And Evidence:**

Yes

**Claims Explanation:**

Overall, the aforementioned strengths **support the paper's major claims**. Primarily, the presentation within the paper can be enhanced to better understand the properties and behavior of AdaCubic, which are essential prerequisites for its practical adoption.

**Requested Changes:**

Please address the aforementioned weaknesses, answer my questions at the end of this review, and resolve the minor issues also listed at the end of this review. The **critical parts** are the weakness (A) and the questions (I-III).

Finally, I would like to note that the paper writing is quite technical, which is good on the one hand. But on the other hand, more intuitive explanations, added to one place or another, would further ease the understanding for non-experienced readers in the optimization literature. For example, outlining the steps of deriving the proposed framework at the start of Section 2 (as also done in Section 3) would make the general idea clearer from the beginning.

---

> ### Author Response · Authors · 2026-01-06
> **Response to reviewer 1HGb**
>
> **Reviewer Requested Changes:** *Please address the aforementioned weaknesses, answer my questions at the end of this review, and resolve the minor issues also listed at the end of this review. The critical parts are the weakness (A) and the questions (I-III). Finally, I would like to note that the paper writing is quite technical, which is good on the one hand. But on the other hand, more intuitive explanations, added to one place or another, would further ease the understanding for non-experienced readers in the optimization literature. For example, outlining the steps of deriving the proposed framework at the start of Section 2 (as also done in Section 3) would make the general idea clearer from the beginning.*
>
> **Author response:** We thank the reviewer for the constructive comments and suggestions. All changes made in the main body of the manuscript in response to this review are highlighted in green, unless stated otherwise. The critical points, namely Weakness (A) and Questions (I–III), are addressed thoroughly below individually. In addition, all answers provided below are also included at the end of the revised manuscript, with quoted text and cross-references for easy navigation of the main text.
>
> Regarding the writing style, we acknowledge that the manuscript is technically dense and agree that additional intuitive explanations can further improve accessibility for non-expert readers. In particular, following the reviewer’s suggestion, we have added a high-level outline of the steps leading to the proposed framework at the beginning of Section 2. For completeness we quote this part below.
>
> >Section 2.1 introduces the fundamental definitions used throughout the paper, including the basic formulation of the CR method, which serves as a core building block of the proposed framework. Section 2.2 then introduces an auxiliary constrained optimization problem that forms the foundation of the AdaCubic. The key intuition is to reformulate the classical CR method as a constrained problem in which the cubic regularization term appears explicitly as a constraint. By leveraging Lagrange multiplier theory, this reformulation yields an adaptive update mechanism in which the strength of the cubic regular-
> ization term of the CR method is automatically adjusted during optimization. To derive this update mechanism Lemmata 1, 2, Theorem 1, Corollary 1, and Theorem 2 are introduced. Lemma 1 establishes that the auxiliary constrained problem admits a global minimizer and ensures that each optimization step is well defined. Lemma 2 is used to establish Theorem 1 which in turn is used to derive Corollary 1. Corollary 1 shows that the auxiliary optimization problem is characterized by strong duality (Boyd & Vandenberghe, 2004, Section 5.4). The latter theoretical results are then combined to derive Theorem 2 which provides the basis to replace the fixed cubic regularization parameter of the CR method with an adaptive one and finally derive the AdaCubic optimizer presented in Section 4.

---

> ### Author Response · Authors · 2026-01-06
> **Response to reviewer 1HGb**
>
> **Reviewer Weakness (A)** *While a main contribution of AdaCubic is its effectiveness under fixed hyperparameter values, providing more details on how these values are selected and how different choices affect its performance would make AdaCubic more accessible for practitioners. Currently, the reader is referred to another paper to understand the physical meanings of the hyperparameters.*
>
> **Author response:** We thank the reviewer for this insightful comment. To address this comment, changes have been made in the revised manuscript marked with green color in Section 4 just below Algorithm 2.
>
> To elaborate further, we note that the specific fixed parameter values used in our experiments follow established recommendations from  (Conn et al., 2000, Chapter 17). In particular,  (Conn et al., 2000, Chapter 17) presents empirical evidence showing that robust performance in trust-region methods is typically obtained by choosing permissive acceptance thresholds (e.g., small $\eta_1 = 0.05$), conservative criteria for declaring very successful steps (e.g., large $\eta_2 = 0.75$), and carefully balanced update factors that regulate how the trust-region is adapted after very successful ($\alpha_1 = 2.5$) and unsuccessful ($\alpha_2 = 0.25$) iterations.
>
> Choosing $\alpha_1 = 2.5$ induces a sufficiently strong adjustment of the trust-region parameter after very successful iterations, allowing the algorithm to expand its search into regions where the local model is deemed highly reliable. Conversely, choosing $\alpha_2 = 0.25$ enforces a strong corrective adjustment after unsuccessful iterations, encouraging the algorithm to focus its search more closely on the current iterate, where the model is expected to be more accurate.
>
> As discussed at the end of the added text below Algorithm 2, the adaptivity of AdaCubic with respect to the dual parameter allows the algorithm to automatically adjust its optimization dynamics, thereby reducing its sensitivity to fixed hyperparameters. This property explains the robustness of AdaCubic across a wide range of benchmarks.
>
> **Reviewer Weakness (B)** *The presentation of the training hyperparameters and model architectures is difficult to follow. For example, the computer vision experiments clearly note the mini-batch size of 256, whereas this information is missing for the other learning tasks. A condensed and standardized tabular overview could resolve such issues.*
>
> **Author response:** We thank the reviewer for this comment. To improve clarity and consistency, we have added a tabular overview of all experimental settings. Specifically, Table 1 summarizes the universal hyperparameter values used by AdaCubic across all experimental evaluations. In addition, Tables 2 and 3 now summarize, for each benchmark, the datasets, model architectures, batch sizes, number of epochs, optimizers, and learning rate. Tables 1, 2 and 3 can be found at the beginning of Section 5.
>
> **Reviewer Weakness (C)** *There is no clear (potential tabular) overview of the differences and commonalities of AdaCubic to other established optimizers, such as SGD, Adagrad, Adam, and AdaHessian. Such an overview would make it easier
> to understand the advantages, e.g., cubic regularization, and the disadvantages, e.g., higher computational complexity due to the extra backward pass for Hutchinson’s method, of AdaCubic.*
>
> **Author response:** We thank the reviewer for this comment. We have addressed it by adding Table 10 to Section 6 of the revised manuscript. A detailed explanation of the table is provided immediately following its appearance and is highlighted in green.
>
> **Reviewer Weakness (D)** *Although the paper notes that the code will be made publicly available upon acceptance, providing it as supplementary material for review would have been insightful.*
>
> **Author response:** The implementation of the algorithm can be found at: https://gitfront.io/r/mysubpapers/3F7bdJzvSvVn/AdaCubic/

---

> ### Author Response · Authors · 2026-01-06
> **Response to reviewer 1HGb**
>
> **Reviewer Question (I)** *In its principal design, the empirical evaluation study seems to be related to the one performed in the AdaHessian paper. For example, both report results for ResNet20 and ResNet32 on CIFAR-10 and rely on the GLUE benchmark. However, specific hyperparameter values differ. Concretely, 500 as the number of training epochs for CIFAR-10 is quite large compared to 160 training epochs used in the AdaHessian paper. Figure 1 also demonstrates that AdaCubic requires more epochs until convergence. What was the reason to increase the number of training epochs?*
>
> **Author response.** For CIFAR-10, the initial learning rate (LR) and LR schedule follow Yao et al. (2021). For the remaining benchmarks, learning rates and schedules were determined through extensive empirical tuning to achieve optimal performance. In this experiment, we train for 500 epochs to study the loss-reduction behavior of AdaCubic. As shown in Figure 1, both Adam and AdaHessian exhibit a sharp decrease in training loss following a learning-rate reduction. However, such learning-rate scheduling strategies do not apply to AdaCubic, as the method does have an LR and its theoretical framework does not support analogous scheduling heuristics.
> Motivated by this distinction, we investigate the evolution of the training loss of AdaCubic over an extended number of epochs. An alternative approach would have been to conduct experiments without reducing the LR for any optimizers. However, our goal here was to achieve the best performance among the remaining optimizers that use a LR and compare them with AdaCubic. We hope this discussion clarifies the rationale behind the experimental design.
>
> **Reviewer Question (II)** *How are the (initial) learning rates of the optimizers SGD, Adam, and AdaHessian tuned? Most likely, these learning rates stem from the respective benchmark papers. However, making this process more transparent
> would be beneficial.*
>
> **Author response.** For CIFAR-10, the initial LR and LR schedule follow Yao et al. (2021). For the remaining benchmarks, learning rates and schedules were selected through extensive empirical tuning to achieve optimal performance. All learning rates are now summarized in Table 3 of the revised manuscript.
>
> **Reviewer Question (III)** *What are the effects of $\kappa_{\text{easy}}\in (0, 1)$ and how is its value defined?*
>
> **Author response.** We thank the reviewer for this comment. In the manuscript, the parameter $\kappa_{\text{easy}}\in(0,1)$ appears explicitly in Algorithm 2 and controls the stopping criterion of the inner root-finding procedure used to compute the cubic-regularized step. Specifically, $\kappa_{\text{easy}}$ defines a tolerance band around the target radius $\xi^{1/3}$, as shown in line 17 of Algorithm 2. Specifically, Newton updates of the dual variable $\nu$ are performed while the condition $\bigl| \lVert \mathbf{s} \rVert_2 - \xi^{1/3} \bigr|\;\le\; \kappa_{\text{easy}}\,\xi^{1/3}$ is satisfied.
>
> As a result, $\kappa_{\text{easy}}$ directly affects the numerical accuracy with which the constraint $|\mathbf{s}|_2^3=\xi$ is satisfied.
>
> Smaller values of $\kappa_{\text{easy}}$ enforce a stricter tolerance, leading to more Newton updates and a more accurate satisfaction of the cubic constraint. In comparison, larger values allow earlier termination of the inner solver and thus cheaper iterations.
>
> In all experiments, we fix $\kappa_{\text{easy}} = 10^{-2}$. This choice enforces a relative tolerance of 1% on the cubic subproblem constraint in Algorithm 2, ensuring that the computed step satisfies $\|\mathbf{s}\|_2^3 \approx \xi$ with sufficient accuracy.
>
> Such fixed relative tolerances are standard in trust-region and adaptive cubic regularization methods, where the theory explicitly allows inexact subproblem solutions, provided that the approximation error is controlled (Conn et al., 2000). Empirically, this value provides a good balance between numerical accuracy and computational efficiency and was kept fixed across all experiments. We have added $\kappa_{\text{easy}} = 10^{-2}$ at the beginning of Section 5 of the revised manuscript. In addition, $\kappa_{\text{easy}} = 10^{-2}$ is also mentioned in the Table 1 added in the revised manuscript. We hope this discussion clarifies the rationale behind the selection $\kappa_{\text{easy}} = 10^{-2}$.
>
> **Reviewer Question (IV)** *In the sentence above Corollary 3, you state that you repeat the proof with the diagonal Hessian matrix $\mathrm{Diag}(\nabla f(x))$ instead of the gradient $\nabla f(x)$. Do you rather mean $\mathrm{Diag}(\nabla^2 f(x))$ instead of the full Hessian matrix $\mathrm{Diag}(\nabla f(x))$?*
>
> **Author response.** We thank the reviewer for this comment, which is addressed in the revised manuscript.
>
> **Response to reviewer minor comments** All minor comments have been addressed.

---

### Comment · Action_Editor_9pqg · 2025-12-20
**Response to reviews**

Dear authors:

Thank you for your submission. Now that all three reviews are public and the discussion phase has started, please note that you have up to two weeks from the start of the discussion phase to post your author response. We encourage you to address all reviewer comments and questions as clearly and concretely as possible.

Best,
AC

---

> ### Author Response · Authors · 2026-01-06
>
> Dear Prof. Yi Zhou,
>
> On behalf of all authors, I would like to express my sincere gratitude to you and the reviewers for taking the time to evaluate our manuscript (submission number 6482). We greatly value the insightful feedback from the reviewers, which has significantly improved the quality and clarity of our work.
>
> In response to the reviewer’s request, the code has been made available at https://gitfront.io/r/mysubpapers/3F7bdJzvSvVn/AdaCubic/
>
> Sincerely
>
> The authors

---

### Decision · Action_Editor_9pqg · 2026-02-01

**Recommendation:** Accept as is

**Audience:**

Yes

**Audience Explanation:**

The paper proposed an adaptive cubic regularization algorithm that may be of interest to the optimization for machine learning community.

**Claims And Evidence:**

Yes

**Claims Explanation:**

This submission proposes AdaCubic, a well-motivated optimizer that brings cubic-regularized Newton ideas into practical deep-learning training by adaptively selecting the cubic regularization via an auxiliary constrained formulation. The paper provides a rigorous theoretical development and a careful implementation strategy using Hutchinson-style diagonal Hessian estimates to keep memory/computation scalable. Overall, the work is technically solid, clearly novel relative to existing second-order deep-learning optimizers, and likely to be of broad interest to both optimization and applied ML audiences.